# Escaping the Gravitational Pull of Softmax

**Jincheng Mei** [1] [4]     **Chenjun Xiao** [1] [4]     **Bo Dai** [4]     **Lihong Li** [3]*
**Csaba Szepesvári** [2] [1]     **Dale Schuurmans** [4] [1]

[1]University of Alberta   [2]DeepMind   [3]Amazon   [4]Google Research, Brain Team

{jmei2,chenjun,daes}@ualberta.ca   {bodai,szepi}@google.com   llh@amazon.com

## Abstract

The softmax is the standard transformation used in machine learning to map real-valued vectors to categorical distributions. Unfortunately, this transform poses serious drawbacks for gradient descent (ascent) optimization. We reveal this difficulty by establishing two negative results: (1) optimizing any expectation with respect to the softmax must exhibit sensitivity to parameter initialization ("softmax gravity well"), and (2) optimizing log-probabilities under the softmax must exhibit slow convergence ("softmax damping"). Both findings are based on an analysis of convergence rates using the Non-uniform Łojasiewicz (NŁ) inequalities. To circumvent these shortcomings we investigate an alternative transformation, the *escort* mapping, that demonstrates better optimization properties. The disadvantages of the softmax and the effectiveness of the escort transformation are further explained using the concept of NŁ coefficient. In addition to proving bounds on convergence rates to firmly establish these results, we also provide experimental evidence for the superiority of the escort transformation.

## 1   Introduction

The probability transformation plays an essential role in machine learning, used whenever the output of a learned model needs to be mapped to a probability distribution. For example, in reinforcement learning (RL), a probability transformation is used to parameterize policy representations that provide a conditional distribution over a finite set of actions given an input state or observation [16]. In supervised learning (SL), particularly classification, a probability transformation is used to parameterize classifiers that provide a conditional distribution over a finite set of classes given an input observation [7]. Attention models [19] also use probability transformations to provide differentiable forms of memory addressing.

Among the myriad ways one might map vectors to probability distributions, the *softmax* transform is the most common. For $\theta \in \mathbb{R}^K$, the transformation $\pi_\theta = \text{softmax}(\theta)$ is defined by $\pi_\theta(a) = \exp\{\theta(a)\}/\sum_{a'} \exp\{\theta(a')\}$ for all $a \in \{1, ..., K\}$, which ensures $\pi_\theta(a) > 0$ and $\sum_a \pi_\theta(a) = 1$ [4]. The softmax transform can also be extended to continuous output spaces through the concept of a Gibbs function [11], but for concreteness we restrict attention to finite output sets.

Despite the ubiquity of the softmax in machine learning, it is not clear why it should be the default choice of probability transformation. Some alternative transformations have been investigated in the literature [6, 10], but a comprehensive understanding of why one choice might be advantageous over another remains incomplete. It is natural to ask what options might be available and what properties are desirable. In fact, we find that the softmax is a particularly *undesirable* choice from the perspective of gradient descent (ascent) optimization. Moreover, better alternatives are readily available at no

computational overhead. This paper seeks to fill the gap in understanding key properties of probability transformations in general and how they compare to the softmax.

We start by considering *reinforcement learning* and investigate gradient ascent optimization of expected reward using the softmax transform, an algorithm we refer to as softmax policy gradient (SPG) [1, 12]. In this setting, we identify an inherent disadvantage of SPG, the "softmax gravity well (SGW)", whereby gradient ascent trajectories are drawn toward suboptimal corners of the probability simplex and subsequently slowed in their progress toward the optimal vertex. We establish these facts both through theoretical analysis and empirical observation, revealing that the behavior of SPG depends strongly on initialization. Then we propose the use of the *escort* transform as an alternative to softmax for expected reward optimization. We analyze the resulting gradient ascent algorithm, escort policy gradient (EPG), and prove that it enjoys *strictly* better convergence behavior than SPG, significantly mitigating sensitivity to initialization. These findings are also verified experimentally.

Next we consider *supervised learning* and investigate gradient descent optimization of cross entropy loss using the softmax transform, an algorithm we refer to as softmax cross entropy (SCE). Here, even though the optimization landscape at the output layer is convex, we identify a detrimental phenomenon we refer to as "softmax damping". In particular, given deterministic ("one-hot") true label distributions, we show that SCE achieves a slower than linear rate of convergence. Then we propose the use of the escort transform as an alternative to softmax for cross entropy minimization. We analyze the resulting gradient descent algorithm, escort cross entropy (ECE), and show that it is guaranteed to enjoy *strictly* faster convergence than SCE. In particular, a special choice of the escort transform fully eliminates the softmax damping phenomenon, preserving the linear convergence rate for cross entropy minimization.

Finally we propose a unifying concept, the Non-uniform Łojasiewicz (NŁ) coefficient, to explain both the softmax gravity well and softmax damping, even when these might otherwise appear to be disconnected phenomena. We show that by increasing the NŁ coefficient, EPG achieves strictly better initialization dependence than SPG. Moreover, by making the NŁ coefficient non-vanishing, ECE enjoys strictly faster convergence than SCE.

## 2   Illustrating the Softmax Gravity Wells with Softmax Policy Gradient

We begin by considering the domain of reinforcement learning (RL), where the goal is to learn a policy that maximizes expected return. A core method in RL is *policy gradient* [17], where a parameterized policy is directly optimized to maximize long-term expected reward. It is conventional in this area to represent parametric policies using a softmax transform to produce conditional action distributions, hence policy gradient in practice is almost always the *softmax* policy gradient (SPG).

Despite the fact that SPG has been a dominant RL method for decades, only recently has it been proved to be globally convergent for general MDPs [1]. This result is far from obvious, since the optimization objective is not concave, nevertheless it was shown that SPG converges to the optimal policy under general conditions [1]. More recently, this result was strengthened to establish a $\Theta(1/t)$ bound on the *rate of convergence* [12], with constants that depend on the problem and initialization.

Although these theoretical results are general and impressive, they seem at odds with the behavior of policy gradient methods, which are notoriously difficult to tune in practice [15]. To reconcile theory with empirical observation, we first demonstrate that the "constants" in these results are in fact important, and understanding their role explains much of the real-world performance of SPG.

**Illustration**   To illustrate the point concretely, consider a simple experiment on a single-state Markov Decision Process (MDP) (i.e., a multi-armed bandit) with $K = 6$ actions. In this case, the SPG of a policy $\pi_\theta$ for a given reward vector $r \in [0,1]^K$ reduces to the update $\theta_{t+1}(a) = \theta_t(a) + \eta \cdot \pi_{\theta_t}(a) \cdot \left[ r(a) - \pi_{\theta_t}^\top r \right], \forall a \in [K] := \{1, ..., K\}$, and $\pi_{\theta_{t+1}} = \text{softmax}(\theta_{t+1})$. Fig. 1 shows the result of multiple runs using SPG with full gradients. Depending on whether the last iteration satisfies $\pi_{\theta_T}(a^*) \geq 0.99$, we group the 20 runs as "good" and "bad" initializations. As shown in Fig. 1(a) and (b), for good initializations, the sub-optimality $(\pi^* - \pi_{\theta_t})^\top r$ quickly approaches 0, whereas for bad initializations, the iterates get stuck near local optima. Subfigure (c) shows average probability of optimal actions, which shows that the trajectories from bad initializations stay near local optima, since the optimal action probabilities stay close to 0. However, we know from the theory that from any initializations SPG must *eventually* converge to the optimal policy $\pi^*$, and that is indeed the case here: Subfigure (d) shows the long-run time to convergence (boxes are 25 to 75th

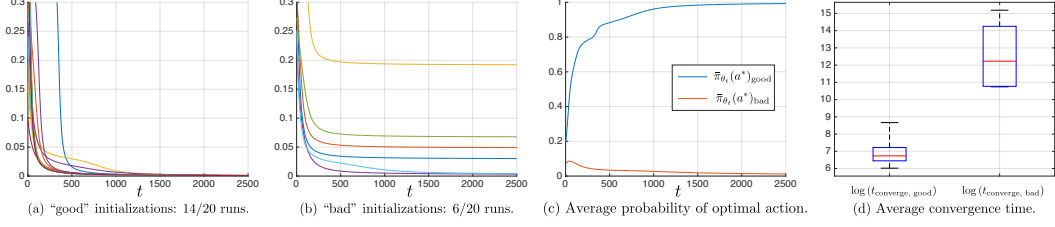

(a) "good" initializations: 14/20 runs.  (b) "bad" initializations: 6/20 runs.  (c) Average probability of optimal action.  (d) Average convergence time.

Figure 1: SPG behavior on single-state MDPs with $K = 6$ arms, fully parameterized policy (no approximation error), rewards randomly generated (uniform within $[0, 1]$ for each $r(a)$) and policy randomly initialized on each run, 20 runs. Full gradient SPG updates with stepsize $\eta = 0.4$ [12] for $T = 3 \times 10^4$ steps. An initialization is "good" if $\pi_{\theta_T}(a^*) \geq 0.99$ at the last iterate.

percentiles) for good versus bad initializations, where the y-axis is $\log T$ such that $\pi_{\theta_T}(a^*) \geq 0.99$, showing bad runs take *many orders of magnitude* longer.

Although these findings seem not to comport with theory, they can in fact be explained by delving deeper into the detailed nature of the $\Theta(1/t)$ rates proved in [12].

**Escape time** To control the effect of initialization, consider a specialization of the previous problem where we let $r = (b + \Delta, b, \dots, b)^\top \in [0, 1]^K$ for some $b$, such that $\Delta > 0$ is the reward gap. For a given initialization, we say that SPG "escapes" at time $t_0$ if for all $t \geq t_0$ it holds that $(\pi^* - \pi_{\theta_t})^\top r < 0.9 \cdot \Delta$, i.e., after $t_0$ the sub-optimality stays "small". Fig. 2(a) shows that as the initial probability of the optimal action $\pi_{\theta_1}(a^*)$ decreases, the "escape time" $t_0$ increases proportionally. In particular, the slope in subfigure (a) approaches $-1$ as $\pi_{\theta_1}(a^*)$ decreases, indicating that $\log t_0 = -\log \pi_{\theta_1}(a^*) + C$, or equivalently $t_0 = C'/\pi_{\theta_1}(a^*)$. Two trajectories for SPG on a single-state MDP with $K = 5$ is shown in Fig. 2(b) and (c). This example reveals that every suboptimal vertex $i \in \{2, 3, 4\}$ has the potential to attract the iterates, while also slowing progress to render the sub-optimality plateaus in subfigure (c). Therefore, SPG spends some "escape time" around each suboptimal corner.

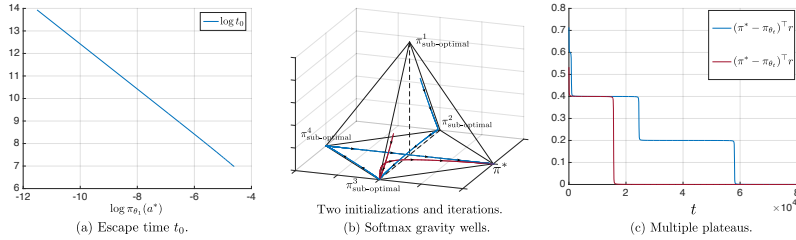

(a) Escape time $t_0$.  (b) Softmax gravity wells.  (c) Multiple plateaus.

Figure 2: Dependence on initialization and softmax gravity wells.

We can see as SPG follows a trajectory defined by exact gradients, it effectively encounters "Softmax Gravity Wells (SGWs)" at the vertices (deterministic policies), each of which attracts the trajectory and significantly slows down progress in their vicinity. To see why the attraction to suboptimal vertices is possible, consider the SPG in detail: for a single-state MDP, $\forall a \in [K]$, we have

$$\frac{d\pi_\theta^\top r}{d\theta(a)} = \pi_\theta(a) \cdot \left[ r(a) - \pi_\theta^\top r \right]. \tag{1}$$

Note that it is possible for an optimal action, say $a_1$, to be less attractive than a suboptimal action $a_2$, even when $r(a_1) > r(a_2)$, since it is possible to have both $r(a_1) - \pi_{\theta_t}^\top r > r(a_2) - \pi_{\theta_t}^\top r > 0$ and $\pi_{\theta_t}(a_2) > \pi_{\theta_t}(a_1)$, and yet still have $\pi_{\theta_t}(a_2) \cdot \left[ r(a_2) - \pi_{\theta_t}^\top r \right] > \pi_{\theta_t}(a_1) \cdot \left[ r(a_1) - \pi_{\theta_t}^\top r \right]$. This configuration causes the probability on the suboptimal action to stay above the optimal action probability, $\pi_{\theta_{t+1}}(a_2) > \pi_{\theta_{t+1}}(a_1)$. Even though the examples and analysis above might seem specific, they provide the foundation for a useful and informative lower bound.

**Theorem 1** (Escape time lower bound). *Even in a single-state MDP, for any learning rate $\eta_t \in (0, 1]$, there exists an initialization of the policy $\pi_{\theta_1}$ and a positive constant $C$, such that SPG with full gradients cannot escape a suboptimal corner before time $t_0 := \frac{C}{\Delta \cdot \pi_{\theta_1}(a^*)}$, i.e., it will hold that*

$$(\pi^* - \pi_{\theta_t})^\top r \geq 0.9 \cdot \Delta, \tag{2}$$

*for all $t \leq t_0$, where $\Delta := r(a^*) - \max_{a \neq a^*} r(a) > 0$ is the reward gap of $r \in [0,1]^K$.*

Theorem 1 shows that for SPG with bounded learning rates (needed for monotonic improvement [1, 12]) the time to escape suboptimal vertices is lower bounded inversely to optimal action probability $\pi_\theta(a^*)$, which is necessarily small near suboptimal vertices, leading to long suboptimal plateaus.

**Failure of SPG heuristics**   Given the insight from Theorem 1, one might wonder if simple heuristics can compensate for the slow progress of SPG, possibly by using large learning rates or normalizing the policy gradient. We show in the appendix that these heuristics unfortunately do not work well even in simple bandit problems.

**Existing observations of plateaus**   SPG plateaus have been observed in the literature. Previous work [12] did observe this effect empirically, but did not take a deeper look into the underlying causes. With function approximation, feature interference has also been considered to be a source of plateaus [14]. In the multi-agent setting, it has been observed that the non-stationary nature of the environment can also cause difficulties for SPG to adapt [8]. However, the analysis in this paper shows that SPG still suffers from plateaus even in the simplest setting (exact gradients, no approximation, stationary environments). In Section 4 we provide additional mathematical insight to explain why the softmax transformation itself is the root cause, which also justifies the name SGW.

## 3   Escort Transform for Policy Gradient

As explained, a difficulty encountered by SPG comes from the $\pi_\theta(a)$ factor that appears in the gradient, Eq. (1). This creates a dependence on the current policy that potentially discounts the signal from high-reward actions. Unfortunately, the problem is unavoidable if using SPG with bounded learning rates to perform updates (Theorem 1). Therefore, we study the following alternative transform, which we refer to as the "escort transform" [3, 18].

**Definition 1** (Escort transform). *Given $\theta : \mathcal{S} \times \mathcal{A} \to \mathbb{R}$, define $\pi_\theta = f_p(\theta)$ for $p \geq 1$ by*

$$\pi_\theta(a|s) = \frac{|\theta(s,a)|^p}{\sum_{a' \in \mathcal{A}} |\theta(s,a')|^p}, \quad \text{for all } (s,a) \in \mathcal{S} \times \mathcal{A}. \tag{3}$$

*If there is only one state, the escort transform is defined as $\pi_\theta(a) = |\theta(a)|^p / \sum_{a'} |\theta(a')|^p$, $\forall a \in [K]$.*

To explain why this alternative transform might help alleviate the problems encountered by the softmax, consider the gradient of expected reward using the escort transform, i.e., the Escort Policy Gradient (EPG), for a single-state MDP, $\forall a \in [K]$ (detailed calculations are shown in the appendix):

$$\frac{d\pi_\theta^\top r}{d\theta(a)} = p \cdot \text{sgn}\{\theta(a)\} \cdot \frac{|\theta(a)|^{p-1}}{\sum_{a'} |\theta(a')|^p} \cdot \left[ r(a) - \pi_\theta^\top r \right] \tag{4}$$

$$= \frac{p}{\|\theta\|_p} \cdot \text{sgn}\{\theta(a)\} \cdot \pi_\theta(a)^{1-1/p} \cdot \left[ r(a) - \pi_\theta^\top r \right]. \tag{5}$$

Note the key difference between SPG and EPG, in which the $\pi_\theta(a)$ term in Eq. (1) now becomes $\pi_\theta(a)^{1-1/p}$ in Eq. (5). Thus, for any $p \geq 1$, we have $1 - 1/p \in [0, 1)$, which implies $\pi_\theta(a)^{1-1/p} > \pi_\theta(a)$ since $\pi_\theta(a) \in [0, 1]$. This change will have important implications in convergence rate.

**Remark 1.** *$\pi_\theta(a)^{1-1/p} \to \pi_\theta(a)$ as $p \to \infty$, which suggests that large values of $p$ lead to similar iteration behavior as SPG, whereas small values of $p$ weaken the dependence on $\pi_\theta(a)$. In particular, if $p = 1$ then $\pi_\theta(a)^{1-1/p} = 1$, which entirely eliminates the dependence on current policy $\pi_\theta$.*

As is the case for the softmax transform, the expected reward objective remains non-concave over parameter $\theta$ when using the alternative escort transform.

**Proposition 1.** *$\theta \mapsto \pi_\theta^\top r$ is a non-concave function over $\mathbb{R}^K$ using the map $\pi_\theta := f_p(\theta)$.*

Despite the non-concavity, we manage to obtain surprisingly strong convergence results for EPG, with proofs provided in the appendix. In particular, thanks to what we call non-uniform smoothness and the Non-uniform Łojasiewicz (NŁ) inequality enjoyed by the objective, EPG is shown to enjoy an upper bound on the sub-optimality for single-state MDPs that has a strictly better initialization dependence than SPG.

**Theorem 2.** *For a single-state MDP, following the escort policy gradient with any initialization such that $|\theta_1(a)| > 0$, $\forall a$, we obtain the following upper bounds on the sub-optimality for all $t \geq 1$:* [2]

*(gradient ascent) for $p \geq 2$, $p = 1$, with $\eta_t = \dfrac{2 \cdot \|\theta_t\|_p^2}{9 \cdot p^2 \cdot K^{1/p}}$,*   $(\pi^* - \pi_{\theta_t})^\top r \leq \dfrac{9 \cdot K^{1/p}}{c^{2-2/p}} \cdot \dfrac{1}{t}$,

*(gradient flow) for $p \geq 1$, with $\eta_t = \dfrac{\|\theta_t\|_p^2}{p^2}$,*   $(\pi^* - \pi_{\theta_t})^\top r \leq \dfrac{1}{c^{2-2/p} \cdot (t-1) + 1}$,

*where $c := \inf_{t \geq 1} \pi_{\theta_t}(a^*) > 0$ depends on the problem and initialization, but is time-independent.*

As $p \to \infty$, Theorem 2 implies an $O(1/(c^2\, t))$ convergence rate, recovering the same rate for SPG [12], as expected (Remark 1). For $p < \infty$, EPG achieves the same $O(1/t)$ rate as SPG, but enjoys a strictly better $c^{2-2/p} > c^2$ dependence. In particular, for $p = 1$, there is no dependence on $c$, which is also consistent with Remark 1. On the other hand, $K^{1/p} \to K$ increases as $p$ decreases, which means it is not always good to choose small $p$ values due to trade-off between $K$ and $c$.

Similar results can in fact be obtained for EPG in general finite MDPs, denoted as $\mathcal{M} = (\mathcal{S}, \mathcal{A}, r, \mathcal{P}, \gamma)$, where $\mathcal{S}$ and $\mathcal{A}$ are finite state and action spaces, $r : \mathcal{S} \times \mathcal{A} \to \mathbb{R}$ is the reward function, $\mathcal{P} : \mathcal{S} \times \mathcal{A} \to \Delta(\mathcal{S})$ is the transition function, $\Delta(\mathcal{X})$ denotes the set of probability distributions over any finite set $\mathcal{X}$, and $\gamma \in [0, 1)$ is the discount factor. Let $V^\pi(\rho) := \mathbb{E}_{s_0 \sim \rho(\cdot), a_t \sim \pi(\cdot|s_t), s_{t+1} \sim \mathcal{P}(\cdot|s_t, a_t)} \sum_{t=0}^\infty \gamma^t r(s_t, a_t)$ denote the expected return (value function) achieved by policy $\pi$, where $\rho \in \Delta(\mathcal{S})$ is an initial state distribution. The goal is to maximize the value function, i.e., to find a policy $\pi^*$ that attains the value $V^*(\rho) := \max_{\pi: \mathcal{S} \to \Delta(\mathcal{A})} V^\pi(\rho)$.

**Theorem 3.** *Following the escort policy gradient with any initialization such that $|\theta_1(s, a)| > 0$, $\forall (s, a)$, and $\eta_t(s) = \dfrac{(1-\gamma)^3 \cdot \|\theta_t(s, \cdot)\|_p^2}{10 \cdot p^2 \cdot A^{2/p}}$ to get $\{\theta_t\}_{t \geq 1}$, for all $t \geq 1$, the following sub-optimality upper bounds hold for $\pi_{\theta_t}$,*

$$\text{for } p \geq 2 \text{ and } p = 1, \qquad V^*(\rho) - V^{\pi_{\theta_t}}(\rho) \leq \frac{20 \cdot A^{2/p} \cdot S}{c^{2-2/p} \cdot (1-\gamma)^6 \cdot t} \cdot \left\| \frac{d_\mu^{\pi^*}}{\mu} \right\|_\infty^2 \cdot \left\| \frac{1}{\mu} \right\|_\infty,$$

*where $c := \inf_{s \in \mathcal{S}} \inf_{t \geq 1} \pi_{\theta_t}(a^*(s)|s) > 0$ is problem- and initialization-dependent constant, $A := |\mathcal{A}|$ and $S := |\mathcal{S}|$ are the total number of actions and states, respectively, and $\mu \in \Delta(\mathcal{S})$ is an initial state distribution which provides initial states for the policy gradient method.*

**Remark 2.** *Using $p = 1$ in Theorem 3, the iteration complexity of EPG depends on polynomial functions of $S$ and $A$, which significantly improves the corresponding results for SPG [12, Theorem 4], where the worst case dependence can be exponential in $S$ and $A$.*

Finally, as for SPG, adding entropy regularization leads to linear convergence rates for EPG. Note that SPG with entropy regularization enjoys a linear convergence rate $O(1/\exp\{c^2 t\})$ with dependence on $c = \inf_{t \geq 1} \min_{(s, a)} \pi_{\theta_t}(a|s)$ [12]. Our results show that EPG with entropy regularization has strictly better dependence than SPG.

**Theorem 4.** *For an entropy regularized MDP with finite states and actions, following the escort policy gradient with any initialization such that $|\theta_1(s, a)| > 0$, $\forall (s, a)$, and $\eta_t = (1-\gamma)^3 / (10 \cdot p^2 \cdot A^{1/p} + c_\tau)$ to get $\{\theta_t\}_{t \geq 1}$, for all $t \geq 1$, the following sub-optimality upper bounds hold for $\pi_{\theta_t}$:*

$$\text{for } p \geq 2, \qquad \tilde{V}^{\pi_\tau^*}(\rho) - \tilde{V}^{\pi_{\theta_t}}(\rho) \leq \frac{\|1/\mu\|_\infty}{\exp\{C_\tau \cdot c'^2 \cdot t\}} \cdot \frac{1 + \tau \log A}{(1-\gamma)^2}, \tag{6}$$

*where $c' > c := \inf_{(s, a)} \inf_{t \geq 1} \pi_{\theta_t}(a|s) > 0$, $\tau$ is the temperature for entropy regularization, $\pi_\tau^*$ is the softmax optimal policy, and $c_\tau$, $C_\tau$ are problem-dependent constants.*

**Relationship to Mirror Descent (MD)**   As an additional observation, note that simply removing $\pi_\theta(a)$ in Eq. (1) yields an update $\theta_{t+1} = \theta_t(a) + \eta_t \cdot r(a)$ and $\pi_{\theta_{t+1}} = \text{softmax}(\theta_{t+1})$, which can be combined to yield an update $\pi_{\theta_{t+1}}(a) \propto \pi_{\theta_t}(a) \cdot \exp\{\eta_t\, r(a)\}$ that is equivalent to Mirror Descent

(MD) with KL regularization. Given this similarity between SPG, EPG and MD, one might hope that EPG could be reduced to a particular version of MD. However, unlike SPG and MD, the EPG gradient does not specify a conservative vector field and cannot be recovered by MD using any regularization.

**Remark 3** (EPG cannot be reduced to MD). *Recall that for a (convex) potential $\Phi : \Delta \to \mathbb{R}$ and its Bregman divergence $D_\Phi : \Delta \times \Delta \to \mathbb{R}$, the MD update is $\pi_{t+1} = \arg\max_{\pi \in \Delta} \pi^\top r - (1/\eta_t) D_\Phi(\pi \| \pi_t)$. In particular, using $\Phi(\pi) = \pi^\top \log \pi$ as the potential and $D_\Phi(\pi \| \pi') = D_{\mathrm{KL}}(\pi \| \pi')$ as the divergence one obtains $\pi_{\theta_{t+1}}(a) \propto \pi_{\theta_t}(a) \cdot \exp\{\eta_t \, r(a)\}$. Equivalently, this update can be expressed $\pi_{\theta_{t+1}} = \arg\max_{\pi \in \Delta} \pi^\top \theta_{t+1} - \Phi(\pi)$ where $\theta_{t+1} = \theta_t(a) + \eta_t \cdot r(a)$.*

*Now suppose EPG is MD, i.e., there is some $\Phi$ such that $f_p(\theta_{t+1}) = \arg\max_{\pi \in \Delta} \pi^\top \theta_{t+1} - \Phi(\pi)$. Then we would have to have $f_p(\theta_{t+1}) = \nabla \Phi^*(\theta_{t+1})$ where $\Phi^*$ is the Fenchel conjugate of $\Phi$. Taking the derivative w.r.t. $\theta$ yields*

$$\left( \frac{d\pi_\theta}{d\theta} \right)^\top = \left( \frac{df_p(\theta)}{d\theta} \right)^\top = p \cdot diag(1/\theta) \left( diag(\pi_\theta) - \pi_\theta \pi_\theta^\top \right) \overset{(?)}{=} \frac{d^2 \Phi^*(\theta)}{d\theta^2}. \tag{7}$$

*By Schwarz's theorem, $\frac{d^2 \Phi^*(\theta)}{d\theta^2}$ must be symmetric, however $diag(1/\theta) \left( diag(\pi_\theta) - \pi_\theta \pi_\theta^\top \right)$ is not symmetric. Therefore, there cannot be a regularizer $\Phi$ that makes EPG equivalent to MD.*

Remark 3 implies that standard techniques for analyzing mirror descent (e.g., Bregman divergence and convex duality) cannot be directly applied to EPG, necessitating our analysis based on the non-uniform smoothness and NŁ inequalities for Theorems 2 to 4.

**Experimental Verification**   To support these findings and reveal some of the practical implications of EPG versus SPG, we conducted a simple experiment on a single-state MDP with $K = 3$ and $r = (0.2, 0.9, 1.0)^\top$. Fig. 3(a) depicts the $\frac{d\pi_\theta(a^*)}{dt}$ values for SPG, where the dark regions around the corners show areas of slow progress. In particular, the region around the lower-right suboptimal corner exhibits $\frac{d\pi_\theta(a^*)}{dt} < 0$, and $\pi_\theta(a^*)$ will actually *decrease* under SPG updating in this region, prolonging the escape time according to Theorem 1. In short, the dark regions correspond to SGWs for SPG. Subfigure (b) further shows how SPG is attracted toward the suboptimal corner, visually consistent with subfigure (a). By contrast, the solid lines indicate EPG methods with different $p$ values. As noted in Remark 1, smaller $p$ values have better resistance against attraction to SPG gravity wells, while larger $p$ values behave more similarly to SPG. We also observe that MD (with KL regularization) has similar performance to EPG with $p = 2$ in this case. Finally, Subfigure (c) plots the suboptimaliy gap before $(\pi^* - \pi_{\theta_t})^\top r \leq 0.005$ is achieved. It is clear that SPG does get stuck on a suboptimal plateau while EPG methods do not suffer from this disadvantage. We note that EPG curves for $p \geq 2$ behave nicer than $p = 1$ since the escort is differentiable when $p \geq 2$.

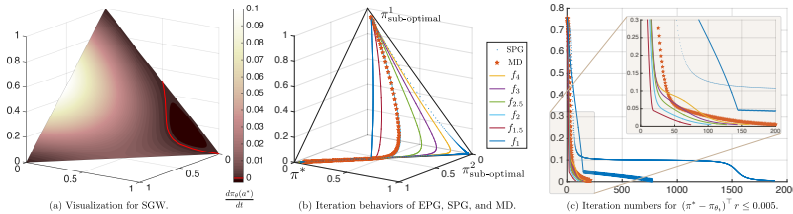

(a) Visualization for SGW.   (b) Iteration behaviors of EPG, SPG, and MD.   (c) Iteration numbers for $(\pi^* - \pi_{\theta_t})^\top r \leq 0.005$.

Figure 3: *Empirical visualization for EPG and SPG.*

## 4   Non-uniform Łojasiewicz Coefficient: An Underlying Explanation

Remark 1 provides an intuition for why EPG has better initialization dependence than SPG. This intuition can be formalized using the notion of Non-uniform Łojasiewicz (NŁ) coefficient, which plays an important role here since both SPG [12] and EPG analyses are based on NŁ inequalities.

**Definition 2** (Non-uniform Łojasiewicz (NŁ) coefficient). *A function $f : \mathcal{X} \to \mathbb{R}$ has NŁ coefficient $C(x) > 0$ if it satisfies NŁ inequality with coefficient $C(x)$, i.e., there exists $\xi \in [0, 1]$ such that for all $x \in \mathcal{X}$, $\|\nabla f(x)\|_2 \geq C(x) \cdot |f(x) - f(x^*)|^{1-\xi}$.*

In Definition 2, $\xi$ is called NŁ degree, which impacts the convergence rates of SPG methods [12, Definition 1]. From a result in [12], if $\pi_\theta = \mathrm{softmax}(\theta)$, then $\pi_\theta^\top r$ has NŁ coefficient $\pi_\theta(a^*)$; that is

$$\left\| \frac{d\pi_\theta^\top r}{d\theta} \right\|_2 = \left\| \left( \mathrm{diag}(\pi_\theta) - \pi_\theta \pi_\theta^\top \right) r \right\|_2 \geq \pi_\theta(a^*) \cdot (\pi^* - \pi_\theta)^\top r. \tag{8}$$

Moreover, this coefficient is not improvable and it appears in the SPG convergence rate $O(1/(c^2\,t))$, where $c := \inf_{t \geq 1} \pi_{\theta_t}(a^*)$. Now consider EPG. If $\pi_\theta = f_p(\theta)$, then we have

$$\left\| \frac{d\pi_\theta^\top r}{d\theta} \right\|_2 = \left\| p \cdot \mathrm{diag}(1/\theta) \left( \mathrm{diag}(\pi_\theta) - \pi_\theta \pi_\theta^\top \right) r \right\|_2 \geq p \cdot \frac{\pi_\theta(a^*)}{|\theta(a^*)|} \cdot (\pi^* - \pi_\theta)^\top r. \tag{9}$$

This implies that $\frac{\pi_\theta(a^*)}{|\theta(a^*)|} = \frac{1}{\|\theta\|_p} \cdot \pi_\theta(a^*)^{1-1/p}$, where $\pi_\theta(a^*)^{1-1/p} > \pi_\theta(a^*)$ provides strictly larger (partial) NŁ coefficient, hence in Theorem 2 EPG obtains a strictly better result than SPG.

The improvement of NŁ coefficient explains a better dependence of EPG on initialization. It is then natural to ask whether the escort transform can also benefit other scenarios, which is answered affirmatively in the next section.

## 5 Escort Transform for Cross Entropy

We now turn to classification, where the goal is to learn a classifier that minimizes the cross-entropy loss. As in RL, the softmax transform is the default choice for parameterizing a probabilistic classifier. Different from RL where the objective is linear, the objective here involves log probabilities:

$$\min_{\theta:\mathcal{A}\to\mathbb{R}} - \log \pi_\theta(a_y) = \mathcal{H}(y) + \min_{\theta:\mathcal{A}\to\mathbb{R}} D_{\mathrm{KL}}(y\|\pi_\theta), \tag{10}$$

where $\pi_\theta = \mathrm{softmax}(\theta)$, $y \in \{0,1\}^K$ is a one-hot vector encoding the class label, and $a_y$ is the true label class so that $y(a_y) = 1$. Note that the entropy $\mathcal{H}(y) = 0$ here. The objective in Eq. (10) is smooth and convex in $\theta$, which implies that gradient descent will achieve an $O(1/t)$ rate [13]. Furthermore, for $\theta$ that satisfies $\min_a \pi_\theta(a) \geq \pi_{\min}$ with some constant $\pi_{\min} > 0$ ($\pi_\theta$ is bounded away from the simplex boundary), the objective is strongly convex, resulting in an even better, linear rate $O(e^{-t})$. Despite these nice properties, we still find that the softmax transform proves problematic for gradient descent optimization. We refer to this new disadvantage as "softmax damping".

**Illustration**   Consider running gradient descent in a simple experiment where $K = 10$ and $y$ a one-hot vector. Let $\delta_t := -\log \pi_{\theta_t}(a_y)$. If one hopes for a linear convergence rate, i.e., $\delta_t = O(e^{-t})$, then $\log \delta_t = -O(t)$ is expected. But Fig. 4(a) shows a different picture with a flattening slope. Subfigure (b) plots $\log \delta_t$ as a function of $\log t$, which shows a straight line for sufficiently large $t$ with a slope approaching $-1$. This figure verifies the convergence rate is indeed $\delta_t = O(1/t)$, instead of the linear $O(e^{-t})$ rate. Subfigure (c) shows the $\ell_2$ measure $\|y - \pi_{\theta_t}\|_2^2$ also has a similar rate, indicating that this is an inherent optimization phenomenon and is independent of the measurement.

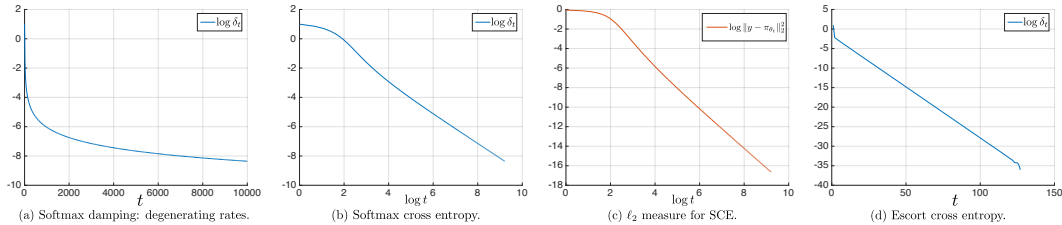

(a) Softmax damping: degenerating rates.   (b) Softmax cross entropy.   (c) $\ell_2$ measure for SCE.   (d) Escort cross entropy.

Figure 4: Softmax damping phenomenon and escort cross entropy.

**NŁ Coefficient Explanation**   The NŁ coefficient can be used to explain why this rate degeneration occurs for softmax cross entropy (SCE). Note that for $\pi_\theta = \mathrm{softmax}(\theta)$ we obtain

$$\left\| \frac{d\{D_{\mathrm{KL}}(y\|\pi_\theta)\}}{d\theta} \right\|_2^2 = \|y - \pi_\theta\|_2^2 \geq \min_a \pi_\theta(a) \cdot D_{\mathrm{KL}}(y\|\pi_\theta). \tag{11}$$

Once again the $\min_a \pi_\theta(a)$ term cannot be eliminated for the softmax transform, but here it has a different consequence than before. To see the NŁ coefficient of SCE cannot be improved, consider the example where $y = (0,1)^\top$ and $\pi = (\epsilon, 1-\epsilon)^\top$, where $\epsilon > 0$ is small. Note that $D_{\mathrm{KL}}(y\|\pi) = -\log(1-\epsilon) \geq \epsilon$ and $\|y-\pi\|_2^2 = 2\epsilon^2$, which means for any constant $C > 0$, we have $C \cdot D_{\mathrm{KL}}(y\|\pi) \geq C \cdot \epsilon > 2\epsilon^2 = \|y-\pi\|_2^2$. Therefore, for any Łojasiewicz-type inequality, $C$ necessarily depends on $\min_a \pi_\theta(a)$. Now for any convergent sequence $\pi_{\theta_t}$, i.e., such that $D_{\mathrm{KL}}(y\|\pi_{\theta_t}) \to 0$, we necessarily have $\min_a \pi_{\theta_t}(a) \to 0$, which makes the gradient information insufficient to sustain a linear convergence rate. That is, the fast convergence rate is "damped" in this case. The difference between this phenomenon and the previous "softmax gravity well" is that here the vanishing NŁ coefficients change the rates rather than the constant in the bound on the sub-optimality gap.

**Escort Cross Entropy**   As in Section 3 for policy gradient, we propose to also use the escort transform for cross entropy minimization. A simple calculation for $\pi_\theta = f_p(\theta)$ shows

$$\left\| \frac{d\{D_{\mathrm{KL}}(y\|\pi_\theta)\}}{d\theta} \right\|_2^2 = \|p \cdot \mathrm{diag}(1/\theta)(y-\pi_\theta)\|_2^2 \geq \frac{p^2}{\|\theta\|_p^2} \cdot \min_a \pi_\theta(a)^{1-2/p} \cdot D_{\mathrm{KL}}(y\|\pi_\theta). \quad (12)$$

Note that the term $\min_a \pi_\theta(a)^{1-2/p} > \min_a \pi_\theta(a)$ is strictly better than the softmax cross entropy when $p \geq 2$. In particular, for $p = 2$, the escort cross entropy (ECE) has (partial) NŁ coefficient $\min_a \pi_\theta(a)^{1-2/p} = 1$, which fully eliminates the dependence on the current policy $\pi_\theta$. This leads to our last main result, which restores the linear convergence rate.

**Theorem 5.** *Using the escort transform with $p = 2$ and gradient descent on the cross entropy objective with learning rate $\eta_t = \frac{\|\theta_t\|_p^2}{4\cdot(3+c_1^2)}$, we obtain for all $t \geq 1$,*

$$-\log \pi_{\theta_t}(a_y) = D_{\mathrm{KL}}(y\|\pi_{\theta_t}) \leq D_{\mathrm{KL}}(y\|\pi_{\theta_1}) \cdot \exp\left\{ -\frac{(t-1)}{2\cdot(3+c_1^2)} \right\}, \quad (13)$$

*where $1/c_1^2 = \pi_{\theta_1}(a_y) \in (0,1]$ only depends on initialization.*

For reference, we run gradient descent on the cross entropy objective in the same experiment above, but with the escort transform. As shown in Fig. 4(d), $\log \delta_t$ now becomes linear in $t$, or equivalently $-\log \pi_{\theta_t}(a_y) = C \cdot e^{-c \cdot t}$, verifying the theoretical finding of Theorem 5.

# 6   Experimental Results

We conduct several experiments to verify the effectiveness of the proposed escort transform in policy gradient and cross entropy minimization.

First, we conduct experiments on one-state MDPs with $K = 10, 50$, and $100$, and $20$ runs for each $K$ value and for each algorithm. In each run, the reward $r \in [0,1]^K$ and $\pi_{\theta_1}$ are randomly generated. The total iteration number $T = 5 \times 10^4$. As shown in Fig. 5(a), EPG with $p = 2$ quickly converges to optimal policies consistently across all the $K$ values, significantly outperforming SPG.

Second, we compare the algorithms on Four-room environment for $20$ runs. There is one goal with reward $1.0$ and $4$ sub-goals ("sub-goals" mean goals with lower rewards) with reward $0.7$ as shown in Fig. 5(b). At a (sub-)goal state, the agent can step away then step back to receive rewards. The policy is parameterized by one hidden layer ReLU neural network with $64$ hidden nodes. In each run, the starting position is randomly generated. We use value iteration to calculate $V^*$ and we calculate the true gradient by closed form. As shown in Fig. 5(c), SPG is easily stuck in plateaus due to the presence of the sub-goals, while EPG with $p = 2$ quickly achieves the optimal goal.

Next, we do experiments on MNIST dataset. For each $(x,y)$, where $x \in \mathbb{R}^{784}$ is image data and $y \in \{0,1\}^{10}$ is the true label, the training objective is $1 - \pi_\theta(a_y|x)$, where $y(a_y) = 1$. We use policy gradient methods, since the mis-classification probability minimization problem is a special case of expected reward maximization. We use one hidden layer neural network with $512$ hidden nodes and ReLU activation to parameterize $\theta$. The dataset is split into training set with $55000$, validation set with $5000$, and testing set with $10000$ data points. As shown in Fig. 6(a) and (b), for both training objective and test error, SPG has plateaus due to SGWs, which is consistent with the observation in [5]. At the same time, EPG with $p = 4$ does not have this disadvantage: it converges quickly and achieves better results than SPG. Experiments for other $p$ values are shown in the appendix.

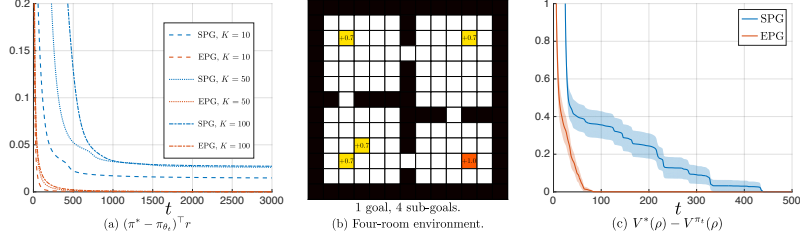

(a) $(\pi^* - \pi_{\theta_t})^\top r$      (b) Four-room environment.      (c) $V^*(\rho) - V^{\pi_t}(\rho)$

Figure 5: *Results on one-state MDPs and Four-room.*

We use mini-batch stochastic gradient descent in this experiment, and the results show that with stochastic gradients and neural network function approximations, (1) SPG still plateaus even when starting from nearly uniform initializations; (2) EPG outperforms SPG in terms of not suffering from plateaus even with estimated gradients.

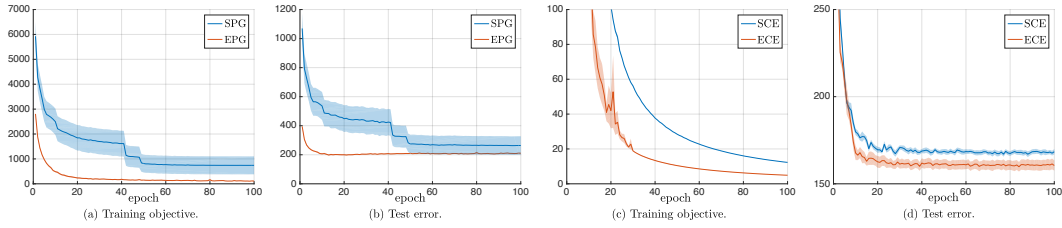

(a) Training objective.     (b) Test error.     (c) Training objective.     (d) Test error.

Figure 6: *Results on MNIST.*

Finally, for SL, we compare ECE and SCE on MNIST. For each training data $(x, y)$, the training objective is $-\log \pi_\theta(a_y|x)$, where $y(a_y) = 1$. The neural network and dataset settings are the same as above. As shown in Fig. 6(c) and (d), ECE with $p = 2$ is faster than SCE to achieve the same training objective, which benefits generalization, providing smaller test error than SCE.

## 7   Conclusion and Future Work

We discovered two phenomena that arise from the use of the standard softmax probability transformation in reinforcement learning and supervised learning, and proposed the escort transform to alleviate or eliminate these disadvantages. Our findings of the softmax gravity well and softmax damping phenomena challenge the common practice of using the softmax transformation in machine learning. However, there are other factors to consider when assessing such transformations in machine learning, such as the "temperature" of the softmax, or how the different transforms might impact generalization in the learned models. An important direction for future work is to investigate whether similar phenomena occur in other scenarios where the softmax is commonly utilized, such as attention models and exponential exploration.

Our underlying explanation using the concept of Non-uniform Łojasiewicz (NŁ) coefficient also provides an alternative theory to systematically study probability transforms, which goes beyond the classic convex "matching loss" theory [2, 9] and guarantees better optimization results. We expect further use of the NŁ coefficient to be beneficial in other problems in machine learning.

## Broader Impact

This research pursues a fundamental and mostly theoretical goal of understanding how a basic component in machine learning, the softmax transformation, impacts the convergence properties of subsequent optimization methods. The implications of this research are very high level and broad, since we investigate a widely used component. It is difficult to identify specific impacts, since this research does not target any specific application area that would directly impact people or society. If forced to make society level claims, we could attempt to claim that modifying architectures in ways that that improve optimization efficiency would have an effect on the overall energy footprint consumed by machine learning technologies, given how much computation is currently being expended on training softmax classifiers and policies.

## Acknowledgments and Disclosure of Funding

Jincheng Mei and Lihong Li would like to thank Christoph Dann for providing feedback on a draft of this manuscript. Csaba Szepesvári gratefully acknowledges funding from the Canada CIFAR AI Chairs Program, Amii and NSERC. Dale Schuurmans acknowledges funding from Amii and NSERC.

## Footnotes

*Work done when Lihong Li was with Google. Correspondence to: Jincheng Mei.

[2]Here, gradient ascent, as expected, refers to $\theta_{t+1} = \theta_t + \eta_t \cdot \dfrac{d\pi_{\theta_t}^\top r}{d\theta_t}$ and gradient flow refers to the continuous version when $\dfrac{d\theta_t}{dt} = \eta_t \cdot \dfrac{d\pi_{\theta_t}^\top r}{d\theta_t}$.

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
