[Supplementary Material]

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

# Appendix

The appendix is organized as follows.

## A  Policy Gradient Method

The policy gradient method we analyze in the main paper is the following Algorithm 1.

---
**Algorithm 1** Policy Gradient Method

---
    **Input:** Learning rate $\eta > 0$.
    Initialize parameter $\theta_1(s, a)$ for all $(s, a) \in \mathcal{S} \times \mathcal{A}$.
    **for** $t = 1$ **to** $T$ **do**
        $\theta_{t+1}(s, a) \leftarrow \theta_t(s, a) + \eta_t(s) \cdot \frac{\partial V^{\pi_{\theta_t}}(\mu)}{\partial \theta_t(s,a)}$ for all $(s, a)$.
    **end for**

---

Note that $\mu \in \Delta(\mathcal{S})$ is an initial state distribution which provides the initial states for policy gradient. Using different parameterizations, we will have different policy gradients $\frac{\partial V^{\pi_{\theta}}(\mu)}{\partial \theta}$ in Algorithm 1.

## B  Proofs for Section 2 (Softmax Gravity Well)

**Lemma 1** (Smoothness [14]). $\forall r \in [0, 1]^K$, $\theta \mapsto \pi_{\theta}^{\top} r$ is $5/2$-smooth, i.e., for all $\pi_{\theta} := \mathrm{softmax}(\theta)$ and $\pi_{\theta'} := \mathrm{softmax}(\theta')$, we have,

$$\left| (\pi_{\theta'} - \pi_{\theta})^{\top} r - \left\langle \frac{d\pi_{\theta}^{\top} r}{d\theta}, \theta' - \theta \right\rangle \right| \leq \frac{5}{4} \cdot \|\theta' - \theta\|_2^2.$$

*Proof.* See the proof in [14, Lemma 2]. □

**Theorem 1** (Escape time lower bound). Even in a single-state MDP, for any learning rate $\eta_t \in (0, 1]$, there exists an initialization of the policy $\pi_{\theta_1}$ and a positive constant $C$, such that SPG with full

gradients cannot escape a suboptimal corner before time $t_0 := \frac{C}{\Delta \cdot \pi_{\theta_1}(a^*)}$, i.e., it will hold that

$$(\pi^* - \pi_{\theta_t})^\top r \geq 0.9 \cdot \Delta, \tag{14}$$

for all $t \leq t_0$, where $\Delta := r(a^*) - \max_{a \neq a^*} r(a) > 0$ is the reward gap of $r \in [0, 1]^K$.

*Proof.* Consider the reward vector $r = (b + \Delta, b, \dots, b)^\top \in [0, 1]^K$ for some $b$, where $\Delta > 0$ is the reward gap. Then we have,

$$\pi_\theta^\top r = \pi_\theta(1) \cdot (b + \Delta) + (1 - \pi_\theta(1)) \cdot b.$$

Note that $a^* = 1$. We have,

$$\begin{aligned} r(a^*) - \pi_\theta^\top r &= b + \Delta - \pi_\theta^\top r \\ &= (1 - \pi_\theta(1)) \cdot \Delta. \end{aligned} \tag{15}$$

And $\forall a \neq 1$, we have,

$$\begin{aligned} r(a) - \pi_\theta^\top r &= b - \pi_\theta^\top r \\ &= -\pi_\theta(1) \cdot \Delta. \end{aligned} \tag{16}$$

Therefore, the $\ell_2$ norm of softmax policy gradient can be upper bounded as

$$\begin{aligned} \left\| \frac{d\pi_\theta^\top r}{d\theta} \right\|_2 &= \left[ \pi_\theta(a^*)^2 \cdot \left( r(a^*) - \pi_\theta^\top r \right)^2 + \sum_{a=2}^{K} \pi_\theta(a)^2 \cdot \left( r(a) - \pi_\theta^\top r \right)^2 \right]^{\frac{1}{2}} \\ &= \left[ \pi_\theta(1)^2 \cdot (1 - \pi_\theta(1))^2 \cdot \Delta^2 + \pi_\theta(1)^2 \cdot \Delta^2 \cdot \sum_{a=2}^{K} \pi_\theta(a)^2 \right]^{\frac{1}{2}} \quad \text{(by Eqs. (15) and (16))} \\ &= \pi_\theta(1) \cdot \Delta \cdot \left[ (1 - \pi_\theta(1))^2 + \sum_{a=2}^{K} \pi_\theta(a)^2 \right]^{\frac{1}{2}} \\ &\leq \pi_\theta(1) \cdot \Delta \cdot \left[ (1 - \pi_\theta(1))^2 + \left( \sum_{a=2}^{K} \pi_\theta(a) \right)^2 \right]^{\frac{1}{2}} \quad (\|x\|_2 \leq \|x\|_1) \\ &= \sqrt{2} \cdot \pi_\theta(1) \cdot (1 - \pi_\theta(1)) \cdot \Delta. \end{aligned} \tag{17}$$

Let $\theta_{t+1} \leftarrow \theta_t + \eta_t \cdot \frac{d\pi_{\theta_t}^\top r}{d\theta_t}$, and $\pi_{\theta_{t+1}} = \text{softmax}(\theta_{t+1})$ be the next policy after one step gradient update. Define the following two kinds of iterations:

$$\begin{aligned} t_{\text{good}} &:= \left\{ t \geq 1 : \pi_{\theta_{t+1}}(1) > \pi_{\theta_t}(1) \right\}, \\ t_{\text{bad}} &:= \left\{ t \geq 1 : \pi_{\theta_{t+1}}(1) \leq \pi_{\theta_t}(1) \right\}. \end{aligned}$$

For all $t \in t_{\text{bad}}$, we have,

$$\frac{1}{\pi_{\theta_t}(1)} - \frac{1}{\pi_{\theta_{t+1}}(1)} = \frac{1}{\pi_{\theta_{t+1}}(1) \cdot \pi_{\theta_t}(1)} \cdot \left( \pi_{\theta_{t+1}}(1) - \pi_{\theta_t}(1) \right) \leq 0. \tag{18}$$

For all $t \in t_{\text{good}}$, we have,

$$\pi_{\theta_{t+1}}(1) - \pi_{\theta_t}(1) = \left[1 - \frac{1}{\Delta} \cdot \left(r(a^*) - \pi_{\theta_{t+1}}^\top r\right)\right] - \left[1 - \frac{1}{\Delta} \cdot \left(r(a^*) - \pi_{\theta_t}^\top r\right)\right] \qquad \text{(by Eq. (15))}$$

$$= \frac{1}{\Delta} \cdot \left[(\pi_{\theta_{t+1}} - \pi_{\theta_t})^\top r - \left\langle \frac{d\pi_{\theta_t}^\top r}{d\theta_t}, \theta_{t+1} - \theta_t \right\rangle + \left\langle \frac{d\pi_{\theta_t}^\top r}{d\theta_t}, \theta_{t+1} - \theta_t \right\rangle\right]$$

$$\leq \frac{1}{\Delta} \cdot \left[\frac{5}{4} \cdot \|\theta_{t+1} - \theta_t\|_2^2 + \left\langle \frac{d\pi_{\theta_t}^\top r}{d\theta_t}, \theta_{t+1} - \theta_t \right\rangle\right] \qquad \text{(by Lemma 1)}$$

$$= \frac{1}{\Delta} \cdot \left(\frac{5\eta_t^2}{4} + \eta_t\right) \cdot \left\|\frac{d\pi_{\theta_t}^\top r}{d\theta_t}\right\|_2^2 \qquad \left(\theta_{t+1} = \theta_t + \eta_t \cdot \frac{d\pi_{\theta_t}^\top r}{d\theta_t}\right)$$

$$\leq \frac{1}{\Delta} \cdot \left(\frac{5\eta_t^2}{4} + \eta_t\right) \cdot 2 \cdot \pi_{\theta_t}(1)^2 \cdot (1 - \pi_{\theta_t}(1))^2 \cdot \Delta^2 \qquad \text{(by Eq. (17))}$$

$$\leq \frac{9}{2} \cdot \pi_{\theta_t}(1)^2 \cdot (1 - \pi_{\theta_t}(1))^2 \cdot \Delta \qquad (\eta_t \in (0, 1])$$

$$\leq \frac{9}{2} \cdot \pi_{\theta_t}(1)^2 \cdot \Delta. \qquad (\pi_{\theta_t}(1) \in [0, 1]) \tag{19}$$

Dividing both sides of Eq. (19) with $\pi_{\theta_{t+1}}(1) \cdot \pi_{\theta_t}(1)$, we have,

$$\frac{1}{\pi_{\theta_t}(1)} - \frac{1}{\pi_{\theta_{t+1}}(1)} \leq \frac{9}{2} \cdot \frac{\pi_{\theta_t}(1)}{\pi_{\theta_{t+1}}(1)} \cdot \Delta$$

$$\leq \frac{9}{2} \cdot \Delta. \qquad \left(\pi_{\theta_{t+1}}(1) \geq \pi_{\theta_t}(1) > 0\right) \tag{20}$$

Therefore, we have,

$$\frac{1}{\pi_{\theta_1}(1)} - \frac{1}{\pi_{\theta_t}(1)} = \sum_{s=1}^{t-1} \left[\frac{1}{\pi_{\theta_s}(1)} - \frac{1}{\pi_{\theta_{s+1}}(1)}\right]$$

$$= \sum_{s=1, \, s \in t_{\text{good}}}^{t-1} \left[\frac{1}{\pi_{\theta_s}(1)} - \frac{1}{\pi_{\theta_{s+1}}(1)}\right] + \sum_{s=1, \, s \in t_{\text{bad}}}^{t-1} \left[\frac{1}{\pi_{\theta_s}(1)} - \frac{1}{\pi_{\theta_{s+1}}(1)}\right]$$

$$\leq \sum_{s=1, \, s \in t_{\text{good}}}^{t-1} \left[\frac{1}{\pi_{\theta_s}(1)} - \frac{1}{\pi_{\theta_{s+1}}(1)}\right] \qquad \text{(by Eq. (18))}$$

$$\leq \sum_{s=1, \, s \in t_{\text{good}}}^{t-1} \left[\frac{9}{2} \cdot \Delta\right] \qquad \text{(by Eq. (20))}$$

$$\leq \frac{9}{2} \cdot \Delta \cdot t. \tag{21}$$

Let $\pi_{\theta_1}(1) \leq \frac{1}{c}$, for some constant $c \geq 11$. If $t \leq \frac{2}{9c} \cdot \frac{1}{\Delta} \cdot \frac{1}{\pi_{\theta_1}(1)}$, then we have,

$$\frac{1}{\pi_{\theta_t}(1)} \geq \frac{1}{\pi_{\theta_1}(1)} - \frac{9}{2} \cdot \Delta \cdot t \qquad \text{(by Eq. (21))}$$

$$\geq \frac{1}{\pi_{\theta_1}(1)} \cdot \left(1 - \frac{1}{c}\right)$$

$$\geq c \cdot \left(1 - \frac{1}{c}\right) = c - 1 \geq 10,$$

which implies $\pi_{\theta_t}(1) \leq \frac{1}{10}$. Therefore, for all $t \leq \frac{2}{9c} \cdot \frac{1}{\Delta} \cdot \frac{1}{\pi_{\theta_1}(1)}$, we have,

$$(\pi^* - \pi_{\theta_t})^\top r = (1 - \pi_{\theta_t}(1)) \cdot \Delta \qquad \text{(by Eq. (15))}$$

$$\geq 0.9 \cdot \Delta. \qquad \qquad \Box$$

# C Proofs for Section 3 (Escort Policy Gradient)

## C.1 Escort Policy Gradient Closed Form in Bandits

For completeness, we show the detailed calculations for the escort policy gradient in bandits, i.e., Eqs. (4) and (5), which are duplicated here for convenience,

$$
\frac{d\pi_\theta^\top r}{d\theta(a)} = p \cdot \mathrm{sgn}\{\theta(a)\} \cdot \frac{|\theta(a)|^{p-1}}{\sum_{a'} |\theta(a')|^p} \cdot \left[ r(a) - \pi_\theta^\top r \right]
$$

$$
= \frac{p}{\|\theta\|_p} \cdot \mathrm{sgn}\{\theta(a)\} \cdot \pi_\theta(a)^{1-1/p} \cdot \left[ r(a) - \pi_\theta^\top r \right].
$$

According to the chain rule, we have,

$$
\frac{d\pi_\theta^\top r}{d\theta} = \left( \frac{d\pi_\theta}{d\theta} \right)^\top \left( \frac{d\pi_\theta^\top r}{d\pi_\theta} \right) = \left( \frac{d\pi_\theta}{d\theta} \right)^\top r. \tag{22}
$$

We calculate the Jacobian of the escort transform $\pi_\theta = f_p(\theta)$. We have, for all $i, j \in [K]$,

$$
\begin{aligned}
\frac{d\pi_\theta(i)}{d\theta(j)} &= \frac{d}{d\theta(j)} \left\{ \frac{|\theta(i)|^p}{\sum_{a'} |\theta(a')|^p} \right\} \\
&= \frac{\delta_{ij} \cdot p \cdot |\theta(i)|^{p-1} \cdot \mathrm{sgn}\{\theta(i)\} \cdot \left( \sum_{a'} |\theta(a')|^p \right) - |\theta(i)|^p \cdot p \cdot |\theta(j)|^{p-1} \cdot \mathrm{sgn}\{\theta(j)\}}{\left( \sum_{a'} |\theta(a')|^p \right)^2} \\
&= \delta_{ij} \cdot p \cdot \mathrm{sgn}\{\theta(i)\} \cdot \frac{|\theta(i)|^{p-1}}{\sum_{a'} |\theta(a')|^p} - p \cdot \mathrm{sgn}\{\theta(j)\} \cdot \frac{|\theta(j)|^{p-1}}{\sum_{a'} |\theta(a')|^p} \cdot \pi_\theta(i),
\end{aligned}
$$

where

$$
\delta_{ij} = \begin{cases} 1, & \text{if } i = j, \\ 0, & \text{otherwise.} \end{cases} \tag{23}
$$

Then we have the Jacobian as,

$$
\left( \frac{d\pi_\theta}{d\theta} \right)^\top = p \cdot \frac{\mathrm{diag}(\mathrm{sgn}\{\theta\}) \mathrm{diag}(|\theta|^{p-1})}{\sum_{a'} |\theta(a')|^p} \left[ \mathbf{Id} - \mathbf{1}\pi_\theta^\top \right]. \tag{24}
$$

Combing Eqs. (22) and (24), we have,

$$
\frac{d\pi_\theta^\top r}{d\theta} = p \cdot \frac{\mathrm{diag}(\mathrm{sgn}\{\theta\}) \mathrm{diag}(|\theta|^{p-1})}{\sum_{a'} |\theta(a')|^p} \left[ r - \mathbf{1} \cdot \left( \pi_\theta^\top r \right) \right],
$$

which implies Eq. (4). Using $\pi_\theta(a) = \frac{|\theta(a)|^p}{\sum_{a'} |\theta(a')|^p}$, we have, if $\theta(a) \neq 0$,

$$
\begin{aligned}
\frac{d\pi_\theta^\top r}{d\theta(a)} &= p \cdot \mathrm{sgn}\{\theta(a)\} \cdot \frac{|\theta(a)|^{p-1}}{\sum_{a'} |\theta(a')|^p} \cdot \left[ r(a) - \pi_\theta^\top r \right] \\
&= \frac{p}{|\theta(a)|} \cdot \mathrm{sgn}\{\theta(a)\} \cdot \frac{|\theta(a)|^p}{\sum_{a'} |\theta(a')|^p} \cdot \left[ r(a) - \pi_\theta^\top r \right] \\
&= \frac{p}{\|\theta\|_p \cdot \pi_\theta(a)^{1/p}} \cdot \mathrm{sgn}\{\theta(a)\} \cdot \pi_\theta(a) \cdot \left[ r(a) - \pi_\theta^\top r \right] \\
&= \frac{p}{\|\theta\|_p} \cdot \mathrm{sgn}\{\theta(a)\} \cdot \pi_\theta(a)^{1-1/p} \cdot \left[ r(a) - \pi_\theta^\top r \right],
\end{aligned}
$$

which is Eq. (5). On the other hand, if $\theta(a) = 0$, then $\mathrm{sgn}\{\theta(a)\} = \mathrm{sgn}\{0\} = 0$ makes Eq. (4) trivially equal to Eq. (5).

## C.2 One-state MDPs

**Proposition 1.** $\theta \mapsto \pi_\theta^\top r$ is a non-concave function over $\mathbb{R}^K$ using the map $\pi_\theta := f_p(\theta)$.

*Proof.* Consider the following example with $K = 3$ and $p = 1$: $r = (1, 9/10, 1/10)^\top$, $\theta_1 = (2, 2, 2)^\top$, $\pi_{\theta_1} = f_1(\theta_1) = (1/3, 1/3, 1/3)^\top$, $\theta_2 = (5, 10, 15)^\top$, and $\pi_{\theta_2} = f_1(\theta_2) = (1/6, 1/3, 1/2)^\top$. We have,

$$\frac{1}{2} \cdot \left(\pi_{\theta_1}^\top r + \pi_{\theta_2}^\top r\right) = \frac{1}{2} \cdot \left(\frac{2}{3} + \frac{31}{60}\right) = \frac{71}{120} = \frac{213}{360}.$$

Denote $\bar{\theta} := \frac{1}{2} \cdot (\theta_1 + \theta_2) = (7/2, 12/2, 17/2)^\top$. We have $\pi_{\bar{\theta}} = f_1(\bar{\theta}) = (7/36, 12/36, 17/36)^\top$,

$$\pi_{\bar{\theta}}^\top r = \frac{195}{360}.$$

Since $\frac{1}{2} \cdot \left(\pi_{\theta_1}^\top r + \pi_{\theta_2}^\top r\right) > \pi_{\bar{\theta}}^\top r$, we see that $\mathbb{E}_{a \sim \pi_\theta(\cdot)}[r(a)]$ is a non-concave function of $\theta$. $\square$

**Lemma 2** (Non-uniform smoothness). *Suppose $r \in [0, 1]^K$. Let $\pi_\theta := f_p(\theta)$, and $\pi_{\theta'} := f_p(\theta')$. Denote $\theta_\xi := \theta + \xi \cdot (\theta' - \theta)$ with some $\xi \in [0, 1]$. Then, we have,*

- *for $p \geq 2$, $\pi_\theta^\top r$ is $\frac{3 \cdot p^2 \cdot K^{1/p}}{\|\theta_\xi\|_p^2}$-smooth, i.e.,*

$$\left|(\pi_{\theta'} - \pi_\theta)^\top r - \left\langle \frac{d\pi_\theta^\top r}{d\theta}, \theta' - \theta \right\rangle\right| \leq \frac{3 \cdot p^2 \cdot K^{1/p}}{2 \cdot \|\theta_\xi\|_p^2} \cdot \|\theta' - \theta\|_2^2.$$

- *for $p = 1$, $\pi_\theta^\top r$ is $\frac{2 \cdot K}{\|\theta_\xi\|_1^2}$-smooth, i.e.,*

$$\left|(\pi_{\theta'} - \pi_\theta)^\top r - \left\langle \frac{d\pi_\theta^\top r}{d\theta}, \theta' - \theta \right\rangle\right| \leq \frac{K}{\|\theta_\xi\|_1^2} \cdot \|\theta' - \theta\|_2^2.$$

*Proof.* Denote the second derivative w.r.t. $\theta$ (i.e., Hessian) as

$$S(r, \theta) = \frac{d}{d\theta}\left\{\frac{d\pi_\theta^\top r}{d\theta}\right\}$$

$$= p \cdot \frac{d}{d\theta}\left\{\operatorname{diag}\left(\frac{\pi_\theta}{\theta}\right)(r - \pi_\theta^\top r \cdot \mathbf{1})\right\}.$$

Note that $S(r, \theta) \in \mathbb{R}^{K \times K}$, whose element at position $(i, j) \in [K]^2$ is

$$S_{i,j} = p \cdot \frac{d\{\frac{\pi_\theta(i)}{\theta(i)} \cdot (r(i) - \pi_\theta^\top r)\}}{d\theta(j)}$$

$$= p \cdot \frac{d\{\frac{\pi_\theta(i)}{\theta(i)}\}}{d\theta(j)} \cdot (r(i) - \pi_\theta^\top r) + p \cdot \frac{\pi_\theta(i)}{\theta(i)} \cdot \frac{d\{r(i) - \pi_\theta^\top r\}}{d\theta(j)}$$

$$= p \cdot \frac{\frac{p}{\theta(j)} \cdot [\delta_{ij} \cdot \pi_\theta(i) - \pi_\theta(i) \cdot \pi_\theta(j)] \cdot \theta(i) - \pi_\theta(i) \cdot \delta_{ij}}{\theta(i)^2} \cdot (r(i) - \pi_\theta^\top r)$$

$$- \frac{\pi_\theta(i)}{\theta(i)} \cdot p^2 \cdot \frac{\pi_\theta(j)}{\theta(j)} \cdot (r(j) - \pi_\theta^\top r)$$

$$= p \cdot (p - 1) \cdot \delta_{ij} \cdot \frac{\pi_\theta(i)}{\theta(i)^2} \cdot (r(i) - \pi_\theta^\top r)$$

$$- p^2 \cdot \frac{\pi_\theta(i)}{\theta(i)} \cdot \frac{\pi_\theta(j)}{\theta(j)} \cdot (r(i) - \pi_\theta^\top r) - p^2 \cdot \frac{\pi_\theta(i)}{\theta(i)} \cdot \frac{\pi_\theta(j)}{\theta(j)} \cdot (r(j) - \pi_\theta^\top r),$$

where $\delta_{ij}$ is defined in Eq. (23). We calculate the spectral radius of $S(r, \theta)$. For any nonzero $y \in \mathbb{R}^K$,

$$
\left| y^\top S(r, \theta) y \right| = \left| \sum_{i=1}^K \sum_{j=1}^K S_{i,j} y(i) y(j) \right|
$$

$$
= \left| p \cdot (p-1) \sum_i \frac{\pi_\theta(i)}{\theta(i)^2} \cdot (r(i) - \pi_\theta^\top r) \cdot y(i)^2 \right.
$$

$$
\left. -2 \cdot p^2 \sum_i \frac{\pi_\theta(i)}{\theta(i)} \cdot y(i) \sum_j \frac{\pi_\theta(j)}{\theta(j)} \cdot (r(j) - \pi_\theta^\top r) \cdot y(j) \right|
$$

$$
\leq p \cdot (p-1) \cdot \left| \sum_i \frac{\pi_\theta(i)}{\theta(i)^2} \cdot (r(i) - \pi_\theta^\top r) \cdot y(i)^2 \right|
$$

$$
+ 2 \cdot p^2 \cdot \left| \sum_i \frac{\pi_\theta(i)}{\theta(i)} \cdot y(i) \sum_j \frac{\pi_\theta(j)}{\theta(j)} \cdot (r(j) - \pi_\theta^\top r) \cdot y(j) \right|, \tag{25}
$$

where the last inequality is by triangle inequality.

**First part.** For $p \geq 2$, the first term in Eq. (25) is upper bounded as,

$$
\left| \sum_i \frac{\pi_\theta(i)}{\theta(i)^2} \cdot (r(i) - \pi_\theta^\top r) \cdot y(i)^2 \right| \leq \sum_i \frac{\pi_\theta(i)}{\theta(i)^2} \cdot \left| r(i) - \pi_\theta^\top r \right| \cdot y(i)^2 \qquad \text{(by triangle inequality)}
$$

$$
\leq \left( \max_a \frac{\pi_\theta(a)}{\theta(a)^2} \cdot |r(a) - \pi_\theta^\top r| \right) \cdot \|y\|_2^2 \qquad \text{(by Hölder's inequality)}
$$

$$
\leq \left( \max_a \frac{\pi_\theta(a)}{\theta(a)^2} \right) \cdot \|y\|_2^2 \qquad (r(a) \in [0, 1], \ \forall a)
$$

$$
= \frac{1}{\|\theta\|_p^2} \cdot \left( \max_a \pi_\theta(a)^{1-2/p} \right) \cdot \|y\|_2^2
$$

$$
\leq \frac{1}{\|\theta\|_p^2} \cdot \|y\|_2^2. \qquad (p \geq 2) \tag{26}
$$

The last term in Eq. (25) is upper bounded as,

$$
\left| \sum_i \frac{\pi_\theta(i)}{\theta(i)} \cdot y(i) \sum_j \frac{\pi_\theta(j)}{\theta(j)} \cdot (r(j) - \pi_\theta^\top r) \cdot y(j) \right|
$$

$$
\leq \left\| \frac{\pi_\theta}{\theta} \right\|_2 \cdot \|y\|_2 \cdot \left\| \mathrm{diag}\left( \frac{\pi_\theta}{\theta} \right) (r - \pi_\theta^\top r \cdot \mathbf{1}) \right\|_2 \cdot \|y\|_2 \qquad \text{(by Cauchy-Schwarz)}
$$

$$
= \left\| \frac{\pi_\theta}{\theta} \right\|_2 \cdot \left[ \sum_a \left( \frac{\pi_\theta(a)}{\theta(a)} \right)^2 \cdot (r(a) - \pi_\theta^\top r)^2 \right]^{\frac{1}{2}} \cdot \|y\|_2^2
$$

$$
\leq \left\| \frac{\pi_\theta}{\theta} \right\|_2 \cdot \left[ \sum_a \left( \frac{\pi_\theta(a)}{\theta(a)} \right)^2 \right]^{\frac{1}{2}} \cdot \|y\|_2^2 \qquad (r(a) \in [0, 1], \ \forall a)
$$

$$
= \sum_a \left( \frac{\pi_\theta(a)}{\theta(a)} \right)^2 \cdot \|y\|_2^2
$$

$$
= \frac{1}{\|\theta\|_p^2} \sum_a \left( \pi_\theta(a)^{1-1/p} \right)^2 \cdot \|y\|_2^2
$$

$$
\leq \frac{1}{\|\theta\|_p^2} \cdot \left( \sum_a \pi_\theta(a)^{1-1/p} \right) \cdot \|y\|_2^2. \qquad \left( \pi_\theta(a)^{1-1/p} \in [0, 1] \right) \tag{27}
$$

The intermediate term is then upper bounded as,

$$\sum_a \pi_\theta(a)^{1-1/p} = K^{1/p} \cdot \frac{1}{K} \sum_a (K \cdot \pi_\theta(a))^{1-1/p}$$

$$\leq K^{1/p} \cdot \left( \sum_a \frac{K \cdot \pi_\theta(a)}{K} \right)^{1-1/p} \qquad \text{(by Jensen's inequality)}$$

$$= K^{1/p}. \tag{28}$$

Combining Eqs. (25) to (28), we have

$$\left| y^\top S(r, \theta) y \right| \leq p \cdot (p-1) \cdot \frac{1}{\|\theta\|_p^2} \cdot \|y\|_2^2 + 2 \cdot p^2 \cdot \frac{1}{\|\theta\|_p^2} \cdot K^{1/p} \cdot \|y\|_2^2$$

$$\leq \frac{3 \cdot p^2 \cdot K^{1/p}}{\|\theta\|_p^2} \cdot \|y\|_2^2. \qquad \left( K^{1/p} \geq 1 \right) \tag{29}$$

According to Taylor's theorem, we have, for $p \geq 2$,

$$\left| (\pi_{\theta'} - \pi_\theta)^\top r - \left\langle \frac{d\pi_\theta^\top r}{d\theta}, \theta' - \theta \right\rangle \right| = \frac{1}{2} \cdot \left| (\theta' - \theta)^\top S(r, \theta_\xi) (\theta' - \theta) \right|$$

$$\leq \frac{3 \cdot p^2 \cdot K^{1/p}}{2 \cdot \|\theta_\xi\|_p^2} \cdot \|\theta' - \theta\|_2^2. \qquad \text{(by Eq. (29))}$$

**Second part.** For $p = 1$, according to Eq. (25), Eqs. (27) and (28), we have,

$$\left| y^\top S(r, \theta) y \right| \leq 2 \cdot p^2 \cdot \frac{1}{\|\theta\|_p^2} \cdot K^{1/p} \cdot \|y\|_2^2 = \frac{2 \cdot K}{\|\theta\|_1^2} \cdot \|y\|_2^2. \tag{30}$$

According to Taylor's theorem, we have, for $p = 1$,

$$\left| (\pi_{\theta'} - \pi_\theta)^\top r - \left\langle \frac{d\pi_\theta^\top r}{d\theta}, \theta' - \theta \right\rangle \right| \leq \frac{K}{\|\theta_\xi\|_1^2} \cdot \|\theta' - \theta\|_2^2. \qquad \square$$

**Lemma 3** (Non-uniform Łojasiewicz). *Let $\pi_\theta = f_p(\theta)$. For any $p > 0$, we have,*

$$\left\| \frac{d\pi_\theta^\top r}{d\theta} \right\|_2 \geq \frac{p}{\|\theta\|_p} \cdot \pi_\theta(a^*)^{1-1/p} \cdot (\pi^* - \pi_\theta)^\top r,$$

*where $\pi^* = \arg\max_{\pi \in \Delta} \pi^\top r$ is the optimal policy.*

*Proof.* The result follows from calculating the gradient norm,

$$\left\| \frac{d\pi_\theta^\top r}{d\theta} \right\|_2 = \left[ \sum_{a=1}^K \left( p \cdot \frac{\pi_\theta(a)}{\theta(a)} \cdot (r(a) - \pi_\theta^\top r) \right)^2 \right]^{\frac{1}{2}}$$

$$\geq \left| p \cdot \frac{\pi_\theta(a^*)}{\theta(a^*)} \cdot (r(a^*) - \pi_\theta^\top r) \right|$$

$$= p \cdot \frac{\pi_\theta(a^*)}{|\theta(a^*)|} \cdot (\pi^* - \pi_\theta)^\top r$$

$$= \frac{p}{\|\theta\|_p} \cdot \pi_\theta(a^*)^{1-1/p} \cdot (\pi^* - \pi_\theta)^\top r. \qquad \left( \pi_\theta(a) = \frac{|\theta(a)|^p}{\sum_{a'} |\theta(a')|^p} \right) \qquad \square$$

**Theorem 2.** For a single-state MDP, following the escort policy gradient with any initialization such that $|\theta_1(a)| > 0, \forall a$, we obtain the following upper bounds on the sub-optimality for all $t \geq 1$,

- (gradient ascent) for $p \geq 2$, with $\eta_t = \frac{2 \cdot \|\theta_t\|_p^2}{9 \cdot p^2 \cdot K^{1/p}}$, we have,

$$(\pi^* - \pi_{\theta_t})^\top r \leq \frac{9 \cdot K^{1/p}}{c^{2-2/p}} \cdot \frac{1}{t};$$

- (gradient ascent) for $p = 1$, with $\eta_t = \frac{2 \cdot \|\theta_t\|_1^2}{9 \cdot K}$, we have,

$$(\pi^* - \pi_{\theta_t})^\top r \leq \frac{9 \cdot K}{t};$$

- (gradient flow) for $p \geq 1$, with $\eta_t = \frac{\|\theta_t\|_p^2}{p^2}$, we have,

$$(\pi^* - \pi_{\theta_t})^\top r \leq \frac{1}{c^{2-2/p} \cdot (t-1) + 1},$$

where $c := \inf_{t \geq 1} \pi_{\theta_t}(a^*) > 0$ depends on the problem and initialization, but is time-independent.

*Proof.* **First part.** For $p \geq 2$, according to Lemma 2,

$$\left| (\pi_{\theta_{t+1}} - \pi_{\theta_t})^\top r - \left\langle \frac{d\pi_{\theta_t}^\top r}{d\theta_t}, \theta_{t+1} - \theta_t \right\rangle \right| \leq \frac{3 \cdot p^2 \cdot K^{1/p}}{2 \cdot \|\theta_{t,\xi}\|_p^2} \cdot \|\theta_{t+1} - \theta_t\|_2^2, \tag{31}$$

where

$$\theta_{t,\xi} := \theta_t + \xi \cdot (\theta_{t+1} - \theta_t) = \theta_t + \xi \cdot \eta_t \cdot \frac{d\pi_{\theta_t}^\top r}{d\theta_t},$$

for some $\xi \in [0, 1]$. The $\ell_p$ gradient norm can be upper bounded as,

$$\left\| \frac{d\pi_\theta^\top r}{d\theta} \right\|_p = \left[ \sum_{a=1}^{K} \left| p \cdot \frac{\pi_\theta(a)}{\theta(a)} \cdot (r(a) - \pi_\theta^\top r) \right|^p \right]^{\frac{1}{p}}$$

$$= p \cdot \left[ \sum_{a=1}^{K} \left( \frac{\pi_\theta(a)}{|\theta(a)|} \cdot |r(a) - \pi_\theta^\top r| \right)^p \right]^{\frac{1}{p}}$$

$$= \frac{p}{\|\theta\|_p} \cdot \left[ \sum_{a=1}^{K} \left( \pi_\theta(a)^{1-1/p} \cdot |r(a) - \pi_\theta^\top r| \right)^p \right]^{\frac{1}{p}}$$

$$\leq \frac{p}{\|\theta\|_p} \cdot \left[ \sum_{a=1}^{K} (1 \cdot 1)^p \right]^{\frac{1}{p}} = \frac{p \cdot K^{1/p}}{\|\theta\|_p}. \tag{32}$$

According to the triangle inequality, we have,

$$\|\theta_{t,\xi}\|_p \geq \|\theta_t\|_p - \xi \cdot \eta_t \cdot \left\| \frac{d\pi_{\theta_t}^\top r}{d\theta_t} \right\|_p$$

$$\geq \|\theta_t\|_p - \xi \cdot \eta_t \cdot \frac{p \cdot K^{1/p}}{\|\theta_t\|_p}. \qquad \text{(by Eq. (32))}$$

$$= \|\theta_t\|_p \cdot \left( 1 - \xi \cdot \frac{2}{9 \cdot p} \right) \qquad \left( \eta_t = \frac{2 \cdot \|\theta_t\|_p^2}{9 \cdot p^2 \cdot K^{1/p}} \right)$$

$$\geq \|\theta_t\|_p \cdot \left( 1 - \frac{2}{9 \cdot p} \right) \qquad (\xi \in [0, 1])$$

$$= \|\theta_t\|_p \cdot \left[ \left( 1 - \frac{2}{\sqrt{6}} \right) \cdot \left( 1 - \frac{2 \cdot (3 + \sqrt{6})}{9 \cdot p} \right) + \frac{2}{\sqrt{6}} \right]$$

$$\geq \frac{2}{\sqrt{6}} \cdot \|\theta_t\|_p. \qquad (p \geq 2) \tag{33}$$

Combining Eqs. (31) and (33), we have,

$$\left| (\pi_{\theta_{t+1}} - \pi_{\theta_t})^\top r - \left\langle \frac{d\pi_{\theta_t}^\top r}{d\theta_t}, \theta_{t+1} - \theta_t \right\rangle \right| \leq \frac{3 \cdot p^2 \cdot K^{1/p}}{2 \cdot \|\theta_{t,\xi}\|_p^2} \cdot \|\theta_{t+1} - \theta_t\|_2^2$$

$$\leq \frac{9 \cdot p^2 \cdot K^{1/p}}{4 \cdot \|\theta_t\|_p^2} \cdot \|\theta_{t+1} - \theta_t\|_2^2,$$

which implies,

$$\pi_{\theta_t}^\top r - \pi_{\theta_{t+1}}^\top r \leq -\left\langle \frac{d\pi_{\theta_t}^\top r}{d\theta_t}, \theta_{t+1} - \theta_t \right\rangle + \frac{9 \cdot p^2 \cdot K^{1/p}}{4 \cdot \|\theta_t\|_p^2} \cdot \|\theta_{t+1} - \theta_t\|_2^2$$

$$= -\eta_t \cdot \left\| \frac{d\pi_{\theta_t}^\top r}{d\theta_t} \right\|_2^2 + \frac{9 \cdot p^2 \cdot K^{1/p}}{4 \cdot \|\theta_t\|_p^2} \cdot \eta_t^2 \cdot \left\| \frac{d\pi_{\theta_t}^\top r}{d\theta_t} \right\|_2^2 \qquad \left( \theta_{t+1} = \theta_t + \eta_t \cdot \frac{d\pi_{\theta_t}^\top r}{d\theta_t} \right)$$

$$= -\frac{\|\theta_t\|_p^2}{9 \cdot p^2 \cdot K^{1/p}} \cdot \left\| \frac{d\pi_{\theta_t}^\top r}{d\theta_t} \right\|_2^2 \qquad \left( \eta_t = \frac{2 \cdot \|\theta_t\|_p^2}{9 \cdot p^2 \cdot K^{1/p}} \right)$$

$$\leq -\frac{\|\theta_t\|_p^2}{9 \cdot p^2 \cdot K^{1/p}} \cdot \left[ \frac{p}{\|\theta_t\|_p} \cdot \pi_{\theta_t}(a^*)^{1-1/p} \cdot (\pi^* - \pi_{\theta_t})^\top r \right]^2 \qquad \text{(by Lemma 3)}$$

$$= -\frac{\pi_{\theta_t}(a^*)^{2-2/p}}{9 \cdot K^{1/p}} \cdot \left[ (\pi^* - \pi_{\theta_t})^\top r \right]^2$$

$$\leq -\frac{c_t^{2-2/p}}{9 \cdot K^{1/p}} \cdot \left[ (\pi^* - \pi_{\theta_t})^\top r \right]^2, \tag{34}$$

where $c_t := \min_{1 \leq s \leq t} \pi_{\theta_s}(a^*) > 0$. Eq. (34) is equivalent to,

$$(\pi^* - \pi_{\theta_{t+1}})^\top r - (\pi^* - \pi_{\theta_t})^\top r \leq -\frac{c_t^{2-2/p}}{9 \cdot K^{1/p}} \cdot \left[ (\pi^* - \pi_{\theta_t})^\top r \right]^2. \tag{35}$$

Denote $\delta_t := (\pi^* - \pi_{\theta_t})^\top r$. We prove $\delta_t \leq \frac{9 \cdot K^{1/p}}{c_t^{2-2/p}} \cdot \frac{1}{t}$ by induction. For $t = 2$, since $c_2 \in (0,1)$,

$$\delta_2 \leq 1 \leq \frac{9 \cdot K^{1/p}}{c_2^{2-2/p}} \cdot \frac{1}{2}.$$

Suppose $\delta_t \leq \frac{9 \cdot K^{1/p}}{c_t^{2-2/p}} \cdot \frac{1}{t}$, $t \geq 2$. Consider $f_t : \mathbb{R} \to \mathbb{R}$, $f_t(x) := x - \frac{c_t^{2-2/p}}{9 \cdot K^{1/p}} \cdot x^2$. Clearly, $f_t$ is monotonically increasing in $\left[ 0, \frac{9 \cdot K^{1/p}}{2 \cdot c_t^{2-2/p}} \right]$. We have,

$$\delta_{t+1} \leq \delta_t - \frac{c_t^{2-2/p}}{9 \cdot K^{1/p}} \cdot \delta_t^2 \qquad \text{(by Eq. (35))}$$

$$\leq \frac{9 \cdot K^{1/p}}{c_t^{2-2/p}} \cdot \frac{1}{t} - \frac{c_t^{2-2/p}}{9 \cdot K^{1/p}} \cdot \left( \frac{9 \cdot K^{1/p}}{c_t^{2-2/p}} \cdot \frac{1}{t} \right)^2 \qquad \left( \delta_t \leq \frac{9 \cdot K^{1/p}}{c_t^{2-2/p}} \cdot \frac{1}{t} \leq \frac{9 \cdot K^{1/p}}{2 \cdot c_t^{2-2/p}}, \ t \geq 2 \right)$$

$$= \frac{9 \cdot K^{1/p}}{c_t^{2-2/p}} \cdot \left( \frac{1}{t} - \frac{1}{t^2} \right)$$

$$\leq \frac{9 \cdot K^{1/p}}{c_t^{2-2/p}} \cdot \frac{1}{t+1}, \tag{36}$$

which completes the proof for $\delta_t \leq \frac{9 \cdot K^{1/p}}{c_t^{2-2/p}} \cdot \frac{1}{t}$. Then we have, for all $t \geq 1$,

$$(\pi^* - \pi_{\theta_t})^\top r \leq \frac{9 \cdot K^{1/p}}{c_t^{2-2/p}} \cdot \frac{1}{t} \leq \frac{9 \cdot K^{1/p}}{(\inf_{t \geq 1} \pi_{\theta_t}(a^*))^{2-2/p}} \cdot \frac{1}{t}.$$

**Second part.** For $p = 1$, according to Lemma 2,

$$\left| (\pi_{\theta_{t+1}} - \pi_{\theta_t})^\top r - \left\langle \frac{d\pi_{\theta_t}^\top r}{d\theta_t}, \theta_{t+1} - \theta_t \right\rangle \right| \leq \frac{K}{\|\theta_{t,\xi}\|_1^2} \cdot \|\theta_{t+1} - \theta_t\|_2^2, \tag{37}$$

where

$$\theta_{t,\xi} := \theta_t + \xi \cdot (\theta_{t+1} - \theta_t) = \theta_t + \xi \cdot \eta_t \cdot \frac{d\pi_{\theta_t}^\top r}{d\theta_t},$$

for some $\xi \in [0, 1]$. The $\ell_1$ norm can be upper bounded as

$$
\begin{aligned}
\left\| \frac{d\pi_{\theta_t}^\top r}{d\theta_t} \right\|_1 &= \sum_{a=1}^K \left| \frac{\pi_{\theta_t}(a)}{\theta_t(a)} \cdot \left( r(a) - \pi_\theta^\top r \right) \right| \\
&= \frac{1}{\|\theta_t\|_1} \sum_{a=1}^K |r(a) - \pi_\theta^\top r| \\
&\leq \frac{K}{\|\theta_t\|_1}. \qquad \left( r \in [0,1]^K \right)
\end{aligned}
\tag{38}
$$

According to triangle inequality, we have,

$$
\begin{aligned}
\|\theta_{t,\xi}\|_1 &\geq \|\theta_t\|_1 - \xi \cdot \eta_t \cdot \left\| \frac{d\pi_{\theta_t}^\top r}{d\theta_t} \right\|_1 \\
&\geq \|\theta_t\|_1 - \xi \cdot \eta_t \cdot \frac{K}{\|\theta_t\|_1}. \qquad \text{(by Eq. (38))} \\
&= \|\theta_t\|_1 \cdot \left( 1 - \xi \cdot \frac{2}{9} \right) \qquad \left( \eta_t = \frac{2 \cdot \|\theta_t\|_1^2}{9 \cdot K} \right) \\
&\geq \frac{2}{3} \cdot \|\theta_t\|_1. \qquad (\xi \in [0,1])
\end{aligned}
\tag{39}
$$

Combining Eqs. (37) and (39), we have,

$$
\begin{aligned}
\left| (\pi_{\theta_{t+1}} - \pi_{\theta_t})^\top r - \left\langle \frac{d\pi_{\theta_t}^\top r}{d\theta_t}, \theta_{t+1} - \theta_t \right\rangle \right| &\leq \frac{K}{\|\theta_{t,\xi}\|_1^2} \cdot \|\theta_{t+1} - \theta_t\|_1^2 \\
&\leq \frac{9 \cdot K}{4 \cdot \|\theta_t\|_1^2} \cdot \|\theta_{t+1} - \theta_t\|_1^2,
\end{aligned}
$$

which implies,

$$
\begin{aligned}
\pi_{\theta_t}^\top r - \pi_{\theta_{t+1}}^\top r &\leq -\left\langle \frac{d\pi_{\theta_t}^\top r}{d\theta_t}, \theta_{t+1} - \theta_t \right\rangle + \frac{9 \cdot K}{4 \cdot \|\theta_t\|_1^2} \cdot \|\theta_{t+1} - \theta_t\|_1^2 \\
&= -\eta_t \cdot \left\| \frac{d\pi_{\theta_t}^\top r}{d\theta_t} \right\|_2^2 + \frac{9 \cdot K}{4 \cdot \|\theta_t\|_1^2} \cdot \eta_t^2 \cdot \left\| \frac{d\pi_{\theta_t}^\top r}{d\theta_t} \right\|_2^2 \qquad \left( \theta_{t+1} = \theta_t + \eta_t \cdot \frac{d\pi_{\theta_t}^\top r}{d\theta_t} \right) \\
&= -\frac{\|\theta_t\|_1^2}{9 \cdot K} \cdot \left\| \frac{d\pi_{\theta_t}^\top r}{d\theta_t} \right\|_2^2 \qquad \left( \eta_t = \frac{2 \cdot \|\theta_t\|_1^2}{9 \cdot K} \right) \\
&\leq -\frac{\|\theta_t\|_1^2}{9 \cdot K} \cdot \left[ \frac{1}{\|\theta_t\|_1} \cdot (\pi^* - \pi_{\theta_t})^\top r \right]^2 \qquad \text{(by Lemma 3)} \\
&= -\frac{1}{9 \cdot K} \cdot \left[ (\pi^* - \pi_{\theta_t})^\top r \right]^2.
\end{aligned}
\tag{40}
$$

Using a similar induction argument as in Eq. (36), we have

$$
(\pi^* - \pi_{\theta_t})^\top r \leq \frac{9 \cdot K}{t}.
$$

**Third part.** For the gradient flow, we have,

$$
\frac{d\{(\pi^* - \pi_{\theta_t})^\top r\}}{dt} = -\frac{d\pi_{\theta_t}^\top r}{dt}
$$

$$
= -\left(\frac{d\theta_t}{dt}\right)^\top \left(\frac{d\pi_{\theta_t}^\top r}{d\theta_t}\right)
$$

$$
= -\eta_t \cdot \left\|\frac{d\pi_{\theta_t}^\top r}{d\theta_t}\right\|_2^2 \qquad \left(\frac{d\theta_t}{dt} = \eta_t \cdot \frac{d\pi_{\theta_t}^\top r}{d\theta_t}\right)
$$

$$
\leq -\eta_t \cdot \left[\frac{p}{\|\theta_t\|_p} \cdot \pi_{\theta_t}(a^*)^{1-1/p} \cdot (\pi^* - \pi_{\theta_t})^\top r\right]^2 \qquad \text{(by Lemma 3)}
$$

$$
= -\pi_{\theta_t}(a^*)^{2-2/p} \cdot \left[(\pi^* - \pi_{\theta_t})^\top r\right]^2 \qquad \left(\eta_t = \frac{\|\theta_t\|_p^2}{p^2}\right)
$$

$$
\leq -c^{2-2p} \cdot [(\pi^* - \pi_{\theta_t})^\top r]^2,
$$

which implies,

$$
\frac{d}{dt}\left\{\frac{1}{(\pi^* - \pi_{\theta_t})^\top r}\right\} = -\frac{1}{[(\pi^* - \pi_{\theta_t})^\top r]^2} \cdot \frac{d\{(\pi^* - \pi_{\theta_t})^\top r\}}{dt} = c^{2-2p}.
$$

Taking integral, we have,

$$
\frac{1}{(\pi^* - \pi_{\theta_t})^\top r} = \frac{1}{(\pi^* - \pi_{\theta_1})^\top r} + c^{2-2p} \cdot (t-1)
$$

$$
\geq 1 + c^{2-2p} \cdot (t-1), \qquad \left((\pi^* - \pi_{\theta_1})^\top r \in (0,1]\right)
$$

which is equivalent to

$$
(\pi^* - \pi_{\theta_t})^\top r \leq \frac{1}{c^{2-2p} \cdot (t-1) + 1}. \qquad \square
$$

## C.3 General MDPs

**Lemma 4** (Policy gradient theorem [19]). *Fix a map $\theta \mapsto \pi_\theta(a|s)$ that for any $(s,a)$ is differentiable and fix an initial distribution $\mu \in \Delta(\mathcal{S})$. Then,*

$$
\frac{\partial V^{\pi_\theta}(\mu)}{\partial \theta} = \frac{1}{1-\gamma} \mathop{\mathbb{E}}_{s \sim d_\mu^{\pi_\theta}} \left[\sum_a \frac{\partial \pi_\theta(a|s)}{\partial \theta} \cdot Q^{\pi_\theta}(s,a)\right].
$$

**Lemma 5.** *The escort policy gradient w.r.t. $\theta$ is*

$$
\frac{\partial V^{\pi_\theta}(\mu)}{\partial \theta(s,a)} = \frac{1}{1-\gamma} \cdot d_\mu^{\pi_\theta}(s) \cdot p \cdot \frac{\pi_\theta(a|s)}{\theta(s,a)} \cdot A^{\pi_\theta}(s,a),
$$

*where $A^{\pi_\theta}(s,a)$ is the advantage function defined as*

$$
A^{\pi_\theta}(s,a) = Q^{\pi_\theta}(s,a) - V^{\pi_\theta}(s),
$$

$$
Q^{\pi_\theta}(s,a) = r(s,a) + \gamma \sum_{s'} \mathcal{P}(s'|s,a) V^{\pi_\theta}(s').
$$

*Proof.* According to Lemma 4, we have,

$$
\frac{\partial V^{\pi_\theta}(\mu)}{\partial \theta} = \frac{1}{1-\gamma} \mathop{\mathbb{E}}_{s' \sim d_\mu^{\pi_\theta}} \left[\sum_a \frac{\partial \pi_\theta(a|s')}{\partial \theta} \cdot Q^{\pi_\theta}(s',a)\right].
$$

For $s' \neq s$, $\frac{\partial \pi_\theta(a|s')}{\partial \theta(s,\cdot)} = 0$ since $\pi_\theta(a|s')$ does not depend on $\theta(s,\cdot)$. Therefore,

$$
\frac{\partial V^{\pi_\theta}(\mu)}{\partial \theta(s,\cdot)} = \frac{1}{1-\gamma} \cdot d_\mu^{\pi_\theta}(s) \cdot \left[ \sum_a \frac{\partial \pi_\theta(a|s)}{\partial \theta(s,\cdot)} \cdot Q^{\pi_\theta}(s,a) \right]
$$

$$
= \frac{1}{1-\gamma} \cdot d_\mu^{\pi_\theta}(s) \cdot \left( \frac{d\pi(\cdot|s)}{d\theta(s,\cdot)} \right)^\top Q^{\pi_\theta}(s,\cdot)
$$

$$
= \frac{1}{1-\gamma} \cdot d_\mu^{\pi_\theta}(s) \cdot p \cdot \mathrm{diag}\left( \frac{\pi(\cdot|s)}{\theta(s,\cdot)} \right) \left( \mathbf{Id} - \mathbf{1}\pi_\theta^\top \right) Q^{\pi_\theta}(s,\cdot).
$$

For each component $a$, we have

$$
\frac{\partial V^{\pi_\theta}(\mu)}{\partial \theta(s,a)} = \frac{1}{1-\gamma} \cdot d_\mu^{\pi_\theta}(s) \cdot p \cdot \frac{\pi_\theta(a|s)}{\theta(s,a)} \cdot \left[ Q^{\pi_\theta}(s,a) - \sum_a \pi_\theta(a|s) \cdot Q^{\pi_\theta}(s,a) \right]
$$

$$
= \frac{1}{1-\gamma} \cdot d_\mu^{\pi_\theta}(s) \cdot p \cdot \frac{\pi_\theta(a|s)}{\theta(s,a)} \cdot (Q^{\pi_\theta}(s,a) - V^{\pi_\theta}(s)) \qquad \left( V^{\pi_\theta}(s) = \sum_a \pi_\theta(a|s) \cdot Q^{\pi_\theta}(s,a) \right)
$$

$$
= \frac{1}{1-\gamma} \cdot d_\mu^{\pi_\theta}(s) \cdot p \cdot \frac{\pi_\theta(a|s)}{\theta(s,a)} \cdot A^{\pi_\theta}(s,a). \qquad \square
$$

**Lemma 6** (Non-uniform smoothness). *Suppose $r(s,a) \in [0,1]$ for all $(s,a)$. Let $\pi_\theta := f_p(\theta)$, and $\pi_{\theta'} := f_p(\theta')$. Denote $\theta_\xi := \theta + \xi \cdot (\theta' - \theta)$ with some $\xi \in [0,1]$. Denote $A := |\mathcal{A}|$ as the total number of actions. Then we have,*

- *for $p \geq 2$, $V^{\pi_\theta}(\rho)$ is $\frac{8 \cdot p^2 \cdot A^{2/p}}{(1-\gamma)^3} \cdot \frac{1}{\min_s \|\theta_\xi(s,\cdot)\|_p^2}$-smooth, i.e.,*

$$
\left| V^{\pi_{\theta'}}(\rho) - V^{\pi_\theta}(\rho) - \left\langle \frac{\partial V^{\pi_\theta}(\rho)}{\partial \theta}, \theta' - \theta \right\rangle \right| \leq \frac{4 \cdot p^2 \cdot A^{2/p}}{(1-\gamma)^3} \cdot \frac{\|\theta' - \theta\|_2^2}{\min_s \|\theta_\xi(s,\cdot)\|_p^2};
$$

- *for $p = 1$, $V^{\pi_\theta}(\rho)$ is $\frac{8 \cdot A^2}{(1-\gamma)^3} \cdot \frac{1}{\min_s \|\theta_\xi(s,\cdot)\|_1^2}$-smooth, i.e.,*

$$
\left| V^{\pi_{\theta'}}(\rho) - V^{\pi_\theta}(\rho) - \left\langle \frac{\partial V^{\pi_\theta}(\rho)}{\partial \theta}, \theta' - \theta \right\rangle \right| \leq \frac{4 \cdot A^2}{(1-\gamma)^3} \cdot \frac{\|\theta' - \theta\|_2^2}{\min_s \|\theta_\xi(s,\cdot)\|_1^2}.
$$

*Proof.* Denote $\theta_\alpha = \theta + \alpha u$, where $\alpha \in \mathbb{R}$ and $u \in \mathbb{R}^{SA}$. For any $s \in \mathcal{S}$,

$$
\sum_a \left| \frac{\partial \pi_{\theta_\alpha}(a|s)}{\partial \alpha} \Big|_{\alpha=0} \right| = \sum_a \left| \left\langle \frac{\partial \pi_{\theta_\alpha}(a|s)}{\partial \theta_\alpha} \Big|_{\alpha=0}, \frac{\partial \theta_\alpha}{\partial \alpha} \right\rangle \right|
$$

$$
= \sum_a \left| \left\langle \frac{\partial \pi_\theta(a|s)}{\partial \theta}, u \right\rangle \right|.
$$

Since $\frac{\partial \pi_\theta(a|s)}{\partial \theta(s',\cdot)} = 0$, for $s' \neq s$,

$$
\sum_a \left| \frac{\partial \pi_{\theta_\alpha}(a|s)}{\partial \alpha} \Big|_{\alpha=0} \right| = \sum_a \left| \left\langle \frac{\partial \pi_\theta(a|s)}{\partial \theta(s,\cdot)}, u(s,\cdot) \right\rangle \right|
$$

$$
= \sum_a p \cdot \frac{\pi_\theta(a|s)}{|\theta(s,a)|} \cdot \left| u(s,a) - \pi_\theta(\cdot|s)^\top u(s,\cdot) \right|
$$

$$
= \sum_a \frac{p}{\|\theta(s,\cdot)\|_p} \cdot \pi_\theta(a|s)^{1-1/p} \cdot \left| u(s,a) - \pi_\theta(\cdot|s)^\top u(s,\cdot) \right|
$$

$$
\leq \frac{p}{\|\theta(s,\cdot)\|_p} \cdot \max_a \left| u(s,a) - \pi_\theta(\cdot|s)^\top u(s,\cdot) \right| \cdot \sum_a \pi_\theta(a|s)^{1-1/p}
$$

$$
\leq \frac{p}{\|\theta(s,\cdot)\|_p} \cdot 2 \cdot \|u\|_\infty \cdot A^{1/p} \qquad \text{(by Eq. (28))}
$$

$$
\leq \frac{2 \cdot p \cdot A^{1/p}}{\|\theta(s,\cdot)\|_p} \cdot \|u\|_2. \tag{41}
$$

Similarly, the second derivative is,

$$\sum_a \left| \frac{\partial^2 \pi_{\theta_\alpha}(a|s)}{\partial \alpha^2} \Big|_{\alpha=0} \right| = \sum_a \left| \left\langle \frac{\partial}{\partial \theta_\alpha} \left\{ \frac{\partial \pi_{\theta_\alpha}(a|s)}{\partial \alpha} \right\} \Big|_{\alpha=0}, \frac{\partial \theta_\alpha}{\partial \alpha} \right\rangle \right|$$

$$= \sum_a \left| \left\langle \frac{\partial^2 \pi_{\theta_\alpha}(a|s)}{\partial \theta_\alpha^2} \Big|_{\alpha=0} \frac{\partial \theta_\alpha}{\partial \alpha}, \frac{\partial \theta_\alpha}{\partial \alpha} \right\rangle \right|$$

$$= \sum_a \left| \left\langle \frac{\partial^2 \pi_\theta(a|s)}{\partial \theta^2(s,\cdot)} u(s,\cdot), u(s,\cdot) \right\rangle \right|.$$

Let $S(a,\theta) = \frac{\partial^2 \pi_\theta(a|s)}{\partial \theta^2(s,\cdot)} \in \mathbb{R}^{A \times A}$. $\forall i, j \in [A]$, the value of $S(a,\theta)$ is,

$$S_{i,j} = p \cdot \frac{\partial \{ \delta_{ia} \cdot \frac{\pi_\theta(a|s)}{\theta(s,a)} - \pi_\theta(a|s) \cdot \frac{\pi_\theta(i|s)}{\theta(s,i)} \}}{\partial \theta(s,j)}$$

$$= p \cdot \delta_{ia} \cdot \frac{\frac{p}{\theta(s,j)} \cdot [\delta_{ja} \pi_\theta(a|s) - \pi_\theta(a|s)\pi_\theta(j|s)] \cdot \theta(s,a) - \delta_{ja}\pi_\theta(a|s)}{\theta(s,a)^2}$$

$$- \frac{p^2}{\theta(s,j)} \cdot [\delta_{ja}\pi_\theta(a|s) - \pi_\theta(a|s)\pi_\theta(j|s)] \cdot \frac{\pi_\theta(i|s)}{\theta(s,i)}$$

$$- p \cdot \pi_\theta(a|s) \cdot \frac{\frac{p}{\theta(s,j)} \cdot [\delta_{ij}\pi_\theta(i|s) - \pi_\theta(i|s)\pi_\theta(j|s)] \cdot \theta(s,i) - \delta_{ij}\pi_\theta(i|s)}{\theta(s,i)^2}$$

$$= \delta_{ia} \cdot \delta_{ja} \cdot p \cdot (p-1) \cdot \frac{\pi_\theta(a|s)}{\theta(s,a)^2} - \delta_{ia} \cdot p^2 \cdot \frac{\pi_\theta(a|s)}{\theta(s,a)} \cdot \frac{\pi_\theta(j|s)}{\theta(s,j)} - \delta_{ja} \cdot p^2 \cdot \frac{\pi_\theta(a|s)}{\theta(s,a)} \cdot \frac{\pi_\theta(i|s)}{\theta(s,i)}$$

$$+ p \cdot \pi_\theta(a|s) \cdot \left[ 2 \cdot p \cdot \frac{\pi_\theta(i|s)}{\theta(s,i)} \cdot \frac{\pi_\theta(j|s)}{\theta(s,j)} - \delta_{ij} \cdot (p-1) \cdot \frac{\pi_\theta(i|s)}{\theta(s,i)^2} \right],$$

where the $\delta$ notation is as defined in Eq. (23). Then we have,

$$\left| \left\langle \frac{\partial^2 \pi_\theta(a|s)}{\partial \theta^2(s,\cdot)} u(s,\cdot), u(s,\cdot) \right\rangle \right| = \left| \sum_{i=1}^A \sum_{j=1}^A S_{i,j} u(s,i) u(s,j) \right|$$

$$\leq p \cdot (p-1) \cdot \frac{\pi_\theta(a|s)}{\theta(s,a)^2} \cdot u(s,a)^2 + 2 \cdot p^2 \cdot \frac{\pi_\theta(a|s)}{|\theta(s,a)|} \cdot |u(s,a)| \cdot \left| \left( \frac{\pi_\theta(\cdot|s)}{\theta(s,\cdot)} \right)^\top u(s,\cdot) \right|$$

$$+ \pi_\theta(a|s) \cdot \left[ 2 \cdot p^2 \cdot \left| \left( \frac{\pi_\theta(\cdot|s)}{\theta(s,\cdot)} \right)^\top u(s,\cdot) \right|^2 + p \cdot (p-1) \cdot \left| \left( \frac{\pi_\theta(\cdot|s)}{\theta(s,\cdot)^2} \right)^\top (u(s,\cdot) \odot u(s,\cdot)) \right| \right]$$

$$= \frac{p \cdot (p-1)}{\|\theta(s,\cdot)\|_p^2} \cdot \pi_\theta(a|s)^{1-2/p} \cdot u(s,a)^2$$

$$+ \frac{2 \cdot p^2}{\|\theta(s,\cdot)\|_p^2} \cdot \pi_\theta(a|s)^{1-1/p} \cdot |u(s,a)| \cdot \left| \left( \pi_\theta(\cdot|s)^{1-1/p} \right)^\top u(s,\cdot) \right|$$

$$+ \frac{2 \cdot p^2}{\|\theta(s,\cdot)\|_p^2} \cdot \pi_\theta(a|s) \cdot \left| \left( \pi_\theta(\cdot|s)^{1-1/p} \right)^\top u(s,\cdot) \right|^2$$

$$+ \frac{p \cdot (p-1)}{\|\theta(s,\cdot)\|_p^2} \cdot \pi_\theta(a|s) \cdot \left| \left( \pi_\theta(\cdot|s)^{1-2/p} \right)^\top (u(s,\cdot) \odot u(s,\cdot)) \right|. \tag{42}$$

**First part.** For $p \geq 2$, according to the Cauchy-Schwarz inequality, we have,

$$
\sum_a \left| \frac{\partial^2 \pi_{\theta_\alpha}(a|s)}{\partial \alpha^2} \Big|_{\alpha=0} \right| \leq \frac{p \cdot (p-1)}{\|\theta(s,\cdot)\|_p^2} \cdot \sum_a u(s,a)^2 + \frac{2 \cdot p^2}{\|\theta(s,\cdot)\|_p^2} \cdot \|u(s,\cdot)\|_2^2 \cdot \|\pi_\theta(\cdot|s)^{1-1/p}\|_2^2
$$

$$
+ \frac{2 \cdot p^2}{\|\theta(s,\cdot)\|_p^2} \cdot \|u(s,\cdot)\|_2^2 \cdot \|\pi_\theta(\cdot|s)^{1-1/p}\|_2^2 + \frac{p \cdot (p-1)}{\|\theta(s,\cdot)\|_p^2} \cdot \| \cdot \|\pi_\theta(\cdot|s)^{1-2/p}\|_\infty \cdot \|u(s,\cdot) \odot u(s,\cdot)\|_1
$$

$$
\leq \frac{2 \cdot p \cdot (p-1)}{\|\theta(s,\cdot)\|_p^2} \cdot \|u(s,\cdot)\|_2^2 + \frac{4 \cdot p^2}{\|\theta(s,\cdot)\|_p^2} \cdot A^{1/p} \cdot \|u(s,\cdot)\|_2^2 \qquad \text{(by Eq. (28))}
$$

$$
\leq \frac{2 \cdot p^2 \cdot (1 + 2 \cdot A^{1/p})}{\|\theta(s,\cdot)\|_p^2} \cdot \|u\|_2^2. \tag{43}
$$

Define $P(\alpha) \in \mathbb{R}^{S \times S}$, where $\forall (s, s')$,

$$
[P(\alpha)]_{(s,s')} = \sum_a \pi_{\theta_\alpha}(a|s) \cdot \mathcal{P}(s'|s, a). \tag{44}
$$

The derivative w.r.t. $\alpha$ is

$$
\left[ \frac{\partial P(\alpha)}{\partial \alpha} \Big|_{\alpha=0} \right]_{(s,s')} = \sum_a \left[ \frac{\partial \pi_{\theta_\alpha}(a|s)}{\partial \alpha} \Big|_{\alpha=0} \right] \cdot \mathcal{P}(s'|s, a).
$$

For any vector $x \in \mathbb{R}^S$, we have,

$$
\left[ \frac{\partial P(\alpha)}{\partial \alpha} \Big|_{\alpha=0} x \right]_{(s)} = \sum_{s'} \sum_a \left[ \frac{\partial \pi_{\theta_\alpha}(a|s)}{\partial \alpha} \Big|_{\alpha=0} \right] \cdot \mathcal{P}(s'|s, a) \cdot x(s').
$$

The $\ell_\infty$ norm is upper bounded as,

$$
\left\| \frac{\partial P(\alpha)}{\partial \alpha} \Big|_{\alpha=0} x \right\|_\infty = \max_s \left| \sum_{s'} \sum_a \left[ \frac{\partial \pi_{\theta_\alpha}(a|s)}{\partial \alpha} \Big|_{\alpha=0} \right] \cdot \mathcal{P}(s'|s, a) \cdot x(s') \right|
$$

$$
\leq \max_s \sum_a \sum_{s'} \mathcal{P}(s'|s, a) \cdot \left| \frac{\partial \pi_{\theta_\alpha}(a|s)}{\partial \alpha} \Big|_{\alpha=0} \right| \cdot \|x\|_\infty
$$

$$
= \max_s \sum_a \left| \frac{\partial \pi_{\theta_\alpha}(a|s)}{\partial \alpha} \Big|_{\alpha=0} \right| \cdot \|x\|_\infty
$$

$$
\leq \max_s \frac{2 \cdot p \cdot A^{1/p}}{\|\theta(s,\cdot)\|_p} \cdot \|u\|_2 \cdot \|x\|_\infty. \qquad \text{(by Eq. (41))} \tag{45}
$$

Similarly, taking second derivative w.r.t. $\alpha$,

$$
\left[ \frac{\partial^2 P(\alpha)}{\partial \alpha^2} \Big|_{\alpha=0} \right]_{(s,s')} = \sum_a \left[ \frac{\partial^2 \pi_{\theta_\alpha}(a|s)}{\partial \alpha^2} \Big|_{\alpha=0} \right] \cdot \mathcal{P}(s'|s, a).
$$

The $\ell_\infty$ norm is upper bounded as,

$$
\left\| \frac{\partial^2 P(\alpha)}{\partial \alpha^2} \Big|_{\alpha=0} x \right\|_\infty = \max_s \left| \sum_{s'} \sum_a \left[ \frac{\partial^2 \pi_{\theta_\alpha}(a|s)}{\partial \alpha^2} \Big|_{\alpha=0} \right] \cdot \mathcal{P}(s'|s, a) \cdot x(s') \right|
$$

$$
\leq \max_s \sum_a \sum_{s'} \mathcal{P}(s'|s, a) \cdot \left| \frac{\partial^2 \pi_{\theta_\alpha}(a|s)}{\partial \alpha^2} \Big|_{\alpha=0} \right| \cdot \|x\|_\infty
$$

$$
= \max_s \sum_a \left| \frac{\partial^2 \pi_{\theta_\alpha}(a|s)}{\partial \alpha^2} \Big|_{\alpha=0} \right| \cdot \|x\|_\infty
$$

$$
\leq \max_s \frac{2 \cdot p^2 \cdot (1 + 2 \cdot A^{1/p})}{\|\theta(s,\cdot)\|_p^2} \cdot \|u\|_2^2 \cdot \|x\|_\infty. \qquad \text{(by Eq. (43))} \tag{46}
$$

Next, consider the state value function of $\pi_{\theta_\alpha}$,

$$V^{\pi_{\theta_\alpha}}(s) = \sum_a \pi_{\theta_\alpha}(a|s) \cdot r(s,a) + \gamma \sum_a \pi_{\theta_\alpha}(a|s) \sum_{s'} \mathcal{P}(s'|s,a) \cdot V^{\pi_{\theta_\alpha}}(s'),$$

which implies,

$$V^{\pi_{\theta_\alpha}}(s) = e_s^\top M(\alpha) r_{\theta_\alpha}, \tag{47}$$

where

$$M(\alpha) = (\mathbf{Id} - \gamma P(\alpha))^{-1}, \tag{48}$$

and $r_{\theta_\alpha} \in \mathbb{R}^S$ for $s \in \mathcal{S}$ is given by

$$r_{\theta_\alpha}(s) = \sum_a \pi_{\theta_\alpha}(a|s) \cdot r(s,a).$$

Since $[P(\alpha)]_{(s,s')} \geq 0, \forall (s,s')$, and

$$M(\alpha) = (\mathbf{Id} - \gamma P(\alpha))^{-1} = \sum_{t=0}^\infty \gamma^t [P(\alpha)]^t,$$

we have $[M(\alpha)]_{(s,s')} \geq 0, \forall (s,s')$. Denote $[M(\alpha)]_{i,:}$ as the $i$-th row vector of $M(\alpha)$. We have

$$\mathbf{1} = \frac{1}{1-\gamma} \cdot (\mathbf{Id} - \gamma P(\alpha)) \mathbf{1} \implies M(\alpha)\mathbf{1} = \frac{1}{1-\gamma} \cdot \mathbf{1},$$

which implies, $\forall i$,

$$\left\| [M(\alpha)]_{i,:} \right\|_1 = \sum_j [M(\alpha)]_{(i,j)} = \frac{1}{1-\gamma}.$$

Therefore, for any vector $x \in \mathbb{R}^S$,

$$\begin{aligned}
\|M(\alpha)x\|_\infty &= \max_i \left| [M(\alpha)]_{i,:}^\top x \right| \\
&\leq \max_i \left\| [M(\alpha)]_{i,:} \right\|_1 \cdot \|x\|_\infty \\
&= \frac{1}{1-\gamma} \cdot \|x\|_\infty.
\end{aligned} \tag{49}$$

Since $r(s,a) \in [0,1], \forall (s,a)$, we have,

$$\|r_{\theta_\alpha}\|_\infty = \max_s |r_{\theta_\alpha}(s)| = \max_s \left| \sum_a \pi_{\theta_\alpha}(a|s) \cdot r(s,a) \right| \leq 1. \tag{50}$$

Since $\frac{\partial \pi_\theta(a|s)}{\partial \theta(s',\cdot)} = 0$, for $s' \neq s$,

$$\begin{aligned}
\left| \frac{\partial r_{\theta_\alpha}(s)}{\partial \alpha} \right| &= \left| \left( \frac{\partial r_{\theta_\alpha}(s)}{\partial \theta_\alpha} \right)^\top \frac{\partial \theta_\alpha}{\partial \alpha} \right| \\
&= \left| \left( \frac{\partial \{\pi_{\theta_\alpha}(\cdot|s)^\top r(s,\cdot)\}}{\partial \theta_\alpha(s,\cdot)} \right)^\top u(s,\cdot) \right| \\
&= \left| p \cdot \left( \mathrm{diag}(1/\theta_\alpha(s,\cdot)) \left( \mathrm{diag}(\pi_{\theta_\alpha}(\cdot|s)) - \pi_{\theta_\alpha}(\cdot|s)\pi_{\theta_\alpha}(\cdot|s)^\top \right) r(s,\cdot) \right)^\top u(s,\cdot) \right| \\
&\leq p \cdot \left\| \mathrm{diag}(1/\theta_\alpha(s,\cdot)) \left( \mathrm{diag}(\pi_{\theta_\alpha}(\cdot|s)) - \pi_{\theta_\alpha}(\cdot|s)\pi_{\theta_\alpha}(\cdot|s)^\top \right) r(s,\cdot) \right\|_1 \cdot \|u(s,\cdot)\|_\infty. \tag{51}
\end{aligned}$$

The $\ell_1$ norm is upper bounded as,

$$\left\|\mathrm{diag}(1/\theta_\alpha(s,\cdot))\left(\mathrm{diag}(\pi_{\theta_\alpha}(\cdot|s)) - \pi_{\theta_\alpha}(\cdot|s)\pi_{\theta_\alpha}(\cdot|s)^\top\right)r(s,\cdot)\right\|_1$$

$$= \sum_a \frac{\pi_{\theta_\alpha}(a|s)^{1-1/p}}{\|\theta_\alpha(s,\cdot)\|_p} \cdot \left|r(s,a) - \pi_{\theta_\alpha}(\cdot|s)^\top r(s,\cdot)\right|$$

$$\leq \frac{1}{\|\theta_\alpha(s,\cdot)\|_p} \cdot \max_a \left|r(s,a) - \pi_{\theta_\alpha}(\cdot|s)^\top r(s,\cdot)\right| \cdot \sum_a \pi_{\theta_\alpha}(a|s)^{1-1/p}$$

$$\leq \frac{1}{\|\theta_\alpha(s,\cdot)\|_p} \cdot \sum_a \pi_{\theta_\alpha}(a|s)^{1-1/p} \qquad (r(s,a) \in [0,1])$$

$$\leq \frac{A^{1/p}}{\|\theta_\alpha(s,\cdot)\|_p}. \qquad \text{(by Eq. (28))} \tag{52}$$

Combining Eqs. (51) and (52), we have,

$$\left\|\frac{\partial r_{\theta_\alpha}}{\partial \alpha}\right\|_\infty = \max_s \left|\frac{\partial r_{\theta_\alpha}(s)}{\partial \alpha}\right|$$

$$\leq \max_s p \cdot \left\|\mathrm{diag}(1/\theta_\alpha(s,\cdot))\left(\mathrm{diag}(\pi_{\theta_\alpha}(\cdot|s)) - \pi_{\theta_\alpha}(\cdot|s)\pi_{\theta_\alpha}(\cdot|s)^\top\right)r(s,\cdot)\right\|_1 \cdot \|u(s,\cdot)\|_\infty$$

$$\leq \max_s \frac{p \cdot A^{1/p}}{\|\theta_\alpha(s,\cdot)\|_p} \cdot \|u\|_2. \tag{53}$$

Similarly, for the second derivative, we have,

$$\left\|\frac{\partial^2 r_{\theta_\alpha}}{\partial \alpha^2}\right\|_\infty = \max_s \left|\frac{\partial^2 r_{\theta_\alpha}(s)}{\partial \alpha^2}\right|$$

$$= \max_s \left|\left(\frac{\partial}{\partial \theta_\alpha}\left\{\frac{\partial r_{\theta_\alpha}(s)}{\partial \alpha}\right\}\right)^\top \frac{\partial \theta_\alpha}{\partial \alpha}\right|$$

$$= \max_s \left|\left(\frac{\partial^2 r_{\theta_\alpha}(s)}{\partial \theta_\alpha^2}\frac{\partial \theta_\alpha}{\partial \alpha}\right)^\top \frac{\partial \theta_\alpha}{\partial \alpha}\right|$$

$$= \max_s \left|u(s,\cdot)^\top \frac{\partial^2\{\pi_{\theta_\alpha}(\cdot|s)^\top r(s,\cdot)\}}{\partial \theta_\alpha(s,\cdot)^2}u(s,\cdot)\right|$$

$$\leq \max_s \frac{3 \cdot p^2 \cdot A^{1/p}}{\|\theta_\alpha(s,\cdot)\|_p^2} \cdot \|u(s,\cdot)\|_2^2 \qquad \text{(by Eq. (29))}$$

$$\leq \max_s \frac{3 \cdot p^2 \cdot A^{1/p}}{\|\theta_\alpha(s,\cdot)\|_p^2} \cdot \|u\|_2^2. \tag{54}$$

Taking derivative w.r.t. $\alpha$ in Eq. (47), we have,

$$\frac{\partial V^{\pi_{\theta_\alpha}}(s)}{\partial \alpha} = \gamma \cdot e_s^\top M(\alpha)\frac{\partial P(\alpha)}{\partial \alpha}M(\alpha)r_{\theta_\alpha} + e_s^\top M(\alpha)\frac{\partial r_{\theta_\alpha}}{\partial \alpha}.$$

Taking second derivative w.r.t. $\alpha$, we have,

$$\frac{\partial^2 V^{\pi_{\theta_\alpha}}(s)}{\partial \alpha^2} = 2\gamma^2 \cdot e_s^\top M(\alpha)\frac{\partial P(\alpha)}{\partial \alpha}M(\alpha)\frac{\partial P(\alpha)}{\partial \alpha}M(\alpha)r_{\theta_\alpha} + \gamma \cdot e_s^\top M(\alpha)\frac{\partial^2 P(\alpha)}{\partial \alpha^2}M(\alpha)r_{\theta_\alpha}$$

$$+ 2\gamma \cdot e_s^\top M(\alpha)\frac{\partial P(\alpha)}{\partial \alpha}M(\alpha)\frac{\partial r_{\theta_\alpha}}{\partial \alpha} + e_s^\top M(\alpha)\frac{\partial^2 r_{\theta_\alpha}}{\partial \alpha^2}. \tag{55}$$

For the last term,

$$\left|e_s^\top M(\alpha)\frac{\partial^2 r_{\theta_\alpha}}{\partial \alpha^2}\Big|_{\alpha=0}\right| \leq \|e_s\|_1 \cdot \left\|M(\alpha)\frac{\partial^2 r_{\theta_\alpha}}{\partial \alpha^2}\Big|_{\alpha=0}\right\|_\infty$$

$$\leq \frac{1}{1-\gamma} \cdot \left\|\frac{\partial^2 r_{\theta_\alpha}}{\partial \alpha^2}\Big|_{\alpha=0}\right\|_\infty \qquad \text{(by Eq. (49))}$$

$$\leq \frac{1}{1-\gamma} \cdot \max_s \frac{3 \cdot p^2 \cdot A^{1/p}}{\|\theta_\alpha(s,\cdot)\|_p^2} \cdot \|u\|_2^2. \qquad \text{(by Eq. (54))} \tag{56}$$

For the second last term,

$$\left| e_s^\top M(\alpha) \frac{\partial P(\alpha)}{\partial \alpha} M(\alpha) \frac{\partial r_{\theta_\alpha}}{\partial \alpha} \Big|_{\alpha=0} \right| \le \left\| M(\alpha) \frac{\partial P(\alpha)}{\partial \alpha} M(\alpha) \frac{\partial r_{\theta_\alpha}}{\partial \alpha} \Big|_{\alpha=0} \right\|_\infty$$

$$\le \frac{1}{1-\gamma} \cdot \left\| \frac{\partial P(\alpha)}{\partial \alpha} M(\alpha) \frac{\partial r_{\theta_\alpha}}{\partial \alpha} \Big|_{\alpha=0} \right\|_\infty \qquad \text{(by Eq. (49))}$$

$$\le \frac{2 \cdot p \cdot A^{1/p} \cdot \|u\|_2}{1-\gamma} \cdot \max_s \frac{1}{\|\theta(s,\cdot)\|_p} \cdot \left\| M(\alpha) \frac{\partial r_{\theta_\alpha}}{\partial \alpha} \Big|_{\alpha=0} \right\|_\infty \qquad \text{(by Eq. (45))}$$

$$\le \frac{2 \cdot p \cdot A^{1/p} \cdot \|u\|_2}{(1-\gamma)^2} \cdot \max_s \frac{1}{\|\theta(s,\cdot)\|_p} \cdot \left\| \frac{\partial r_{\theta_\alpha}}{\partial \alpha} \Big|_{\alpha=0} \right\|_\infty \qquad \text{(by Eq. (49))}$$

$$\le \frac{2 \cdot p^2 \cdot A^{2/p} \cdot \|u\|_2}{(1-\gamma)^2} \cdot \max_s \frac{1}{\|\theta(s,\cdot)\|_p^2} \cdot \|u\|_2. \qquad \text{(by Eq. (53))} \qquad (57)$$

For the second term,

$$\left| e_s^\top M(\alpha) \frac{\partial^2 P(\alpha)}{\partial \alpha^2} M(\alpha) r_{\theta_\alpha} \Big|_{\alpha=0} \right| \le \left\| M(\alpha) \frac{\partial^2 P(\alpha)}{\partial \alpha^2} M(\alpha) r_{\theta_\alpha} \Big|_{\alpha=0} \right\|_\infty$$

$$\le \frac{1}{1-\gamma} \cdot \left\| \frac{\partial^2 P(\alpha)}{\partial \alpha^2} M(\alpha) r_{\theta_\alpha} \Big|_{\alpha=0} \right\|_\infty \qquad \text{(by Eq. (49))}$$

$$\le \frac{2 \cdot p^2 \cdot (1 + 2 \cdot A^{1/p}) \cdot \|u\|_2^2}{1-\gamma} \cdot \max_s \frac{1}{\|\theta(s,\cdot)\|_p^2} \cdot \left\| M(\alpha) r_{\theta_\alpha} \Big|_{\alpha=0} \right\|_\infty \qquad \text{(by Eq. (46))}$$

$$\le \frac{2 \cdot p^2 \cdot (1 + 2 \cdot A^{1/p}) \cdot \|u\|_2^2}{(1-\gamma)^2} \cdot \max_s \frac{1}{\|\theta(s,\cdot)\|_p^2} \cdot \left\| r_{\theta_\alpha} \Big|_{\alpha=0} \right\|_\infty \qquad \text{(by Eq. (49))}$$

$$\le \frac{2 \cdot p^2 \cdot (1 + 2 \cdot A^{1/p})}{(1-\gamma)^2} \cdot \max_s \frac{1}{\|\theta(s,\cdot)\|_p^2} \cdot \|u\|_2^2. \qquad \text{(by Eq. (50))} \qquad (58)$$

For the first term, according to Eq. (45), Eqs. (49) and (50),

$$\left| e_s^\top M(\alpha) \frac{\partial P(\alpha)}{\partial \alpha} M(\alpha) \frac{\partial P(\alpha)}{\partial \alpha} M(\alpha) r_{\theta_\alpha} \Big|_{\alpha=0} \right| \le \left\| M(\alpha) \frac{\partial P(\alpha)}{\partial \alpha} M(\alpha) \frac{\partial P(\alpha)}{\partial \alpha} M(\alpha) r_{\theta_\alpha} \Big|_{\alpha=0} \right\|_\infty$$

$$\le \frac{1}{1-\gamma} \cdot \max_s \frac{2 \cdot p \cdot A^{1/p}}{\|\theta(s,\cdot)\|_p} \cdot \|u\|_2 \cdot \frac{1}{1-\gamma} \cdot \max_s \frac{2 \cdot p \cdot A^{1/p}}{\|\theta(s,\cdot)\|_p} \cdot \|u\|_2 \cdot \frac{1}{1-\gamma} \cdot 1$$

$$= \frac{4 \cdot p^2 \cdot A^{2/p}}{(1-\gamma)^3} \cdot \max_s \frac{1}{\|\theta(s,\cdot)\|_p^2} \cdot \|u\|_2^2. \qquad (59)$$

Combining Eqs. (56) to (59) with Eq. (55), we have,

$$\left| \frac{\partial^2 V^{\pi_{\theta_\alpha}}(s)}{\partial \alpha^2} \Big|_{\alpha=0} \right| \le 2\gamma^2 \cdot \left| e_s^\top M(\alpha) \frac{\partial P(\alpha)}{\partial \alpha} M(\alpha) \frac{\partial P(\alpha)}{\partial \alpha} M(\alpha) r_{\theta_\alpha} \Big|_{\alpha=0} \right|$$

$$+ \gamma \cdot \left| e_s^\top M(\alpha) \frac{\partial^2 P(\alpha)}{\partial \alpha^2} M(\alpha) r_{\theta_\alpha} \Big|_{\alpha=0} \right|$$

$$+ 2\gamma \cdot \left| e_s^\top M(\alpha) \frac{\partial P(\alpha)}{\partial \alpha} M(\alpha) \frac{\partial r_{\theta_\alpha}}{\partial \alpha} \Big|_{\alpha=0} \right| + \left| e_s^\top M(\alpha) \frac{\partial^2 r_{\theta_\alpha}}{\partial \alpha^2} \Big|_{\alpha=0} \right|$$

$$\le \left( \frac{8 \cdot \gamma^2 \cdot p^2 \cdot A^{2/p}}{(1-\gamma)^3} + \frac{2 \cdot \gamma \cdot p^2 \cdot (1 + 2 \cdot A^{1/p})}{(1-\gamma)^2} + \frac{4 \cdot \gamma \cdot p^2 \cdot A^{2/p}}{(1-\gamma)^2} + \frac{3 \cdot p^2 \cdot A^{1/p}}{1-\gamma} \right) \cdot \max_s \frac{1}{\|\theta(s,\cdot)\|_p^2} \cdot \|u\|_2^2$$

$$\le \frac{8 \cdot p^2 \cdot A^{2/p}}{(1-\gamma)^3} \cdot \max_s \frac{1}{\|\theta(s,\cdot)\|_p^2} \cdot \|u\|_2^2, \qquad (60)$$

which implies for all $y \in \mathbb{R}^{SA}$ and $\theta$,

$$\left| y^\top \frac{\partial^2 V^{\pi_\theta}(s)}{\partial \theta^2} y \right| = \left| \left( \frac{y}{\|y\|_2} \right)^\top \frac{\partial^2 V^{\pi_\theta}(s)}{\partial \theta^2} \left( \frac{y}{\|y\|_2} \right) \right| \cdot \|y\|_2^2$$

$$\leq \max_{\|u\|_2=1} \left| \left\langle \frac{\partial^2 V^{\pi_\theta}(s)}{\partial \theta^2} u, u \right\rangle \right| \cdot \|y\|_2^2$$

$$= \max_{\|u\|_2=1} \left| \left\langle \frac{\partial^2 V^{\pi_{\theta_\alpha}}(s)}{\partial \theta_\alpha^2} \Big|_{\alpha=0} \frac{\partial \theta_\alpha}{\partial \alpha}, \frac{\partial \theta_\alpha}{\partial \alpha} \right\rangle \right| \cdot \|y\|_2^2$$

$$= \max_{\|u\|_2=1} \left| \left\langle \frac{\partial}{\partial \theta_\alpha} \left\{ \frac{\partial V^{\pi_{\theta_\alpha}}(s)}{\partial \alpha} \right\} \Big|_{\alpha=0}, \frac{\partial \theta_\alpha}{\partial \alpha} \right\rangle \right| \cdot \|y\|_2^2$$

$$= \max_{\|u\|_2=1} \left| \frac{\partial^2 V^{\pi_{\theta_\alpha}}(s)}{\partial \alpha^2} \Big|_{\alpha=0} \right| \cdot \|y\|_2^2$$

$$\leq \frac{8 \cdot p^2 \cdot A^{2/p}}{(1-\gamma)^3} \cdot \max_s \frac{1}{\|\theta(s,\cdot)\|_p^2} \cdot \|y\|_2^2. \qquad \text{(by Eq. (60))} \qquad (61)$$

Denote $\theta_\xi = \theta + \xi(\theta' - \theta)$, where $\xi \in [0,1]$. According to Taylor's theorem, $\forall s, \forall \theta, \theta'$,

$$\left| V^{\pi_{\theta'}}(s) - V^{\pi_\theta}(s) - \left\langle \frac{\partial V^{\pi_\theta}(s)}{\partial \theta}, \theta' - \theta \right\rangle \right| = \frac{1}{2} \cdot \left| (\theta' - \theta)^\top \frac{\partial^2 V^{\pi_{\theta_\xi}}(s)}{\partial \theta_\xi^2} (\theta' - \theta) \right|$$

$$\leq \frac{4 \cdot p^2 \cdot A^{2/p}}{(1-\gamma)^3} \cdot \max_s \frac{1}{\|\theta_\xi(s,\cdot)\|_p^2} \cdot \|\theta' - \theta\|_2^2 \qquad \text{(by Eq. (61))}$$

$$= \frac{4 \cdot p^2 \cdot A^{2/p}}{(1-\gamma)^3} \cdot \frac{1}{\min_s \|\theta_\xi(s,\cdot)\|_p^2} \cdot \|\theta' - \theta\|_2^2. \qquad (62)$$

Since $V^{\pi_\theta}(s)$ is $\frac{8 \cdot p^2 \cdot A^{2/p}}{(1-\gamma)^3} \cdot \frac{1}{\min_s \|\theta_\xi(s,\cdot)\|_p^2}$-smooth, for any state $s$, $V^{\pi_\theta}(\rho) = \mathbb{E}_{s \sim \rho}[V^{\pi_\theta}(s)]$ is also $\frac{8 \cdot p^2 \cdot A^{2/p}}{(1-\gamma)^3} \cdot \frac{1}{\min_s \|\theta_\xi(s,\cdot)\|_p^2}$-smooth.

**Second part.** For $p = 1$, we have,

$$\left| \left\langle \frac{\partial^2 \pi_\theta(a|s)}{\partial \theta^2(s,\cdot)} u(s,\cdot), u(s,\cdot) \right\rangle \right| \leq \frac{2 \cdot p^2}{\|\theta(s,\cdot)\|_p^2} \cdot \pi_\theta(a|s)^{1-1/p} \cdot |u(s,a)| \cdot \left| \left( \pi_\theta(\cdot|s)^{1-1/p} \right)^\top u(s,\cdot) \right|$$

$$+ \frac{2 \cdot p^2}{\|\theta(s,\cdot)\|_p^2} \cdot \pi_\theta(a|s) \cdot \left| \left( \pi_\theta(\cdot|s)^{1-1/p} \right)^\top u(s,\cdot) \right|^2. \qquad \text{(by Eq. (42))}$$

Therefore we have,

$$\sum_a \left| \frac{\partial^2 \pi_{\theta_\alpha}(a|s)}{\partial \alpha^2} \Big|_{\alpha=0} \right| \leq \frac{4 \cdot p^2}{\|\theta(s,\cdot)\|_p^2} \cdot \|u(s,\cdot)\|_2^2 \cdot \|\pi_\theta(\cdot|s)^{1-1/p}\|_2^2$$

$$\leq \frac{4 \cdot p^2 \cdot A}{\|\theta(s,\cdot)\|_p^2} \cdot \|u\|_2^2. \qquad (63)$$

Similar to Eq. (46), we have,

$$\left\| \frac{\partial^2 P(\alpha)}{\partial \alpha^2} \Big|_{\alpha=0} x \right\|_\infty \leq \max_s \sum_a \left| \frac{\partial^2 \pi_{\theta_\alpha}(a|s)}{\partial \alpha^2} \Big|_{\alpha=0} \right| \cdot \|x\|_\infty$$

$$\leq \max_s \frac{4 \cdot p^2 \cdot A}{\|\theta(s,\cdot)\|_p^2} \cdot \|u\|_2^2 \cdot \|x\|_\infty. \qquad \text{(by Eq. (63))} \qquad (64)$$

Similar to Eq. (54), for the second derivative, we have,

$$\left\|\frac{\partial^2 r_{\theta_\alpha}}{\partial \alpha^2}\right\|_\infty = \max_s \left| u(s,\cdot)^\top \frac{\partial^2 \{\pi_{\theta_\alpha}(\cdot|s)^\top r(s,\cdot)\}}{\partial \theta_\alpha(s,\cdot)^2} u(s,\cdot)\right|$$

$$\leq \max_s \frac{2 \cdot p^2 \cdot A^{1/p}}{\|\theta_\alpha(s,\cdot)\|_p^2} \cdot \|u(s,\cdot)\|_2^2 \qquad \text{(by Eq. (30))}$$

$$\leq \max_s \frac{2 \cdot p^2 \cdot A^{1/p}}{\|\theta_\alpha(s,\cdot)\|_p^2} \cdot \|u\|_2^2. \tag{65}$$

Similar to Eq. (56), we have,

$$\left| e_s^\top M(\alpha) \frac{\partial^2 r_{\theta_\alpha}}{\partial \alpha^2}\Big|_{\alpha=0}\right| \leq \|e_s\|_1 \cdot \left\| M(\alpha) \frac{\partial^2 r_{\theta_\alpha}}{\partial \alpha^2}\Big|_{\alpha=0}\right\|_\infty$$

$$\leq \frac{1}{1-\gamma} \cdot \left\|\frac{\partial^2 r_{\theta_\alpha}}{\partial \alpha^2}\Big|_{\alpha=0}\right\|_\infty \qquad \text{(by Eq. (49))}$$

$$\leq \frac{1}{1-\gamma} \cdot \max_s \frac{2 \cdot p^2 \cdot A^{1/p}}{\|\theta_\alpha(s,\cdot)\|_p^2} \cdot \|u\|_2^2. \qquad \text{(by Eq. (65))} \tag{66}$$

Similar to Eq. (58), we have,

$$\left| e_s^\top M(\alpha) \frac{\partial^2 P(\alpha)}{\partial \alpha^2} M(\alpha) r_{\theta_\alpha}\Big|_{\alpha=0}\right| \leq \frac{1}{1-\gamma} \cdot \left\|\frac{\partial^2 P(\alpha)}{\partial \alpha^2} M(\alpha) r_{\theta_\alpha}\Big|_{\alpha=0}\right\|_\infty \qquad \text{(by Eq. (49))}$$

$$\leq \frac{4 \cdot p^2 \cdot A \cdot \|u\|_2^2}{1-\gamma} \cdot \max_s \frac{1}{\|\theta(s,\cdot)\|_p^2} \cdot \left\| M(\alpha) r_{\theta_\alpha}\Big|_{\alpha=0}\right\|_\infty \qquad \text{(by Eq. (64))}$$

$$\leq \frac{4 \cdot p^2 \cdot A \cdot \|u\|_2^2}{(1-\gamma)^2} \cdot \max_s \frac{1}{\|\theta(s,\cdot)\|_p^2} \cdot \left\| r_{\theta_\alpha}\Big|_{\alpha=0}\right\|_\infty \qquad \text{(by Eq. (49))}$$

$$\leq \frac{4 \cdot p^2 \cdot A}{(1-\gamma)^2} \cdot \max_s \frac{1}{\|\theta(s,\cdot)\|_p^2} \cdot \|u\|_2^2. \qquad \text{(by Eq. (50))} \tag{67}$$

Combining Eqs. (57) and (59), Eqs. (66) and (67) with Eq. (55), we have,

$$\left|\frac{\partial^2 V^{\pi_{\theta_\alpha}}(s)}{\partial \alpha^2}\Big|_{\alpha=0}\right| \leq 2\gamma^2 \cdot \left| e_s^\top M(\alpha) \frac{\partial P(\alpha)}{\partial \alpha} M(\alpha) \frac{\partial P(\alpha)}{\partial \alpha} M(\alpha) r_{\theta_\alpha}\Big|_{\alpha=0}\right|$$

$$+ \gamma \cdot \left| e_s^\top M(\alpha) \frac{\partial^2 P(\alpha)}{\partial \alpha^2} M(\alpha) r_{\theta_\alpha}\Big|_{\alpha=0}\right|$$

$$+ 2\gamma \cdot \left| e_s^\top M(\alpha) \frac{\partial P(\alpha)}{\partial \alpha} M(\alpha) \frac{\partial r_{\theta_\alpha}}{\partial \alpha}\Big|_{\alpha=0}\right| + \left| e_s^\top M(\alpha) \frac{\partial^2 r_{\theta_\alpha}}{\partial \alpha^2}\Big|_{\alpha=0}\right|$$

$$\leq \left(\frac{8 \cdot \gamma^2 \cdot p^2 \cdot A^{2/p}}{(1-\gamma)^3} + \frac{4 \cdot \gamma \cdot p^2 \cdot A}{(1-\gamma)^2} + \frac{4 \cdot \gamma \cdot p^2 \cdot A^{2/p}}{(1-\gamma)^2} + \frac{2 \cdot p^2 \cdot A^{1/p}}{1-\gamma}\right) \cdot \max_s \frac{1}{\|\theta(s,\cdot)\|_p^2} \cdot \|u\|_2^2$$

$$\leq \frac{8 \cdot A^2}{(1-\gamma)^3} \cdot \max_s \frac{1}{\|\theta(s,\cdot)\|_1^2} \cdot \|u\|_2^2. \qquad (p=1) \tag{68}$$

Similar to Eq. (61), Eq. (68) implies for all $y \in \mathbb{R}^{SA}$ and $\theta$,

$$\left| y^\top \frac{\partial^2 V^{\pi_\theta}(s)}{\partial \theta^2} y\right| \leq \max_{\|u\|_2=1} \left|\frac{\partial^2 V^{\pi_{\theta_\alpha}}(s)}{\partial \alpha^2}\Big|_{\alpha=0}\right| \cdot \|y\|_2^2$$

$$\leq \frac{8 \cdot A^2}{(1-\gamma)^3} \cdot \max_s \frac{1}{\|\theta(s,\cdot)\|_1^2} \cdot \|y\|_2^2. \qquad \text{(Eq. (68))} \tag{69}$$

Similar to Eq. (62), we have, $\forall s, \forall \theta, \theta'$,

$$\left| V^{\pi_{\theta'}}(s) - V^{\pi_\theta}(s) - \left\langle\frac{\partial V^{\pi_\theta}(s)}{\partial \theta}, \theta' - \theta\right\rangle\right| \leq \frac{4 \cdot A^2}{(1-\gamma)^3} \cdot \max_s \frac{\|\theta' - \theta\|_2^2}{\|\theta_\xi(s,\cdot)\|_1^2} \qquad \text{(Eq. (69))}$$

$$= \frac{4 \cdot A^2}{(1-\gamma)^3} \cdot \frac{\|\theta' - \theta\|_2^2}{\min_s \|\theta_\xi(s,\cdot)\|_1^2}.$$

Since $V^{\pi_\theta}(s)$ is $\frac{8 \cdot A^2}{(1-\gamma)^3} \cdot \frac{1}{\min_s \|\theta_\xi(s,\cdot)\|_1^2}$-smooth, for any state $s$, $V^{\pi_\theta}(\rho) = \mathbb{E}_{s \sim \rho}[V^{\pi_\theta}(s)]$ is also $\frac{8 \cdot A^2}{(1-\gamma)^3} \cdot \frac{1}{\min_s \|\theta_\xi(s,\cdot)\|_1^2}$-smooth. $\qquad \square$

**Lemma 7** (Non-uniform Łojasiewicz). *Suppose $\mu(s) > 0$ for all state $s$ and $\pi_\theta := f_p(\theta)$. Then,*

$$\left\| \frac{\partial V^{\pi_\theta}(\mu)}{\partial \theta} \right\|_2 \geq \frac{p}{\sqrt{S}} \cdot \left\| \frac{d_\rho^{\pi^*}}{d_\mu^{\pi_\theta}} \right\|_\infty^{-1} \cdot \frac{\min_s \pi_\theta(a^*(s)|s)^{1-1/p}}{\max_s \|\theta(s,\cdot)\|_p} \cdot [V^*(\rho) - V^{\pi_\theta}(\rho)],$$

*where $a^*(s) := \arg\max_a \pi^*(a|s)$, $\forall s \in \mathcal{S}$, is the action that the optimal policy $\pi^*$ takes under $s$.*

*Proof.* Note that $a^*(s)$ is the action that optimal policy $\pi^*$ selects under state $s$.

$$\left\| \frac{\partial V^{\pi_\theta}(\mu)}{\partial \theta} \right\|_2 = \left[ \sum_{s,a} \left( \frac{\partial V^{\pi_\theta}(\mu)}{\partial \theta(s,a)} \right)^2 \right]^{\frac{1}{2}}$$

$$\geq \left[ \sum_s \left( \frac{\partial V^{\pi_\theta}(\mu)}{\partial \theta(s,a^*(s))} \right)^2 \right]^{\frac{1}{2}}$$

$$\geq \frac{1}{\sqrt{S}} \sum_s \left| \frac{\partial V^{\pi_\theta}(\mu)}{\partial \theta(s,a^*(s))} \right| \qquad \text{(by Cauchy-Schwarz, } \|x\|_1 = |\langle \mathbf{1}, |x|\rangle| \leq \|\mathbf{1}\|_2 \cdot \|x\|_2)$$

$$= \frac{1}{1-\gamma} \cdot \frac{1}{\sqrt{S}} \sum_s \left| d_\mu^{\pi_\theta}(s) \cdot p \cdot \frac{\pi_\theta(a^*(s)|s)}{\theta(s,a^*(s))} \cdot A^{\pi_\theta}(s,a^*(s)) \right| \qquad \text{(by Lemma 5)}$$

$$= \frac{1}{1-\gamma} \cdot \frac{1}{\sqrt{S}} \sum_s d_\mu^{\pi_\theta}(s) \cdot p \cdot \frac{\pi_\theta(a^*(s)|s)}{|\theta(s,a^*(s))|} \cdot |A^{\pi_\theta}(s,a^*(s))| . \qquad \left( d_\mu^{\pi_\theta}(s) \geq 0, \ \pi_\theta(a^*(s)|s) \geq 0 \right)$$

Define the distribution mismatch coefficient as $\left\| \frac{d_\rho^{\pi^*}}{d_\mu^{\pi_\theta}} \right\|_\infty := \max_s \frac{d_\rho^{\pi^*}(s)}{d_\mu^{\pi_\theta}(s)}$. We have,

$$\left\| \frac{\partial V^{\pi_\theta}(\mu)}{\partial \theta} \right\|_2 \geq \frac{1}{1-\gamma} \cdot \frac{1}{\sqrt{S}} \sum_s \frac{d_\mu^{\pi_\theta}(s)}{d_\rho^{\pi^*}(s)} \cdot d_\rho^{\pi^*}(s) \cdot p \cdot \frac{\pi_\theta(a^*(s)|s)}{|\theta(s,a^*(s))|} \cdot |A^{\pi_\theta}(s,a^*(s))|$$

$$= \frac{1}{1-\gamma} \cdot \frac{1}{\sqrt{S}} \sum_s \frac{d_\mu^{\pi_\theta}(s)}{d_\rho^{\pi^*}(s)} \cdot d_\rho^{\pi^*}(s) \cdot p \cdot \frac{1}{\|\theta(s,\cdot)\|_p} \cdot (\pi_\theta(a^*(s)|s))^{1-1/p} \cdot |A^{\pi_\theta}(s,a^*(s))|$$

$$\geq \frac{1}{1-\gamma} \cdot \frac{1}{\sqrt{S}} \cdot \left\| \frac{d_\rho^{\pi^*}}{d_\mu^{\pi_\theta}} \right\|_\infty^{-1} \cdot p \cdot \min_s \frac{1}{\|\theta(s,\cdot)\|_p} \cdot \min_s \pi_\theta(a^*(s)|s)^{1-1/p} \cdot \sum_s d_\rho^{\pi^*}(s) \cdot |A^{\pi_\theta}(s,a^*(s))|$$

$$\geq \frac{1}{1-\gamma} \cdot \frac{1}{\sqrt{S}} \cdot \left\| \frac{d_\rho^{\pi^*}}{d_\mu^{\pi_\theta}} \right\|_\infty^{-1} \cdot p \cdot \min_s \frac{1}{\|\theta(s,\cdot)\|_p} \cdot \min_s \pi_\theta(a^*(s)|s)^{1-1/p} \cdot \sum_s d_\rho^{\pi^*}(s) \cdot A^{\pi_\theta}(s,a^*(s))$$

$$= \frac{p}{\sqrt{S}} \cdot \left\| \frac{d_\rho^{\pi^*}}{d_\mu^{\pi_\theta}} \right\|_\infty^{-1} \cdot \frac{\min_s \pi_\theta(a^*(s)|s)^{1-1/p}}{\max_s \|\theta(s,\cdot)\|_p} \cdot \frac{1}{1-\gamma} \sum_s d_\rho^{\pi^*}(s) \sum_a \pi^*(a|s) \cdot A^{\pi_\theta}(s,a)$$

$$= \frac{p}{\sqrt{S}} \cdot \left\| \frac{d_\rho^{\pi^*}}{d_\mu^{\pi_\theta}} \right\|_\infty^{-1} \cdot \frac{\min_s \pi_\theta(a^*(s)|s)^{1-1/p}}{\max_s \|\theta(s,\cdot)\|_p} \cdot [V^*(\rho) - V^{\pi_\theta}(\rho)],$$

where the last equation is according to the performance difference lemma of Lemma 8. $\qquad \square$

**Lemma 8** (Performance difference lemma [9]). *For any policies $\pi$ and $\pi'$,*

$$V^\pi(\rho) - V^{\pi'}(\rho) = \frac{1}{1-\gamma} \sum_s d_\rho^\pi(s) \sum_a \pi(a|s) \cdot A^{\pi'}(s,a).$$

---
**Algorithm 2** Escort Policy Gradient Method with Parameter Normalization
---
  **Input:** Learning rate $\eta > 0$.
  **Output:** Policies $\pi_{\theta_t} = f_p(\theta_t)$.
  Initialize parameter $\theta_1(s,a)$ for all $(s,a) \in \mathcal{S} \times \mathcal{A}$.
  Normalize parameter $\tilde{\theta}_1(s,a) \leftarrow \frac{\theta_1(s,a)}{\|\theta_1(s,\cdot)\|_p}$ for all $(s,a) \in \mathcal{S} \times \mathcal{A}$.
  **for** $t = 1$ **to** $T$ **do**
    $\tilde{\zeta}_{t+1}(s,a) \leftarrow \tilde{\theta}_t(s,a) + \eta \cdot \frac{\partial V^{\pi_{\tilde{\theta}_t}}(\mu)}{\partial \tilde{\theta}_t(s,a)}$ for all $(s,a)$.
    $\tilde{\theta}_{t+1}(s,a) \leftarrow \frac{\tilde{\zeta}_{t+1}(s,a)}{\|\tilde{\zeta}_{t+1}(s,\cdot)\|_p}$ for all $(s,a)$.
  **end for**
---

### C.3.1 An equivalent algorithm

For convenience of analysis, we introduce Algorithm 2, which is equivalent to Algorithm 1 as shown in Lemma 9.

**Lemma 9.** *Using the escort transform $\pi_\theta = f_p(\theta)$, Algorithm 2 with constant learning rate $\eta$ and Algorithm 1 with learning rate $\eta_t(s) = \eta \cdot \|\theta_t(s,\cdot)\|_p^2$ are equivalent, i.e., for all $(s,a)$,*

$$\tilde{\theta}_t(s,a) = \frac{\theta_t(s,a)}{\|\theta_t(s,\cdot)\|_p}, \text{ and}$$

$$\pi_{\tilde{\theta}_t}(a|s) = \pi_{\theta_t}(a|s).$$

*Proof.* For $t = 1$, according to Algorithm 2, we have, for all $(s,a)$, $\tilde{\theta}_1(s,a) = \frac{\theta_1(s,a)}{\|\theta_1(s,\cdot)\|_p}$, and,

$$\pi_{\tilde{\theta}_1}(a|s) = \frac{|\tilde{\theta}_1(s,a)|^p}{\sum_{a'}|\tilde{\theta}_1(s,a')|^p} = \frac{|\theta_1(s,a)|^p}{\sum_{a'}|\theta_1(s,a')|^p} \cdot \frac{1}{\|\theta_1(s,\cdot)\|_p^p} \cdot \|\theta_1(s,\cdot)\|_p^p = \pi_{\theta_1}(a|s). \quad (70)$$

Suppose $\tilde{\theta}_t(s,a) = \frac{\theta_t(s,a)}{\|\theta_t(s,\cdot)\|_p}$ for some $t \geq 1$. Using similar calculation as in Eq. (70), we have, for all $(s,a)$, $\pi_{\tilde{\theta}_t}(a|s) = \pi_{\theta_t}(a|s)$, and,

$$
\begin{aligned}
\tilde{\zeta}_{t+1}(s,a) &\leftarrow \tilde{\theta}_t(s,a) + \eta \cdot \frac{\partial V^{\pi_{\tilde{\theta}_t}}(\mu)}{\partial \tilde{\theta}_t(s,a)} \quad &\text{(Algorithm 2)} \\
&= \frac{\theta_t(s,a)}{\|\theta_t(s,\cdot)\|_p} + \eta \cdot \|\theta_t(s,\cdot)\|_p \cdot \frac{\partial V^{\pi_{\theta_t}}(\mu)}{\partial \theta_t(s,a)} \quad &\text{(induction hypothesis and } \pi_{\tilde{\theta}_t} = \pi_{\theta_t}) \\
&= \frac{\theta_t(s,a)}{\|\theta_t(s,\cdot)\|_p} + \eta_t(s) \cdot \frac{1}{\|\theta_t(s,\cdot)\|_p} \cdot \frac{\partial V^{\pi_{\theta_t}}(\mu)}{\partial \theta_t(s,a)} \quad &\left(\eta_t(s) = \eta \cdot \|\theta_t(s,\cdot)\|_p^2\right) \\
&= \frac{1}{\|\theta_t(s,\cdot)\|_p} \cdot \left(\theta_t(s,a) + \eta_t(s) \cdot \frac{\partial V^{\pi_{\theta_t}}(\mu)}{\partial \theta_t(s,a)}\right) \\
&= \frac{1}{\|\theta_t(s,\cdot)\|_p} \cdot \theta_{t+1}(s,a). \quad &\text{(Algorithm 1)} \quad (71)
\end{aligned}
$$

Therefore we have,

$$
\begin{aligned}
\tilde{\theta}_{t+1}(s,a) &\leftarrow \frac{\tilde{\zeta}_{t+1}(s,a)}{\|\tilde{\zeta}_{t+1}(s,\cdot)\|_p} \quad &\text{(Algorithm 2)} \\
&= \frac{1}{\|\theta_t(s,\cdot)\|_p} \cdot \theta_{t+1}(s,a) \cdot \frac{\|\theta_t(s,\cdot)\|_p}{\|\theta_{t+1}(s,\cdot)\|_p} \quad &\text{(by Eq. (71))} \\
&= \frac{\theta_{t+1}(s,a)}{\|\theta_{t+1}(s,\cdot)\|_p}.
\end{aligned}
$$

Using similar calculation as in Eq. (70), we have, for all $(s,a)$, $\pi_{\tilde{\theta}_{t+1}}(a|s) = \pi_{\theta_{t+1}}(a|s)$. $\qquad \square$

**Theorem 3.** Following the escort policy gradient Algorithm 1 with any initialization such that $|\theta_1(s,a)| > 0, \forall(s,a)$, and $\eta_t(s) = \frac{(1-\gamma)^3 \cdot \|\theta_t(s,\cdot)\|_p^2}{10 \cdot p^2 \cdot A^{2/p}}$ to get $\{\theta_t\}_{t \geq 1}$, for all $t \geq 1$, the following sub-optimality upper bounds hold for $\pi_{\theta_t}$,

- for $p \geq 2$, we have,

$$V^*(\rho) - V^{\pi_{\theta_t}}(\rho) \leq \frac{20 \cdot A^{2/p} \cdot S}{c^{2-2/p} \cdot (1-\gamma)^6 \cdot t} \cdot \left\| \frac{d_\mu^{\pi^*}}{\mu} \right\|_\infty^2 \cdot \left\| \frac{1}{\mu} \right\|_\infty ;$$

- for $p = 1$, we have,

$$V^*(\mu) - V^{\pi_{\theta_t}}(\mu) \leq \frac{20 \cdot A^2 \cdot S}{(1-\gamma)^6 \cdot t} \cdot \left\| \frac{d_\mu^{\pi^*}}{\mu} \right\|_\infty^2 \cdot \left\| \frac{1}{\mu} \right\|_\infty ,$$

where $c := \inf_{s \in \mathcal{S}} \inf_{t \geq 1} \pi_{\theta_t}(a^*(s)|s) > 0$ is problem- and initialization-dependent constant, $A := |\mathcal{A}|$ and $S := |\mathcal{S}|$ are the total number of actions and states, respectively, and $\mu \in \Delta(\mathcal{S})$ is an initial state distribution which provides initial states for the policy gradient Algorithm 1.

*Proof.* Note that for any $\theta$ and $\mu$,

$$
\begin{aligned}
d_\mu^{\pi_\theta}(s) &= \mathop{\mathbb{E}}_{s_0 \sim \mu} \left[ d_\mu^{\pi_\theta}(s) \right] \\
&= \mathop{\mathbb{E}}_{s_0 \sim \mu} \left[ (1-\gamma) \sum_{t=0}^\infty \gamma^t \Pr(s_t = s | s_0, \pi_\theta, \mathcal{P}) \right] \\
&\geq \mathop{\mathbb{E}}_{s_0 \sim \mu} \left[ (1-\gamma) \Pr(s_0 = s | s_0) \right] \\
&= (1-\gamma) \cdot \mu(s).
\end{aligned}
\tag{72}
$$

According to the value sub-optimality lemma of Lemma 10,

$$
\begin{aligned}
V^*(\rho) - V^{\pi_\theta}(\rho) &= \frac{1}{1-\gamma} \sum_s d_\rho^{\pi_\theta}(s) \sum_a (\pi^*(a|s) - \pi_\theta(a|s)) \cdot Q^*(s,a) \\
&= \frac{1}{1-\gamma} \sum_s \frac{d_\rho^{\pi_\theta}(s)}{d_\mu^{\pi_\theta}(s)} \cdot d_\mu^{\pi_\theta}(s) \sum_a (\pi^*(a|s) - \pi_\theta(a|s)) \cdot Q^*(s,a) \\
&\leq \frac{1}{1-\gamma} \cdot \left\| \frac{1}{d_\mu^{\pi_\theta}} \right\|_\infty \sum_s d_\mu^{\pi_\theta}(s) \sum_a (\pi^*(a|s) - \pi_\theta(a|s)) \cdot Q^*(s,a) \\
&\leq \frac{1}{(1-\gamma)^2} \cdot \left\| \frac{1}{\mu} \right\|_\infty \sum_s d_\mu^{\pi_\theta}(s) \sum_a (\pi^*(a|s) - \pi_\theta(a|s)) \cdot Q^*(s,a) \qquad \left( \text{by Eq. (72) and } \min_s \mu(s) > 0 \right) \\
&= \frac{1}{1-\gamma} \cdot \left\| \frac{1}{\mu} \right\|_\infty \cdot [V^*(\mu) - V^{\pi_\theta}(\mu)],
\end{aligned}
\tag{73}
$$

where the first inequality is because of

$$\sum_a (\pi^*(a|s) - \pi_\theta(a|s)) \cdot Q^*(s,a) \geq 0,$$

and the last equation is again by Lemma 10.

For $p \geq 2$ and $p = 1$, according to Lemma 6, $V^{\pi_\theta}(\mu)$ is $\beta$-smooth with $\beta = \frac{8 \cdot p^2 \cdot A^{2/p}}{(1-\gamma)^3} \cdot \frac{1}{\min_s \|\theta_\xi(s,\cdot)\|_p^2}$, i.e., we have, in Algorithm 2,

$$\left| V^{\pi_{\tilde\zeta_{t+1}}}(\mu) - V^{\pi_{\tilde\theta_t}}(\mu) - \left\langle \frac{\partial V^{\pi_{\tilde\theta_t}}(\mu)}{\partial \tilde\theta_t}, \tilde\zeta_{t+1} - \tilde\theta_t \right\rangle \right| \leq \frac{4 \cdot p^2 \cdot A^{2/p}}{(1-\gamma)^3} \cdot \frac{\|\tilde\zeta_{t+1} - \tilde\theta_t\|_2^2}{\min_s \|\tilde\theta_{t,\xi}(s,\cdot)\|_p^2}, \tag{74}$$

where

$$
\begin{aligned}
\tilde\theta_{t,\xi} &:= \tilde\theta_t + \xi \cdot (\tilde\zeta_{t+1} - \tilde\theta_t) \\
&= \tilde\theta_t + \xi \cdot \eta \cdot \frac{\partial V^{\pi_{\tilde\theta_t}}(\mu)}{\partial \tilde\theta_t}, \qquad \text{(Algorithm 2)}
\end{aligned}
$$

for some $\xi \in [0, 1]$. Denote $s_\xi := \arg\min_s \|\tilde{\theta}_{t,\xi}(s, \cdot)\|_p^2$. We have,

$$\|\tilde{\theta}_{t,\xi}(s_\xi, \cdot)\|_p \geq \|\tilde{\theta}_t(s_\xi, \cdot)\|_p - \xi \cdot \eta \cdot \left\| \frac{\partial V^{\pi_{\tilde{\theta}_t}}(\mu)}{\partial \tilde{\theta}_t(s_\xi, \cdot)} \right\|_p \qquad \text{(triangle inequality)}$$

$$\geq \min_s \|\tilde{\theta}_t(s, \cdot)\|_p - \xi \cdot \eta \cdot \left\| \frac{\partial V^{\pi_{\tilde{\theta}_t}}(\mu)}{\partial \tilde{\theta}_t(s_\xi, \cdot)} \right\|_p . \tag{75}$$

The $\ell_p$ gradient norm can be upper bounded as,

$$\left\| \frac{\partial V^{\pi_\theta}(\mu)}{\partial \theta(s, \cdot)} \right\|_p = \left[ \sum_a \left| \frac{1}{1-\gamma} \cdot d_\mu^{\pi_\theta}(s) \cdot p \cdot \frac{\pi_\theta(a|s)}{\theta(s, a)} \cdot A^{\pi_\theta}(s, a) \right|^p \right]^{\frac{1}{p}} \qquad \text{(by Lemma 5)}$$

$$\leq \frac{p}{1-\gamma} \cdot \left[ \sum_a \left| \frac{\pi_\theta(a|s)}{\theta(s, a)} \cdot A^{\pi_\theta}(s, a) \right|^p \right]^{\frac{1}{p}}$$

$$= \frac{p}{1-\gamma} \cdot \frac{1}{\|\theta(s, \cdot)\|_p} \cdot \left[ \sum_a \left( \pi_\theta(a|s)^{1-1/p} \cdot |A^{\pi_\theta}(s, a)| \right)^p \right]^{\frac{1}{p}}$$

$$\leq \frac{p}{1-\gamma} \cdot \frac{1}{\|\theta(s, \cdot)\|_p} \cdot \left[ \sum_a \left( 1 \cdot \frac{1}{1-\gamma} \right)^p \right]^{\frac{1}{p}}$$

$$\leq \frac{p \cdot A^{1/p}}{(1-\gamma)^2} \cdot \max_s \frac{1}{\|\theta(s, \cdot)\|_p} . \tag{76}$$

Combining Eqs. (75) and (76), we have,

$$\min_s \|\tilde{\theta}_{t,\xi}(s, \cdot)\|_p \geq \min_s \|\tilde{\theta}_t(s, \cdot)\|_p - \xi \cdot \eta \cdot \frac{p \cdot A^{1/p}}{(1-\gamma)^2} \cdot \frac{1}{\min_s \|\tilde{\theta}_t(s, \cdot)\|_p}$$

$$= 1 - \xi \cdot \eta \cdot \frac{p \cdot A^{1/p}}{(1-\gamma)^2}. \qquad \left( \|\tilde{\theta}_t(s, \cdot)\|_p = 1, \text{ for all } s, \text{ Algorithm 2} \right)$$

$$= 1 - \xi \cdot \frac{1-\gamma}{10 \cdot p \cdot A^{1/p}} \qquad \left( \eta = \frac{(1-\gamma)^3}{10 \cdot p^2 \cdot A^{2/p}}, \text{ by Lemma 9} \right)$$

$$\geq 1 - \frac{1-\gamma}{10 \cdot p \cdot A^{1/p}} \qquad (\xi \in [0, 1])$$

$$= \left( 1 - \frac{2}{\sqrt{5}} \right) \cdot \left( 1 - \frac{5 + 2\sqrt{5}}{10} \cdot \frac{1-\gamma}{p \cdot A^{1/p}} \right) + \frac{2}{\sqrt{5}}$$

$$\geq \frac{2}{\sqrt{5}}. \qquad \left( p \geq 2, \ A^{1/p} \geq 1, \ 1-\gamma \in (0, 1] \right) \tag{77}$$

Combining Eqs. (74) and (77), we have,

$$\left| V^{\pi_{\tilde{\zeta}_{t+1}}}(\mu) - V^{\pi_{\tilde{\theta}_t}}(\mu) - \left\langle \frac{\partial V^{\pi_{\tilde{\theta}_t}}(\mu)}{\partial \tilde{\theta}_t}, \tilde{\zeta}_{t+1} - \tilde{\theta}_t \right\rangle \right| \leq \frac{4 \cdot p^2 \cdot A^{2/p}}{(1-\gamma)^3} \cdot \frac{\|\tilde{\zeta}_{t+1} - \tilde{\theta}_t\|_2^2}{\min_s \|\tilde{\theta}_{t,\xi}(s, \cdot)\|_p^2}$$

$$\leq \frac{5 \cdot p^2 \cdot A^{2/p}}{(1-\gamma)^3} \cdot \|\tilde{\zeta}_{t+1} - \tilde{\theta}_t\|_2^2, \tag{78}$$

which implies,

$$V^{\pi_{\tilde{\theta}_t}}(\mu) - V^{\pi_{\tilde{\theta}_{t+1}}}(\mu) = V^{\pi_{\tilde{\theta}_t}}(\mu) - V^{\pi_{\tilde{\zeta}_{t+1}}}(\mu) \qquad \left( \tilde{\theta}_{t+1}(s,a) = \frac{\tilde{\zeta}_{t+1}(s,a)}{\|\tilde{\zeta}_{t+1}(s,\cdot)\|_p}, \text{ Algorithm 2} \right)$$

$$\leq -\left\langle \frac{\partial V^{\pi_{\tilde{\theta}_t}}(\mu)}{\partial \tilde{\theta}_t}, \tilde{\zeta}_{t+1} - \tilde{\theta}_t \right\rangle + \frac{5 \cdot p^2 \cdot A^{2/p}}{(1-\gamma)^3} \cdot \|\tilde{\zeta}_{t+1} - \tilde{\theta}_t\|_2^2 \qquad \text{(Eq. (78))}$$

$$= -\eta \cdot \left\| \frac{\partial V^{\pi_{\tilde{\theta}_t}}(\mu)}{\partial \tilde{\theta}_t} \right\|_2^2 + \frac{5 \cdot p^2 \cdot A^{2/p}}{(1-\gamma)^3} \cdot \eta^2 \cdot \left\| \frac{\partial V^{\pi_{\tilde{\theta}_t}}(\mu)}{\partial \tilde{\theta}_t} \right\|_2^2 \qquad \left( \tilde{\zeta}_{t+1} = \tilde{\theta}_t + \eta \cdot \frac{\partial V^{\pi_{\tilde{\theta}_t}}(\mu)}{\partial \tilde{\theta}_t}, \text{ Algorithm 2} \right)$$

$$= -\frac{(1-\gamma)^3}{20 \cdot p^2 \cdot A^{2/p}} \cdot \left\| \frac{\partial V^{\pi_{\tilde{\theta}_t}}(\mu)}{\partial \tilde{\theta}_t} \right\|_2^2 \qquad \left( \eta = \frac{(1-\gamma)^3}{10 \cdot p^2 \cdot A^{2/p}} \right)$$

$$\leq -\frac{(1-\gamma)^3}{20 \cdot \not{p}^2 \cdot A^{2/p}} \cdot \left[ \frac{\not{p}}{\sqrt{S}} \cdot \left\| \frac{d_\mu^{\pi^*}}{d_\mu^{\pi_{\tilde{\theta}_t}}} \right\|_\infty^{-1} \cdot \frac{\min_s \pi_{\tilde{\theta}_t}(a^*(s)|s)^{1-1/p}}{\max_s \|\tilde{\theta}_t(s,\cdot)\|_p} \cdot [V^*(\mu) - V^{\pi_{\tilde{\theta}_t}}(\mu)] \right]^2 \qquad \text{(Lemma 7)}$$

$$= -\frac{(1-\gamma)^3}{20 \cdot A^{2/p} \cdot S} \cdot \left\| \frac{d_\mu^{\pi^*}}{d_\mu^{\pi_{\tilde{\theta}_t}}} \right\|_\infty^{-2} \cdot \min_s \pi_{\tilde{\theta}_t}(a^*(s)|s)^{2-2/p} \cdot [V^*(\mu) - V^{\pi_{\tilde{\theta}_t}}(\mu)]^2 \qquad \left( \|\tilde{\theta}_t(s,\cdot)\|_p = 1, \text{ for all } s \right)$$

$$\leq -\frac{(1-\gamma)^5}{20 \cdot A^{2/p} \cdot S} \cdot \left\| \frac{d_\mu^{\pi^*}}{\mu} \right\|_\infty^{-2} \cdot \min_s \pi_{\tilde{\theta}_t}(a^*(s)|s)^{2-2/p} \cdot [V^*(\mu) - V^{\pi_{\tilde{\theta}_t}}(\mu)]^2, \qquad (79)$$

where the last inequality is by Eq. (72). Then we have,

$$V^{\pi_{\theta_t}}(\mu) - V^{\pi_{\theta_{t+1}}}(\mu) = V^{\pi_{\tilde{\theta}_t}}(\mu) - V^{\pi_{\tilde{\theta}_{t+1}}}(\mu) \qquad \text{(by Lemma 9)}$$

$$\leq -\frac{(1-\gamma)^5}{20 \cdot A^{2/p} \cdot S} \cdot \left\| \frac{d_\mu^{\pi^*}}{\mu} \right\|_\infty^{-2} \cdot \min_s \pi_{\theta_t}(a^*(s)|s)^{2-2/p} \cdot [V^*(\mu) - V^{\pi_{\theta_t}}(\mu)]^2, \qquad \text{(by Eq. (79) and Lemma 9)}$$

$$\leq -\frac{(1-\gamma)^5}{20 \cdot A^{2/p} \cdot S} \cdot \left\| \frac{d_\mu^{\pi^*}}{\mu} \right\|_\infty^{-2} \cdot c^{2-2/p} \cdot [V^*(\mu) - V^{\pi_{\theta_t}}(\mu)]^2,$$

which is equivalent to,

$$\delta_{t+1} - \delta_t \leq -\frac{(1-\gamma)^5}{20 \cdot A^{2/p} \cdot S} \cdot \left\| \frac{d_\mu^{\pi^*}}{\mu} \right\|_\infty^{-2} \cdot c^{2-2/p} \cdot \delta_t^2,$$

where $\delta_t = V^*(\mu) - V^{\pi_{\theta_t}}(\mu)$. Using the similar induction argument as in Eq. (36), we have,

$$V^*(\mu) - V^{\pi_{\theta_t}}(\mu) \leq \frac{20 \cdot A^{2/p} \cdot S}{(1-\gamma)^5 \cdot t} \cdot \frac{1}{c^{2-2/p}} \cdot \left\| \frac{d_\mu^{\pi^*}}{\mu} \right\|_\infty^2,$$

which leads to the final result,

$$V^*(\rho) - V^{\pi_{\theta_t}}(\rho) \leq \frac{1}{1-\gamma} \cdot \left\| \frac{1}{\mu} \right\|_\infty \cdot [V^*(\mu) - V^{\pi_{\theta_t}}(\mu)] \qquad \text{(by Eq. (73))}$$

$$\leq \frac{20 \cdot A^{2/p} \cdot S}{c^{2-2/p} \cdot (1-\gamma)^6 \cdot t} \cdot \left\| \frac{d_\mu^{\pi^*}}{\mu} \right\|_\infty^2 \cdot \left\| \frac{1}{\mu} \right\|_\infty. \qquad \square$$

**Lemma 10** (Value sub-optimality lemma [14]). *For any policy $\pi$,*

$$V^*(\rho) - V^\pi(\rho) = \frac{1}{1-\gamma} \sum_s d_\rho^\pi(s) \sum_a (\pi^*(a|s) - \pi(a|s)) \cdot Q^*(s,a).$$

*Proof.* See the proof in [14, Lemma 21]. $\square$

## C.4  Entropy Regularized MDPs

The objective for the entropy regularized policy gradient method is,

$$\tilde{V}^\pi(\rho) := V^\pi(\rho) + \tau \cdot \mathbb{H}(\rho, \pi), \tag{80}$$

where $\mathbb{H}(\rho, \pi)$ is the "discounted entropy", defined as

$$\mathbb{H}(\rho, \pi) := \mathbb{E}_{\substack{s_0 \sim \rho, a_t \sim \pi(\cdot|s_t), \\ s_{t+1} \sim \mathcal{P}(\cdot|s_t, a_t)}} \left[ \sum_{t=0}^\infty -\gamma^t \log \pi(a_t|s_t) \right], \tag{81}$$

and $\tau \geq 0$ is "temperature" of the regularization.

**Lemma 11.** *The entropy regularized escort policy gradient w.r.t. $\theta$ is*

$$\frac{\partial \tilde{V}^{\pi_\theta}(\mu)}{\partial \theta(s, a)} = \frac{1}{1-\gamma} \cdot d_\mu^{\pi_\theta}(s) \cdot p \cdot \frac{\pi_\theta(a|s)}{\theta(s, a)} \cdot \tilde{A}^{\pi_\theta}(s, a),$$

$$\frac{\partial \tilde{V}^{\pi_\theta}(\mu)}{\partial \theta(s, \cdot)} = \frac{1}{1-\gamma} \cdot d_\mu^{\pi_\theta}(s) \cdot p \cdot diag\left( \frac{\pi_\theta(\cdot|s)}{\theta(s, \cdot)} \right) \left( \mathbf{Id} - \mathbf{1}\pi_\theta(\cdot|s)^\top \right) \left[ \tilde{Q}^{\pi_\theta}(s, \cdot) - \tau \log \pi_\theta(\cdot|s) \right].$$

*where $\tilde{A}^{\pi_\theta}(s, a)$ is the soft advantage function defined as*

$$\tilde{A}^{\pi_\theta}(s, a) = \tilde{Q}^{\pi_\theta}(s, a) - \tau \log \pi_\theta(a|s) - \tilde{V}^{\pi_\theta}(s)$$

$$\tilde{Q}^{\pi_\theta}(s, a) = r(s, a) + \gamma \sum_{s'} \mathcal{P}(s'|s, a) \tilde{V}^{\pi_\theta}(s').$$

*Proof.* According to the definition of $\tilde{V}^{\pi_\theta}$,

$$\tilde{V}^{\pi_\theta}(\mu) = \mathbb{E}_{s \sim \mu} \sum_a \pi_\theta(a|s) \cdot \left[ \tilde{Q}^{\pi_\theta}(s, a) - \tau \log \pi_\theta(a|s) \right].$$

Taking derivative w.r.t. $\theta$,

$$\frac{\partial \tilde{V}^{\pi_\theta}(\mu)}{\partial \theta} = \mathbb{E}_{s \sim \mu} \sum_a \frac{\partial \pi_\theta(a|s)}{\partial \theta} \cdot \left[ \tilde{Q}^{\pi_\theta}(s, a) - \tau \log \pi_\theta(a|s) \right]$$

$$+ \mathbb{E}_{s \sim \mu} \sum_a \pi_\theta(a|s) \cdot \left[ \frac{\partial \tilde{Q}^{\pi_\theta}(s, a)}{\partial \theta} - \tau \frac{1}{\pi_\theta(a|s)} \frac{\partial \pi_\theta(a|s)}{\partial \theta} \right]$$

$$= \mathbb{E}_{s \sim \mu} \sum_a \frac{\partial \pi_\theta(a|s)}{\partial \theta} \cdot \left[ \tilde{Q}^{\pi_\theta}(s, a) - \tau \log \pi_\theta(a|s) \right] + \mathbb{E}_{s \sim \mu} \sum_a \pi_\theta(a|s) \cdot \frac{\partial \tilde{Q}^{\pi_\theta}(s, a)}{\partial \theta}$$

$$= \mathbb{E}_{s \sim \mu} \sum_a \frac{\partial \pi_\theta(a|s)}{\partial \theta} \cdot \left[ \tilde{Q}^{\pi_\theta}(s, a) - \tau \log \pi_\theta(a|s) \right]$$

$$+ \gamma \cdot \mathbb{E}_{s \sim \mu} \sum_a \pi_\theta(a|s) \sum_{s'} \mathcal{P}(s'|s, a) \cdot \frac{\partial \tilde{V}^{\pi_\theta}(s')}{\partial \theta}$$

$$= \frac{1}{1-\gamma} \sum_s d_\mu^{\pi_\theta}(s) \sum_a \frac{\partial \pi_\theta(a|s)}{\partial \theta} \cdot \left[ \tilde{Q}^{\pi_\theta}(s, a) - \tau \log \pi_\theta(a|s) \right],$$

where the second equation is because of

$$\sum_a \pi_\theta(a|s) \cdot \left[ \frac{1}{\pi_\theta(a|s)} \frac{\partial \pi_\theta(a|s)}{\partial \theta} \right] = \sum_a \frac{\partial \pi_\theta(a|s)}{\partial \theta} = \frac{\partial}{\partial \theta} \sum_a \pi_\theta(a|s) = \frac{\partial 1}{\partial \theta} = 0.$$

Using similar arguments as in the proof for Lemma 5, i.e., for $s' \neq s$, $\frac{\partial \pi_\theta(a|s)}{\partial \theta(s', \cdot)} = \mathbf{0}$,

$$\frac{\partial \tilde{V}^{\pi_\theta}(\mu)}{\partial \theta(s, \cdot)} = \frac{1}{1-\gamma} \cdot d_\mu^{\pi_\theta}(s) \cdot \left[ \sum_a \frac{\partial \pi_\theta(a|s)}{\partial \theta(s, \cdot)} \cdot \left[ \tilde{Q}^{\pi_\theta}(s, a) - \tau \log \pi_\theta(a|s) \right] \right]$$

$$= \frac{1}{1-\gamma} \cdot d_\mu^{\pi_\theta}(s) \cdot \left( \frac{d\pi_\theta(\cdot|s)}{d\theta(s, \cdot)} \right)^\top \left[ \tilde{Q}^{\pi_\theta}(s, \cdot) - \tau \log \pi_\theta(\cdot|s) \right]$$

$$= \frac{1}{1-\gamma} \cdot d_\mu^{\pi_\theta}(s) \cdot p \cdot \mathrm{diag}\left( \frac{\pi_\theta(\cdot|s)}{\theta(s, \cdot)} \right) \left( \mathbf{Id} - \mathbf{1}\pi_\theta(\cdot|s)^\top \right) \left[ \tilde{Q}^{\pi_\theta}(s, \cdot) - \tau \log \pi_\theta(\cdot|s) \right].$$

For each component $a$, we have

$$\frac{\partial \tilde{V}^{\pi_\theta}(\mu)}{\partial \theta(s,a)} = \frac{1}{1-\gamma} \cdot d_\mu^{\pi_\theta}(s) \cdot p \cdot \frac{\pi_\theta(a|s)}{\theta(s,a)} \cdot \Big[ \tilde{Q}^{\pi_\theta}(s,a) - \tau \log \pi_\theta(a|s)$$

$$- \sum_a \pi_\theta(a|s) \cdot \Big[ \tilde{Q}^{\pi_\theta}(s,a) - \tau \log \pi_\theta(a|s) \Big] \Big]$$

$$= \frac{1}{1-\gamma} \cdot d_\mu^{\pi_\theta}(s) \cdot p \cdot \frac{\pi_\theta(a|s)}{\theta(s,a)} \cdot \Big[ \tilde{Q}^{\pi_\theta}(s,a) - \tau \log \pi_\theta(a|s) - \tilde{V}^{\pi_\theta}(s) \Big]$$

$$= \frac{1}{1-\gamma} \cdot d_\mu^{\pi_\theta}(s) \cdot p \cdot \frac{\pi_\theta(a|s)}{\theta(s,a)} \cdot \tilde{A}^{\pi_\theta}(s,a). \qquad \square$$

**Lemma 12** (Non-uniform Łojasiewicz)**.** *Suppose* $\mu(s) > 0$ *for all* $s \in \mathcal{S}$ *and* $\pi_\theta = f_p(\theta)$. *Then,*

$$\left\| \frac{\partial \tilde{V}^{\pi_\theta}(\mu)}{\partial \theta} \right\|_2 \geq \frac{\sqrt{2\tau}}{\sqrt{S}} \cdot \min_s \sqrt{\mu(s)} \cdot \frac{p}{\max_s \|\theta(s,\cdot)\|_p} \cdot \min_{s,a} \pi_\theta(a|s)^{1-1/p} \cdot \left\| \frac{d_\rho^{\pi_\tau^*}}{d_\mu^{\pi_\theta}} \right\|_\infty^{-\frac{1}{2}} \cdot \Big[ \tilde{V}^{\pi_\tau^*}(\rho) - \tilde{V}^{\pi_\theta}(\rho) \Big]^{\frac{1}{2}}.$$

*Proof.* According to the definition of soft value functions,

$$\tilde{V}^{\pi_\tau^*}(\rho) - \tilde{V}^{\pi_\theta}(\rho) = \mathop{\mathbb{E}}_{\substack{s_0 \sim \rho, a_t \sim \pi_\tau^*(\cdot|s_t), \\ s_{t+1} \sim \mathcal{P}(\cdot|s_t,a_t)}} \left[ \sum_{t=0}^\infty \gamma^t (r(s_t,a_t) - \tau \log \pi_\tau^*(a_t|s_t)) \right] - \tilde{V}^{\pi_\theta}(\rho)$$

$$= \mathop{\mathbb{E}}_{\substack{s_0 \sim \rho, a_t \sim \pi_\tau^*(\cdot|s_t), \\ s_{t+1} \sim \mathcal{P}(\cdot|s_t,a_t)}} \left[ \sum_{t=0}^\infty \gamma^t (r(s_t,a_t) - \tau \log \pi_\tau^*(a_t|s_t) + \tilde{V}^{\pi_\theta}(s_t) - \tilde{V}^{\pi_\theta}(s_t)) \right] - \tilde{V}^{\pi_\theta}(\rho)$$

$$= \mathop{\mathbb{E}}_{\substack{s_0 \sim \rho, a_t \sim \pi_\tau^*(\cdot|s_t), \\ s_{t+1} \sim \mathcal{P}(\cdot|s_t,a_t)}} \left[ \sum_{t=0}^\infty \gamma^t (r(s_t,a_t) - \tau \log \pi_\tau^*(a_t|s_t) + \gamma \tilde{V}^{\pi_\theta}(s_{t+1}) - \tilde{V}^{\pi_\theta}(s_t)) \right]$$

$$= \frac{1}{1-\gamma} \sum_s d_\rho^{\pi_\tau^*}(s) \cdot \left[ \sum_a \pi_\tau^*(a|s) \cdot \left( r(s,a) - \tau \log \pi_\tau^*(a|s) + \gamma \sum_{s'} \mathcal{P}(s'|s,a) \tilde{V}^{\pi_\theta}(s') - \tilde{V}^{\pi_\theta}(s) \right) \right]$$

$$= \frac{1}{1-\gamma} \sum_s d_\rho^{\pi_\tau^*}(s) \cdot \left[ \sum_a \pi_\tau^*(a|s) \cdot \Big[ \tilde{Q}^{\pi_\theta}(s,a) - \tau \log \pi_\tau^*(a|s) \Big] - \tilde{V}^{\pi_\theta}(s) \right]. \qquad (82)$$

Next, define the "soft greedy policy" $\bar{\pi}_\theta(\cdot|s) = \text{softmax}(\tilde{Q}^{\pi_\theta}(s,\cdot)/\tau), \forall s$, i.e.,

$$\bar{\pi}_\theta(a|s) = \frac{\exp\left\{ \tilde{Q}^{\pi_\theta}(s,a)/\tau \right\}}{\sum_{a'} \exp\left\{ \tilde{Q}^{\pi_\theta}(s,a')/\tau \right\}}, \quad \forall a. \qquad (83)$$

We have, $\forall s$,

$$\sum_a \pi_\tau^*(a|s) \cdot \Big[ \tilde{Q}^{\pi_\theta}(s,a) - \tau \log \pi_\tau^*(a|s) \Big] \leq \max_{\pi(\cdot|s)} \sum_a \pi(a|s) \cdot \Big[ \tilde{Q}^{\pi_\theta}(s,a) - \tau \log \pi(a|s) \Big]$$

$$= \sum_a \bar{\pi}_\theta(a|s) \cdot \Big[ \tilde{Q}^{\pi_\theta}(s,a) - \tau \log \bar{\pi}_\theta(a|s) \Big]$$

$$= \tau \log \sum_a \exp\left\{ \tilde{Q}^{\pi_\theta}(s,a)/\tau \right\}. \qquad (84)$$

Also note that,

$$
\begin{aligned}
\tilde{V}^{\pi_\theta}(s) &= \sum_a \pi_\theta(a|s) \cdot \left[ \tilde{Q}^{\pi_\theta}(s,a) - \tau \log \pi_\theta(a|s) \right] \\
&= \sum_a \pi_\theta(a|s) \cdot \left[ \tilde{Q}^{\pi_\theta}(s,a) - \tau \log \bar{\pi}_\theta(a|s) + \tau \log \bar{\pi}_\theta(a|s) - \tau \log \pi_\theta(a|s) \right] \\
&= \sum_a \pi_\theta(a|s) \cdot \left[ \tilde{Q}^{\pi_\theta}(s,a) - \tau \log \bar{\pi}_\theta(a|s) \right] - \tau D_{\mathrm{KL}}(\pi_\theta(\cdot|s)\|\bar{\pi}_\theta(\cdot|s)) \\
&= \tau \log \sum_a \exp\left\{ \tilde{Q}^{\pi_\theta}(s,a)/\tau \right\} - \tau \cdot D_{\mathrm{KL}}(\pi_\theta(\cdot|s)\|\bar{\pi}_\theta(\cdot|s)). \tag{85}
\end{aligned}
$$

Combining Eq. (82), Eqs. (84) and (85), we have,

$$
\begin{aligned}
\tilde{V}^{\pi_\tau^*}(\rho) - \tilde{V}^{\pi_\theta}(\rho) &= \frac{1}{1-\gamma} \sum_s d_\rho^{\pi_\tau^*}(s) \cdot \left[ \sum_a \pi_\tau^*(a|s) \cdot \left[ \tilde{Q}^{\pi_\theta}(s,a) - \tau \log \pi_\tau^*(a|s) \right] - \tilde{V}^{\pi_\theta}(s) \right] \\
&\leq \frac{1}{1-\gamma} \sum_s d_\rho^{\pi_\tau^*}(s) \cdot \left[ \tau \log \sum_a \exp\left\{ \tilde{Q}^{\pi_\theta}(s,a)/\tau \right\} - \tilde{V}^{\pi_\theta}(s) \right] \\
&= \frac{1}{1-\gamma} \sum_s d_\rho^{\pi_\tau^*}(s) \cdot \tau \cdot D_{\mathrm{KL}}(\pi_\theta(\cdot|s)\|\bar{\pi}_\theta(\cdot|s)) \\
&\leq \frac{1}{1-\gamma} \sum_s d_\rho^{\pi_\tau^*}(s) \cdot \frac{\tau}{2} \cdot \left\| \frac{\tilde{Q}^{\pi_\theta}(s,\cdot)}{\tau} - \log \pi_\theta(s,\cdot) - \frac{c_\theta(s)}{\tau} \cdot \mathbf{1} \right\|_\infty^2 \quad \text{(by Lemma 13)} \\
&= \frac{1}{1-\gamma} \sum_s d_\rho^{\pi_\tau^*}(s) \cdot \frac{1}{2\tau} \cdot \left\| \tilde{Q}^{\pi_\theta}(s,\cdot) - \tau \log \pi_\theta(s,\cdot) - c_\theta(s) \cdot \mathbf{1} \right\|_\infty^2, \tag{86}
\end{aligned}
$$

where $c_\theta(s) = \frac{\left( \tilde{Q}^{\pi_\theta}(s,\cdot) - \tau \log \pi_\theta(s,\cdot) \right)^\top \mathbf{1}}{A}$. Taking square root of Eq. (86), we have,

$$
\begin{aligned}
\left[ \tilde{V}^{\pi_\tau^*}(\rho) - \tilde{V}^{\pi_\theta}(\rho) \right]^{\frac{1}{2}} &\leq \frac{1}{\sqrt{1-\gamma}} \cdot \left[ \sum_s d_\rho^{\pi_\tau^*}(s) \cdot \frac{1}{2\tau} \cdot \left\| \tilde{Q}^{\pi_\theta}(s,\cdot) - \tau \log \pi_\theta(s,\cdot) - c_\theta(s) \cdot \mathbf{1} \right\|_\infty^2 \right]^{\frac{1}{2}} \\
&= \frac{1}{\sqrt{1-\gamma}} \cdot \left[ \sum_s \left( \sqrt{d_\rho^{\pi_\tau^*}(s)} \cdot \frac{1}{\sqrt{2\tau}} \cdot \left\| \tilde{Q}^{\pi_\theta}(s,\cdot) - \tau \log \pi_\theta(s,\cdot) - c_\theta(s) \cdot \mathbf{1} \right\|_\infty \right)^2 \right]^{\frac{1}{2}} \\
&\leq \frac{1}{\sqrt{1-\gamma}} \cdot \sum_s \sqrt{d_\rho^{\pi_\tau^*}(s)} \cdot \frac{1}{\sqrt{2\tau}} \cdot \left\| \tilde{Q}^{\pi_\theta}(s,\cdot) - \tau \log \pi_\theta(s,\cdot) - c_\theta(s) \cdot \mathbf{1} \right\|_\infty \quad (\|x\|_2 \leq \|x\|_1) \\
&\leq \frac{1}{\sqrt{1-\gamma}} \cdot \frac{1}{\sqrt{2\tau}} \cdot \left\| \frac{d_\rho^{\pi_\tau^*}}{d_\mu^{\pi_\theta}} \right\|_\infty^{\frac{1}{2}} \sum_s \sqrt{d_\mu^{\pi_\theta}(s)} \cdot \left\| \tilde{Q}^{\pi_\theta}(s,\cdot) - \tau \log \pi_\theta(s,\cdot) - c_\theta(s) \cdot \mathbf{1} \right\|_\infty.
\end{aligned}
$$
$$\tag{87}$$

On the other hand, the entropy regularized policy gradient norm is lower bounded as

$$\left\|\frac{\partial \tilde{V}^{\pi_\theta}(\mu)}{\partial \theta}\right\|_2 = \left[\sum_s \left\|\frac{\partial \tilde{V}^{\pi_\theta}(\mu)}{\partial \theta(s,\cdot)}\right\|_2^2\right]^{\frac{1}{2}}$$

$$\geq \frac{1}{\sqrt{S}}\sum_s \left\|\frac{\partial \tilde{V}^{\pi_\theta}(\mu)}{\partial \theta(s,\cdot)}\right\|_2, \qquad \text{(by Cauchy-Schwarz , } \|x\|_1 = |\langle \mathbf{1},\ |x|\rangle| \leq \|\mathbf{1}\|_2 \cdot \|x\|_2)$$

$$\geq \frac{1}{\sqrt{S}}\cdot\frac{1}{1-\gamma}\sum_s d_\mu^{\pi_\theta}(s)$$

$$\cdot\left\|p\cdot\text{diag}\left(\frac{\pi_\theta(\cdot|s)}{\theta(s,\cdot)}\right)\left(\mathbf{Id}-\mathbf{1}\pi_\theta(\cdot|s)^\top\right)\left[\tilde{Q}^{\pi_\theta}(s,\cdot)-\tau\log\pi_\theta(s,\cdot)-c_\theta(s)\cdot\mathbf{1}\right]\right\|_2 \qquad \text{(by Lemma 11)}$$

$$=\frac{1}{\sqrt{S}}\cdot\frac{1}{1-\gamma}\sum_s d_\mu^{\pi_\theta}(s)\cdot\frac{p}{\|\theta(s,\cdot)\|_p}$$

$$\cdot\left\|\text{diag}\left(\pi_\theta(\cdot|s)^{1-1/p}\right)\left(\mathbf{Id}-\mathbf{1}\pi_\theta(\cdot|s)^\top\right)\left[\tilde{Q}^{\pi_\theta}(s,\cdot)-\tau\log\pi_\theta(s,\cdot)-c_\theta(s)\cdot\mathbf{1}\right]\right\|_2$$

$$\geq\frac{1}{\sqrt{S}}\cdot\frac{1}{1-\gamma}\sum_s d_\mu^{\pi_\theta}(s)\cdot\frac{p}{\|\theta(s,\cdot)\|_p}\cdot\min_a\pi_\theta(a|s)^{1-1/p}\cdot\left\|\tilde{Q}^{\pi_\theta}(s,\cdot)-\tau\log\pi_\theta(s,\cdot)-c_\theta(s)\cdot\mathbf{1}\right\|_2. \text{ (by Lemma 14)}$$

Denote $\zeta_\theta(s)=\tilde{Q}^{\pi_\theta}(s,\cdot)-\tau\log\pi_\theta(s,\cdot)-c_\theta(s)\cdot\mathbf{1}$. We have,

$$\left\|\frac{\partial \tilde{V}^{\pi_\theta}(\mu)}{\partial\theta}\right\|_2 \geq \frac{1}{\sqrt{S}}\cdot\frac{1}{1-\gamma}\sum_s d_\mu^{\pi_\theta}(s)\cdot\frac{p}{\|\theta(s,\cdot)\|_p}\cdot\min_a\pi_\theta(a|s)^{1-1/p}\cdot\|\zeta_\theta(s)\|_2$$

$$\geq\frac{1}{\sqrt{S}}\cdot\frac{1}{1-\gamma}\sum_s d_\mu^{\pi_\theta}(s)\cdot\frac{p}{\|\theta(s,\cdot)\|_p}\cdot\min_a\pi_\theta(a|s)^{1-1/p}\cdot\|\zeta_\theta(s)\|_\infty$$

$$\geq\frac{1}{\sqrt{S}}\cdot\frac{1}{\sqrt{1-\gamma}}\cdot\min_s\sqrt{d_\mu^{\pi_\theta}(s)}\cdot\min_s\frac{p}{\|\theta(s,\cdot)\|_p}\cdot\min_{s,a}\pi_\theta(a|s)^{1-1/p}\cdot\sqrt{2\tau}\cdot\left\|\frac{d_\rho^{\pi_\tau^*}}{d_\mu^{\pi_\theta}}\right\|_\infty^{-\frac{1}{2}}$$

$$\cdot\left[\frac{1}{\sqrt{1-\gamma}}\cdot\frac{1}{\sqrt{2\tau}}\cdot\left\|\frac{d_\rho^{\pi_\tau^*}}{d_\mu^{\pi_\theta}}\right\|_\infty^{\frac{1}{2}}\sum_s\sqrt{d_\mu^{\pi_\theta}(s)}\cdot\|\zeta_\theta(s)\|_\infty\right]$$

$$\geq\frac{1}{\sqrt{S}}\cdot\frac{1}{\sqrt{1-\gamma}}\cdot\min_s\sqrt{d_\mu^{\pi_\theta}(s)}\cdot\min_s\frac{p}{\|\theta(s,\cdot)\|_p}\cdot\min_{s,a}\pi_\theta(a|s)^{1-1/p}\cdot\sqrt{2\tau}\cdot\left\|\frac{d_\rho^{\pi_\tau^*}}{d_\mu^{\pi_\theta}}\right\|_\infty^{-\frac{1}{2}}$$

$$\cdot\left[\tilde{V}^{\pi_\tau^*}(\rho)-\tilde{V}^{\pi_\theta}(\rho)\right]^{\frac{1}{2}} \qquad \text{(by Eq. (87))}$$

$$\geq\frac{\sqrt{2\tau}}{\sqrt{S}}\cdot\min_s\sqrt{\mu(s)}\cdot\min_s\frac{p}{\|\theta(s,\cdot)\|_p}\cdot\min_{s,a}\pi_\theta(a|s)^{1-1/p}\cdot\left\|\frac{d_\rho^{\pi_\tau^*}}{d_\mu^{\pi_\theta}}\right\|_\infty^{-\frac{1}{2}}\cdot\left[\tilde{V}^{\pi_\tau^*}(\rho)-\tilde{V}^{\pi_\theta}(\rho)\right]^{\frac{1}{2}},$$

where the last inequality is by $d_\mu^{\pi_\theta}(s)\geq(1-\gamma)\cdot\mu(s)$ (cf. Eq. (72)). Note that $\min_{s,a}\pi_\theta(a|s)^{1-1/p}\geq\min_{s,a}\pi_\theta(a|s)$, which is a better dependence than [14, Lemma 15]. $\square$

**Lemma 13** (KL-Logit inequality [14]). *Let $\pi_\theta=\text{softmax}(\theta)$ and $\pi_{\theta'}=\text{softmax}(\theta')$. Then for any constant $c\in\mathbb{R}$,*

$$D_{\text{KL}}(\pi_\theta\|\pi_{\theta'})\leq\frac{1}{2}\cdot\|\theta'-\theta-c\cdot\mathbf{1}\|_\infty^2.$$

*Proof.* See the proof in [14, Lemma 27]. $\square$

**Lemma 14.** *Let $\pi\in\Delta(\mathcal{A})$ and $q\geq0$. For any vector $x\in\mathbb{R}^K$, we have,*

$$\left\|\text{diag}(q)\left(\mathbf{Id}-\mathbf{1}\pi^\top\right)\left(x-\frac{x^\top\mathbf{1}}{K}\cdot\mathbf{1}\right)\right\|_2\geq\min_a q(a)\cdot\left\|x-\frac{x^\top\mathbf{1}}{K}\cdot\mathbf{1}\right\|_2.$$

*Proof.* Denote $G = G(\pi, q) = \mathrm{diag}(q)\left(\mathbf{Id} - \mathbf{1}\pi^\top\right) \in \mathbb{R}^{K \times K}$. Denote the eigenvalues of $G^\top G$ as

$$\lambda_1 \le \lambda_2 \le \cdots \le \lambda_K.$$

First, we show that $\lambda_1 = 0$.

$$\begin{aligned}
G^\top G \mathbf{1} &= G^\top \mathrm{diag}(q)\left(\mathbf{Id} - \mathbf{1}\pi^\top\right)\mathbf{1} \\
&= G^\top \mathrm{diag}(q)\left(\mathbf{1} - \mathbf{1}\right) = 0 \cdot \mathbf{1},
\end{aligned}$$

which means $\mathbf{1}$ is an eigenvector of $G^\top G$ with eigenvalue 0. And for any vector $x \in \mathbb{R}^K$, we have,

$$x^\top G^\top G x = \|Gx\|_2^2 \ge 0,$$

which means $G^\top G$ is semi-positive definite. Therefore $\lambda_1 = 0$.

Second, for any vector $x \in \mathbb{R}^K$, $x$ can be written as linear combination of eigenvectors of $G^\top G$,

$$\begin{aligned}
x &= a_1 \cdot \frac{\mathbf{1}}{\sqrt{K}} + a_2 \cdot v_2 + \cdots + a_K \cdot v_K \\
&= \frac{x^\top \mathbf{1}}{K} \cdot \mathbf{1} + a_2 \cdot v_2 + \cdots + a_K \cdot v_K.
\end{aligned}$$

Since $G^\top G$ is symmetric, $\left\{\frac{1}{\sqrt{K}}, v_2, \ldots, v_K\right\}$ are orthonormal. The last equation is because the representation is unique, and

$$a_1 = x^\top \frac{\mathbf{1}}{\sqrt{K}} = \frac{x^\top \mathbf{1}}{\sqrt{K}}.$$

Denote

$$x' = x - \frac{x^\top \mathbf{1}}{K} \cdot \mathbf{1} = a_2 \cdot v_2 + \cdots + a_K \cdot v_K.$$

We have,

$$\|x'\|_2^2 = a_2^2 + \cdots + a_K^2. \tag{88}$$

Since $v_2, \ldots, v_K$ are eigenvectors of $G^\top G$,

$$G^\top G x' = a_2 \cdot \lambda_2 \cdot v_2 + \cdots + a_K \cdot \lambda_K \cdot v_K.$$

Therefore we have,

$$\begin{aligned}
\|Gx'\|_2 = \left(\|Gx'\|_2^2\right)^{\frac{1}{2}} &= \left(x'^\top G^\top G x'\right)^{\frac{1}{2}} \\
&= \left(a_2^2 \cdot \lambda_2 + \cdots + a_K^2 \cdot \lambda_K\right)^{\frac{1}{2}} \\
&\ge \left(a_2^2 \cdot \lambda_2 + \cdots + a_K^2 \cdot \lambda_2\right)^{\frac{1}{2}} \\
&= \sqrt{\lambda_2} \cdot \|x'\|_2. \qquad \text{(by Eq. (88))} \tag{89}
\end{aligned}$$

Next, we have,

$$\begin{aligned}
\lambda_2 = \frac{v_2^\top G^\top G v_2}{v_2^\top v_2} &= \frac{1}{v_2^\top v_2} \cdot \|Gv_2\|_2^2 = \frac{1}{v_2^\top v_2} \cdot \left\|\mathrm{diag}(q)\left(v_2 - \pi^\top v_2 \cdot \mathbf{1}\right)\right\|_2^2 \\
&= \frac{1}{v_2^\top v_2} \cdot \left[\sum_{a=1}^K q(a)^2 \cdot \left(v_2(a) - \pi^\top v_2\right)^2\right] \\
&\ge \frac{1}{v_2^\top v_2} \cdot \min_a q(a)^2 \cdot \left\|v_2 - \pi^\top v_2 \cdot \mathbf{1}\right\|_2^2 \\
&= \min_a q(a)^2 \cdot \frac{v_2^\top v_2 + K \cdot (\pi^\top v_2)^2}{v_2^\top v_2} \qquad \left(v_2^\top \mathbf{1} = 0\right) \\
&\ge \min_a q(a)^2. \tag{90}
\end{aligned}$$

Combining Eqs. (89) and (90), we have,

$$\left\| \text{diag}(q)\left(\mathbf{Id} - \mathbf{1}\pi^\top\right)\left(x - \frac{x^\top\mathbf{1}}{K}\cdot\mathbf{1}\right)\right\|_2 = \|Gx'\|_2$$

$$\geq \sqrt{\lambda_2}\cdot\|x'\|_2$$

$$\geq \min_a q(a)\cdot\left\|x - \frac{x^\top\mathbf{1}}{K}\cdot\mathbf{1}\right\|_2. \qquad\square$$

**Theorem 4.** For an entropy regularized MDP with finite states and actions, following the escort policy gradient with any initialization such that $|\theta_1(s,a)| > 0$, $\forall(s,a)$, and $\eta_t = (1-\gamma)^3/(10\cdot p^2\cdot A^{1/p}+c_\tau)$ to get $\{\theta_t\}_{t\geq 1}$, for all $t \geq 1$, the following sub-optimality upper bounds hold for $\pi_{\theta_t}$:

$$\text{for } p \geq 2, \qquad \tilde{V}^{\pi_\tau^*}(\rho) - \tilde{V}^{\pi_{\theta_t}}(\rho) \leq \frac{\|1/\mu\|_\infty}{\exp\{C_\tau\cdot c'^2\cdot t\}}\cdot\frac{1 + \tau\log A}{(1-\gamma)^2}, \qquad (91)$$

where $c' > c := \inf_{(s,a)}\inf_{t\geq 1}\pi_{\theta_t}(a|s) > 0$, $\tau$ is the temperature for entropy regularization, $\pi_\tau^*$ is the softmax optimal policy, and $c_\tau$, $C_\tau$ are problem-dependent constants.

*Proof.* According to the soft sub-optimality lemma of Lemma 15,

$$\tilde{V}^{\pi_\tau^*}(\rho) - \tilde{V}^{\pi_{\theta_t}}(\rho) = \frac{1}{1-\gamma}\sum_s\left[d_\rho^{\pi_{\theta_t}}(s)\cdot\tau\cdot D_{\text{KL}}(\pi_{\theta_t}(\cdot|s)\|\pi_\tau^*(\cdot|s))\right]$$

$$= \frac{1}{1-\gamma}\sum_s\frac{d_\rho^{\pi_{\theta_t}}(s)}{d_\mu^{\pi_{\theta_t}}(s)}\cdot\left[d_\mu^{\pi_{\theta_t}}(s)\cdot\tau\cdot D_{\text{KL}}(\pi_{\theta_t}(\cdot|s)\|\pi_\tau^*(\cdot|s))\right]$$

$$\leq \frac{1}{(1-\gamma)^2}\sum_s\frac{1}{\mu(s)}\cdot\left[d_\mu^{\pi_{\theta_t}}(s)\cdot\tau\cdot D_{\text{KL}}(\pi_{\theta_t}(\cdot|s)\|\pi_\tau^*(\cdot|s))\right]$$

$$\leq \frac{1}{(1-\gamma)^2}\cdot\left\|\frac{1}{\mu}\right\|_\infty\sum_s\left[d_\mu^{\pi_{\theta_t}}(s)\cdot\tau\cdot D_{\text{KL}}(\pi_{\theta_t}(\cdot|s)\|\pi_\tau^*(\cdot|s))\right]$$

$$= \frac{1}{1-\gamma}\cdot\left\|\frac{1}{\mu}\right\|_\infty\cdot\left[\tilde{V}^{\pi_\tau^*}(\mu) - \tilde{V}^{\pi_{\theta_t}}(\mu)\right], \qquad (92)$$

where the last equation is again by Lemma 15, and the first inequality is according to $d_\mu^{\pi_{\theta_t}}(s) \geq (1-\gamma)\cdot\mu(s)$ (cf. Eq. (72)).

According to Lemma 9, using $\frac{\partial\tilde{V}^{\pi_{\theta_t}}(\mu)}{\partial\theta_t}$ in Algorithm 1 with learning rate $\eta_t(s) = \eta\cdot\|\theta_t(s,\cdot)\|_p^2$ is equivalent to using $\frac{\partial\tilde{V}^{\pi_{\tilde{\theta}_t}}(\mu)}{\partial\tilde{\theta}_t}$ in Algorithm 2 with learning rate $\eta$. We have, in Algorithm 2,

$$\left|\tilde{V}^{\pi_{\tilde{\zeta}_{t+1}}}(\mu) - \tilde{V}^{\pi_{\tilde{\theta}_t}}(\mu) - \left\langle\frac{\partial\tilde{V}^{\pi_{\tilde{\theta}_t}}(\mu)}{\partial\tilde{\theta}_t}, \tilde{\zeta}_{t+1} - \tilde{\theta}_t\right\rangle\right| \leq \frac{4\cdot p^2\cdot A^{2/p} + c_\tau}{(1-\gamma)^3}\cdot\frac{\|\tilde{\zeta}_{t+1} - \tilde{\theta}_t\|_2^2}{\min_s\|\tilde{\theta}_{t,\xi}(s,\cdot)\|_p^2}, \qquad (93)$$

where

$$\tilde{\theta}_{t,\xi} := \tilde{\theta}_t + \xi\cdot(\tilde{\zeta}_{t+1} - \tilde{\theta}_t)$$

$$= \tilde{\theta}_t + \xi\cdot\eta\cdot\frac{\partial\tilde{V}^{\pi_{\tilde{\theta}_t}}(\mu)}{\partial\tilde{\theta}_t}, \qquad \text{(Algorithm 2)}$$

for some $\xi \in [0,1]$. Denote $s_\xi := \arg\min_s\|\tilde{\theta}_{t,\xi}(s,\cdot)\|_p^2$. We have,

$$\|\tilde{\theta}_{t,\xi}(s_\xi,\cdot)\|_p \geq \|\tilde{\theta}_t(s_\xi,\cdot)\|_p - \xi\cdot\eta\cdot\left\|\frac{\partial\tilde{V}^{\pi_{\tilde{\theta}_t}}(\mu)}{\partial\tilde{\theta}_t(s_\xi,\cdot)}\right\|_p \qquad \text{(by triangle inequality)}$$

$$\geq \min_s\|\tilde{\theta}_t(s,\cdot)\|_p - \xi\cdot\eta\cdot\left\|\frac{\partial\tilde{V}^{\pi_{\tilde{\theta}_t}}(\mu)}{\partial\tilde{\theta}_t(s_\xi,\cdot)}\right\|_p. \qquad (94)$$

The $\ell_p$ gradient norm can be upper bounded as,

$$\left\| \frac{\partial \tilde{V}^{\pi_\theta}(\mu)}{\partial \theta(s, \cdot)} \right\|_p = \left[ \sum_a \left| \frac{1}{1 - \gamma} \cdot d_\mu^{\pi_\theta}(s) \cdot p \cdot \frac{\pi_\theta(a|s)}{\theta(s, a)} \cdot \tilde{A}^{\pi_\theta}(s, a) \right|^p \right]^{\frac{1}{p}} \quad \text{(by Lemma 11)}$$

$$\leq \frac{p}{1 - \gamma} \cdot \left[ \sum_a \left| \frac{\pi_\theta(a|s)}{\theta(s, a)} \cdot \tilde{A}^{\pi_\theta}(s, a) \right|^p \right]^{\frac{1}{p}}$$

$$= \frac{p}{1 - \gamma} \cdot \frac{1}{\|\theta(s, \cdot)\|_p} \cdot \left[ \sum_a \left( \pi_\theta(a|s)^{1 - 1/p} \cdot |\tilde{A}^{\pi_\theta}(s, a)| \right)^p \right]^{\frac{1}{p}}$$

$$\leq \frac{p}{1 - \gamma} \cdot \frac{1}{\|\theta(s, \cdot)\|_p} \cdot \left[ \sum_a \left( 1 \cdot \frac{1 + \tau \log A}{1 - \gamma} \right)^p \right]^{\frac{1}{p}}$$

$$\leq \frac{p \cdot A^{1/p} \cdot (1 + \tau \log A)}{(1 - \gamma)^2} \cdot \max_s \frac{1}{\|\theta(s, \cdot)\|_p}. \tag{95}$$

Combining Eqs. (94) and (95), we have,

$$\min_s \|\tilde{\theta}_{t,\xi}(s, \cdot)\|_p \geq \min_s \|\tilde{\theta}_t(s, \cdot)\|_p - \xi \cdot \eta \cdot \frac{p \cdot A^{1/p} \cdot (1 + \tau \log A)}{(1 - \gamma)^2} \cdot \frac{1}{\min_s \|\tilde{\theta}_t(s, \cdot)\|_p}$$

$$= 1 - \xi \cdot \eta \cdot \frac{p \cdot A^{1/p} \cdot (1 + \tau \log A)}{(1 - \gamma)^2} \quad \left( \|\tilde{\theta}_t(s, \cdot)\|_p = 1, \text{ for all } s, \text{ Algorithm 2} \right) \tag{96}$$

Note that $\eta = \frac{(1-\gamma)^3}{10 \cdot p^2 \cdot A^{2/p} \cdot (1 + \tau \log A)}$. We have,

$$\min_s \|\tilde{\theta}_{t,\xi}(s, \cdot)\|_p \geq 1 - \xi \cdot \eta \cdot \frac{p \cdot A^{1/p} \cdot (1 + \tau \log A)}{(1 - \gamma)^2} \quad \text{(by Eq. (96))}$$

$$= 1 - \xi \cdot \frac{(1 - \gamma)^3}{10 \cdot p^2 \cdot A^{2/p}} \cdot \frac{p \cdot A^{1/p}}{(1 - \gamma)^2}$$

$$\geq 1 - \frac{1 - \gamma}{10 \cdot p \cdot A^{1/p}}$$

$$= \left( 1 - \frac{2}{\sqrt{5}} \right) \cdot \left( 1 - \frac{5 + 2\sqrt{5}}{10} \cdot \frac{1 - \gamma}{p \cdot A^{1/p}} \right) + \frac{2}{\sqrt{5}}$$

$$\geq \frac{2}{\sqrt{5}}. \quad \left( p \geq 2, \ A^{1/p} \geq 1, \ 1 - \gamma \in (0, 1] \right) \tag{97}$$

Combining Eqs. (93) and (97), we have,

$$\left| \tilde{V}^{\pi_{\tilde{\zeta}_{t+1}}}(\mu) - \tilde{V}^{\pi_{\tilde{\theta}_t}}(\mu) - \left\langle \frac{\partial \tilde{V}^{\pi_{\tilde{\theta}_t}}(\mu)}{\partial \tilde{\theta}_t}, \tilde{\zeta}_{t+1} - \tilde{\theta}_t \right\rangle \right| \leq \frac{4 \cdot p^2 \cdot A^{2/p} + c_\tau}{(1 - \gamma)^3} \cdot \frac{\|\tilde{\zeta}_{t+1} - \tilde{\theta}_t\|_2^2}{\min_s \|\tilde{\theta}_{t,\xi}(s, \cdot)\|_p^2}$$

$$\leq \frac{4 \cdot p^2 \cdot A^{2/p} + c_\tau}{(1 - \gamma)^3} \cdot \frac{5}{4} \cdot \|\tilde{\zeta}_{t+1} - \tilde{\theta}_t\|_2^2$$

$$= \frac{5 \cdot p^2 \cdot A^{2/p} + c_\tau}{(1 - \gamma)^3} \cdot \|\tilde{\zeta}_{t+1} - \tilde{\theta}_t\|_2^2, \tag{98}$$

which implies,

$$\tilde{V}^{\pi_{\tilde\theta_t}}(\mu) - \tilde{V}^{\pi_{\tilde\theta_{t+1}}}(\mu) = \tilde{V}^{\pi_{\tilde\theta_t}}(\mu) - \tilde{V}^{\pi_{\tilde\zeta_{t+1}}}(\mu) \qquad \left(\tilde\theta_{t+1}(s,a) = \frac{\tilde\zeta_{t+1}(s,a)}{\|\tilde\zeta_{t+1}(s,\cdot)\|_p}, \text{ Algorithm 2}\right)$$

$$\leq -\left\langle \frac{\partial \tilde{V}^{\pi_{\tilde\theta_t}}(\mu)}{\partial \tilde\theta_t}, \tilde\zeta_{t+1} - \tilde\theta_t \right\rangle + \frac{5\cdot p^2 \cdot A^{2/p} + c_\tau}{(1-\gamma)^3} \cdot \|\tilde\zeta_{t+1} - \tilde\theta_t\|_2^2 \qquad \text{(by Eq. (98))}$$

$$= -\eta \cdot \left\| \frac{\partial \tilde{V}^{\pi_{\tilde\theta_t}}(\mu)}{\partial \tilde\theta_t} \right\|_2^2 + \frac{5\cdot p^2 \cdot A^{2/p} + c_\tau}{(1-\gamma)^3} \cdot \eta^2 \cdot \left\| \frac{\partial \tilde{V}^{\pi_{\tilde\theta_t}}(\mu)}{\partial \tilde\theta_t} \right\|_2^2 \qquad \left(\tilde\zeta_{t+1} = \tilde\theta_t + \eta \cdot \frac{\partial \tilde{V}^{\pi_{\tilde\theta_t}}(\mu)}{\partial \tilde\theta_t}, \text{ Algorithm 2}\right)$$

$$= -\frac{(1-\gamma)^3}{20\cdot p^2 \cdot A^{2/p} + c_\tau} \cdot \left\| \frac{\partial \tilde{V}^{\pi_{\tilde\theta_t}}(\mu)}{\partial \tilde\theta_t} \right\|_2^2 \qquad \left(\eta = \frac{(1-\gamma)^3}{10\cdot p^2 \cdot A^{2/p} + c_\tau}\right)$$

$$\leq -\frac{(1-\gamma)^3}{20\cdot \cancel{p^2} \cdot A^{2/p} + c_\tau} \cdot \frac{2\tau}{S} \cdot \min_s \mu(s) \cdot \frac{\cancel{p^2} \cdot \min_{s,a} \pi_{\tilde\theta_t}(a|s)^{2-2/p}}{\max_s \|\tilde\theta_t(s,\cdot)\|_p^2} \cdot \left\| \frac{d_\mu^{\pi_\tau^*}}{d_\mu^{\pi_{\tilde\theta_t}}} \right\|_\infty^{-1} \cdot \left[\tilde{V}^{\pi_\tau^*}(\mu) - \tilde{V}^{\pi_{\tilde\theta_t}}(\mu)\right] \quad \text{(Lemma 12)}$$

$$= -\frac{(1-\gamma)^3 \cdot \tau}{(10\cdot A^{2/p} + c_\tau)\cdot S} \cdot \left\| \frac{d_\mu^{\pi_\tau^*}}{d_\mu^{\pi_{\tilde\theta_t}}} \right\|_\infty^{-1} \cdot \min_{s,a} \pi_{\tilde\theta_t}(a|s)^{2-2/p} \cdot \left[\tilde{V}^{\pi_\tau^*}(\mu) - \tilde{V}^{\pi_{\tilde\theta_t}}(\mu)\right] \qquad \left(\|\tilde\theta_t(s,\cdot)\|_p = 1, \text{ for all } s\right)$$

$$\leq -\frac{(1-\gamma)^4 \cdot \tau}{(10\cdot A^{2/p} + c_\tau)\cdot S} \cdot \left\| \frac{d_\mu^{\pi_\tau^*}}{\mu} \right\|_\infty^{-1} \cdot \min_{s,a} \pi_{\tilde\theta_t}(a|s)^{2-2/p} \cdot \left[\tilde{V}^{\pi_\tau^*}(\mu) - \tilde{V}^{\pi_{\tilde\theta_t}}(\mu)\right], \qquad \text{(by Eq. (72))}$$

$$\tag{99}$$

which implies,

$$\tilde{V}^{\pi_{\theta_t}}(\mu) - \tilde{V}^{\pi_{\theta_{t+1}}}(\mu) = \tilde{V}^{\pi_{\tilde\theta_t}}(\mu) - \tilde{V}^{\pi_{\tilde\theta_{t+1}}}(\mu) \qquad \text{(by Lemma 9)}$$

$$\leq -\frac{(1-\gamma)^4 \cdot \tau}{(10\cdot A^{2/p} + c_\tau)\cdot S} \cdot \left\| \frac{d_\mu^{\pi_\tau^*}}{\mu} \right\|_\infty^{-1} \cdot \min_{s,a} \pi_{\theta_t}(a|s)^{2-2/p} \cdot \left[\tilde{V}^{\pi_\tau^*}(\mu) - \tilde{V}^{\pi_{\theta_t}}(\mu)\right], \qquad \text{(by Eq. (99) and Lemma 9)}$$

$$\leq -\frac{(1-\gamma)^4 \cdot \tau}{(10\cdot A^{2/p} + c_\tau)\cdot S} \cdot \left\| \frac{d_\mu^{\pi_\tau^*}}{\mu} \right\|_\infty^{-1} \cdot c^{2-2/p} \cdot \left[\tilde{V}^{\pi_\tau^*}(\mu) - \tilde{V}^{\pi_{\theta_t}}(\mu)\right],$$

which is equivalent to,

$$\tilde{V}^{\pi_\tau^*}(\mu) - \tilde{V}^{\pi_{\theta_t}}(\mu) \leq \left(1 - \frac{(1-\gamma)^4 \cdot \tau}{(10\cdot A^{2/p} + c_\tau)\cdot S} \cdot \left\| \frac{d_\mu^{\pi_\tau^*}}{\mu} \right\|_\infty^{-1} \cdot c^{2-2/p}\right) \cdot \left[\tilde{V}^{\pi_\tau^*}(\mu) - \tilde{V}^{\pi_{\theta_{t-1}}}(\mu)\right]$$

$$\leq \exp\left\{-\frac{(1-\gamma)^4 \cdot \tau}{(10\cdot A^{2/p} + c_\tau)\cdot S} \cdot \left\| \frac{d_\mu^{\pi_\tau^*}}{\mu} \right\|_\infty^{-1} \cdot c^{2-2/p}\right\} \cdot \left[\tilde{V}^{\pi_\tau^*}(\mu) - \tilde{V}^{\pi_{\theta_{t-1}}}(\mu)\right]$$

$$\leq \exp\left\{-\frac{(1-\gamma)^4 \cdot \tau}{(10\cdot A^{2/p} + c_\tau)\cdot S} \cdot \left\| \frac{d_\mu^{\pi_\tau^*}}{\mu} \right\|_\infty^{-1} \cdot c^{2-2/p} \cdot (t-1)\right\} \cdot \left[\tilde{V}^{\pi_\tau^*}(\mu) - \tilde{V}^{\pi_{\theta_1}}(\mu)\right]$$

$$\leq \exp\left\{-\frac{(1-\gamma)^4 \cdot \tau}{(10\cdot A^{2/p} + c_\tau)\cdot S} \cdot \left\| \frac{d_\mu^{\pi_\tau^*}}{\mu} \right\|_\infty^{-1} \cdot c^{2-2/p} \cdot (t-1)\right\} \cdot \frac{1 + \tau \log A}{1 - \gamma},$$

which leads to the final result,

$$\tilde{V}^{\pi_\tau^*}(\rho) - \tilde{V}^{\pi_{\theta_t}}(\rho) \leq \frac{1}{1-\gamma} \cdot \left\| \frac{1}{\mu} \right\|_\infty \cdot \left[\tilde{V}^{\pi_\tau^*}(\mu) - \tilde{V}^{\pi_{\theta_t}}(\mu)\right] \qquad \text{(by Eq. (92))}$$

$$\leq \left\| \frac{1}{\mu} \right\|_\infty \cdot \exp\left\{-\frac{(1-\gamma)^4 \cdot \tau}{(10\cdot A^{2/p} + c_\tau)\cdot S} \cdot \left\| \frac{d_\mu^{\pi_\tau^*}}{\mu} \right\|_\infty^{-1} \cdot c'^2 \cdot (t-1)\right\} \cdot \frac{1 + \tau \log A}{(1 - \gamma)^2},$$

where $c' = c^{1-1/p} \geq c = \inf_{(s,a)} \inf_t \pi_{\theta_t}(a|s) > 0$. $\qquad\qquad\square$

**Lemma 15** (Soft sub-optimality lemma [14]). *For any policy $\pi$,*

$$\tilde{V}^{\pi_\tau^*}(\rho) - \tilde{V}^\pi(\rho) = \frac{1}{1-\gamma}\sum_s \left[ d_\rho^\pi(s) \cdot \tau \cdot D_{\mathrm{KL}}(\pi(\cdot|s)\|\pi_\tau^*(\cdot|s)) \right].$$

*Proof.* See the proof in [14, Lemma 26]. $\qquad\qquad\qquad\qquad\qquad\qquad\qquad\qquad\qquad\qquad\square$

## D   Proofs for Section 5 (Escort Cross Entropy)

**Lemma 16** (Non-uniform smoothness). *Let $\pi_\theta := f_p(\theta)$, and $\pi_{\theta'} := f_p(\theta')$. Denote $\theta_\xi := \theta + \xi \cdot (\theta' - \theta)$ with some $\xi \in [0,1]$. Then for $p = 2$, we have $D_{\mathrm{KL}}(y\|\pi_\theta)$ is $\beta$-smooth, i.e.,*

$$\left| D_{\mathrm{KL}}(y\|\pi_{\theta'}) - D_{\mathrm{KL}}(y\|\pi_\theta) - \left\langle \frac{d\{D_{\mathrm{KL}}(y\|\pi_\theta)\}}{d\theta}, \theta' - \theta \right\rangle \right| \le \frac{\beta}{2} \cdot \|\theta' - \theta\|_2^2,$$

*with $\beta = \frac{6}{\|\theta_\xi\|_p^2} + 2 \cdot \left( \max_i \frac{y(i)}{\theta(i)^2} \right)$.*

*Proof.* The gradient of $D_{\mathrm{KL}}(y\|\pi_\theta)$ w.r.t. $\theta$ is

$$
\begin{aligned}
\frac{d\{D_{\mathrm{KL}}(y\|\pi_\theta)\}}{d\theta} &= \frac{d\{-y^\top \log \pi_\theta\}}{d\theta} \\
&= \left( \frac{d\pi_\theta}{d\theta} \right)^\top \left( \frac{d\{-y^\top \log \pi_\theta\}}{d\pi_\theta} \right) \\
&= p \cdot \mathrm{diag}\left( \frac{1}{\theta} \right) \left( \mathrm{diag}(\pi_\theta) - \pi_\theta \pi_\theta^\top \right) \mathrm{diag}\left( \frac{1}{\pi_\theta} \right)(-y) \\
&= p \cdot \mathrm{diag}\left( \frac{1}{\theta} \right)(\pi_\theta - y).
\end{aligned}
$$

Denote the second derivative w.r.t. $\theta$ (i.e., Hessian) as

$$
\begin{aligned}
K(y,\theta) &= \frac{d}{d\theta}\left\{ \frac{d\{D_{\mathrm{KL}}(y\|\pi_\theta)\}}{d\theta} \right\} \\
&= p \cdot \frac{d}{d\theta}\left\{ \mathrm{diag}\left( \frac{1}{\theta} \right)(\pi_\theta - y) \right\}.
\end{aligned}
$$

We have $K(y,\theta) \in \mathbb{R}^{K \times K}$, whose element at position $(i,j) \in [K]^2$ is

$$
\begin{aligned}
K_{i,j} &= p \cdot \frac{d\{\frac{\pi_\theta(i) - y(i)}{\theta(i)}\}}{d\theta(j)} \\
&= p \cdot \frac{\frac{p}{\theta(j)} \cdot [\delta_{ij}\pi_\theta(j) - \pi_\theta(i)\pi_\theta(j)] \cdot \theta(i) - (\pi_\theta(i) - y(i)) \cdot \delta_{ij}}{\theta(i)^2} \\
&= p \cdot (p-1) \cdot \delta_{ij} \cdot \frac{\pi_\theta(i)}{\theta(i)^2} - p^2 \cdot \frac{\pi_\theta(i)}{\theta(i)} \cdot \frac{\pi_\theta(j)}{\theta(j)} + \delta_{ij} \cdot p \cdot \frac{y(i)}{\theta(i)^2},
\end{aligned}
$$

where the $\delta$ notation is defined in Eq. (23). For any $x \in \mathbb{R}^K$,

$$
\begin{aligned}
\left| x^\top K(y,\theta)x \right| &= \left| \sum_{i=1}^K \sum_{j=1}^K K_{i,j} x(i)x(j) \right| \\
&= \left| p \cdot (p-1) \sum_i \frac{\pi_\theta(i)}{\theta(i)^2} \cdot x(i)^2 - p^2 \sum_i \frac{\pi_\theta(i)}{\theta(i)} \cdot x(i) \sum_j \frac{\pi_\theta(j)}{\theta(j)} \cdot x(j) + p \sum_i \frac{y(i)}{\theta(i)^2} \cdot x(i)^2 \right| \\
&\le p \cdot (p-1) \cdot \left[ \sum_i \frac{\pi_\theta(i)}{\theta(i)^2} \cdot x(i)^2 \right] + p^2 \cdot \left[ \sum_i \frac{\pi_\theta(i)}{\theta(i)} \cdot x(i) \right]^2 + p \sum_i \frac{y(i)}{\theta(i)^2} \cdot x(i)^2, \quad (100)
\end{aligned}
$$

where the last inequality is by triangle inequality. The first term is upper bounded as,

$$\sum_i \frac{\pi_\theta(i)}{\theta(i)^2} \cdot x(i)^2 = \frac{1}{\|\theta\|_p^2} \sum_i \pi_\theta(i)^{1-2/p} \cdot x(i)^2$$

$$\le \frac{1}{\|\theta\|_p^2} \sum_i 1 \cdot x(i)^2 \qquad (p=2)$$

$$= \frac{1}{\|\theta\|_p^2} \cdot \|x\|_2^2. \tag{101}$$

The second term is upper bounded as,

$$\left[\sum_i \frac{\pi_\theta(i)}{\theta(i)} \cdot x(i)\right]^2 \le \sum_i \left(\frac{\pi_\theta(i)}{\theta(i)}\right)^2 \cdot \|x\|_2^2 \qquad \text{(by Cauchy-Schwarz)}$$

$$= \frac{1}{\|\theta\|_p^2} \cdot \sum_i \left(\pi_\theta(i)^{1-1/p}\right)^2 \cdot \|x\|_2^2$$

$$\le \frac{1}{\|\theta\|_p^2} \cdot \left[\sum_i \pi_\theta(i)\right] \cdot \|x\|_2^2 \qquad (p=2)$$

$$= \frac{1}{\|\theta\|_p^2} \cdot \|x\|_2^2. \tag{102}$$

The last term is upper bounded as,

$$\sum_i \frac{y(i)}{\theta(i)^2} \cdot x(i)^2 \le \left(\max_i \frac{y(i)}{\theta(i)^2}\right) \cdot \|x\|_2^2. \tag{103}$$

Combining Eqs. (100) to (103), for $p=2$, for any $x \in \mathbb{R}^K$, we have,

$$\left|x^\top K(y,\theta)x\right| \le p \cdot (p-1) \cdot \frac{1}{\|\theta\|_p^2} \cdot \|x\|_2^2 + p^2 \cdot \frac{1}{\|\theta\|_p^2} \cdot \|x\|_2^2 + p \cdot \left\|\frac{y}{\theta \odot \theta}\right\|_\infty \cdot \|x\|_2^2$$

$$= \frac{6}{\|\theta\|_p^2} \cdot \|x\|_2^2 + 2 \cdot \left(\max_i \frac{y(i)}{\theta(i)^2}\right) \cdot \|x\|_2^2.$$

According to Taylor's theorem, we have,

$$\left|D_{\mathrm{KL}}(y\|\pi_{\theta'}) - D_{\mathrm{KL}}(y\|\pi_\theta) - \left\langle\frac{d\{D_{\mathrm{KL}}(y\|\pi_\theta)\}}{d\theta}, \theta' - \theta\right\rangle\right| = \frac{1}{2} \cdot \left|(\theta'-\theta)^\top K(y,\theta_\xi)(\theta'-\theta)\right|$$

$$\le \left[\frac{3}{\|\theta_\xi\|_p^2} + \max_i \frac{y(i)}{\theta_\xi(i)^2}\right] \cdot \|\theta'-\theta\|_2^2. \qquad \square$$

**Lemma 17** (Non-uniform Łojasiewicz). *Let $\pi_\theta = f_p(\theta)$. For any $p \ge 2$, we have,*

$$\left\|\frac{d\{D_{\mathrm{KL}}(y\|\pi_\theta)\}}{d\theta}\right\|_2^2 \ge \frac{p^2}{\|\theta\|_p^2} \cdot \min_a \pi_\theta(a)^{1-2/p} \cdot D_{\mathrm{KL}}(y\|\pi_\theta).$$

*Proof.* According to the definition of KL-divergence, we have,

$$D_{\mathrm{KL}}(y\|\pi_\theta) = \sum_a y(a) \cdot \log\left(\frac{y(a)}{\pi_\theta(a)}\right)$$

$$\leq \sum_a y(a) \cdot \left(\frac{y(a)}{\pi_\theta(a)} - 1\right) \qquad (\log x \leq x - 1)$$

$$= \sum_a (y(a) - \pi_\theta(a) + \pi_\theta(a)) \cdot \frac{y(a) - \pi_\theta(a)}{\pi_\theta(a)}$$

$$= \sum_a \frac{(y(a) - \pi_\theta(a))^2}{\pi_\theta(a)}$$

$$= \sum_a \frac{(y(a) - \pi_\theta(a))^2}{\pi_\theta(a)^{2/p}} \cdot \frac{1}{\pi_\theta(a)^{1-2/p}}$$

$$= \sum_a \frac{(y(a) - \pi_\theta(a))^2}{\theta(a)^2} \cdot \|\theta\|_p^2 \cdot \frac{1}{\pi_\theta(a)^{1-2/p}} \qquad \left(\pi_\theta(a) = \frac{|\theta(a)|^p}{\sum_{a'}|\theta(a')|^p}\right)$$

$$\leq \|\theta\|_p^2 \cdot \frac{1}{\min_a \pi_\theta(a)^{1-2/p}} \cdot \sum_a \frac{(y(a) - \pi_\theta(a))^2}{\theta(a)^2}$$

$$= \|\theta\|_p^2 \cdot \frac{1}{\min_a \pi_\theta(a)^{1-2/p}} \cdot \frac{1}{p^2} \cdot \left\| p \cdot \mathrm{diag}\left(\frac{1}{\theta}\right)(y - \pi_\theta) \right\|_2^2.$$

The proof is completed with the observation that

$$\frac{d\{D_{\mathrm{KL}}(y\|\pi_\theta)\}}{d\theta} = p \cdot \mathrm{diag}\left(\frac{1}{\theta}\right)(\pi_\theta - y). \qquad \square$$

**Theorem 5.** Using the escort transform on the cross entropy objective, we have, for all $t \geq 1$,

- (gradient flow) for $p = 2$, with $\eta_t = \frac{\|\theta_t\|_p^2}{p^2}$,

$$D_{\mathrm{KL}}(y\|\pi_{\theta_t}) \leq D_{\mathrm{KL}}(y\|\pi_{\theta_1}) \cdot e^{-(t-1)};$$

- (gradient descent) for $p = 2$, with $\eta_t = \frac{\|\theta_t\|_p^2}{4 \cdot (3 + c_1^2)}$,

$$D_{\mathrm{KL}}(y\|\pi_{\theta_t}) \leq D_{\mathrm{KL}}(y\|\pi_{\theta_1}) \cdot \exp\left\{ -\frac{(t-1)}{2 \cdot (3 + c_1^2)} \right\},$$

where $\frac{1}{c_1^2} = \frac{|\theta_1(a_y)|^2}{\|\theta_1\|_2^2} = \pi_{\theta_1}(a_y) \in (0, 1]$ only depends on initialization.

*Proof.* **First part.** For the gradient flow, we have the following update,

$$\frac{d\theta_t}{dt} = -\eta_t \cdot \frac{d\{D_{\mathrm{KL}}(y\|\pi_{\theta_t})\}}{d\theta_t}. \tag{104}$$

Then we have,

$$\frac{d\{D_{\mathrm{KL}}(y\|\pi_{\theta_t})\}}{dt} = \left(\frac{d\theta_t}{dt}\right)^\top \left(\frac{d\{D_{\mathrm{KL}}(y\|\pi_{\theta_t})\}}{d\theta_t}\right)$$

$$= -\eta_t \cdot \left\| \frac{d\{D_{\mathrm{KL}}(y\|\pi_{\theta_t})\}}{d\theta_t} \right\|_2^2 \qquad \text{(by Eq. (104))}$$

$$\leq -\eta_t \cdot \frac{p^2}{\|\theta_t\|_p^2} \cdot \min_a \pi_{\theta_t}(a)^{1-2/p} \cdot D_{\mathrm{KL}}(y\|\pi_{\theta_t}) \qquad \text{(by Lemma 17)}$$

$$= -\min_a \pi_{\theta_t}(a)^{1-2/p} \cdot D_{\mathrm{KL}}(y\|\pi_{\theta_t}) \qquad \left(\eta_t = \frac{\|\theta_t\|_p^2}{p^2}\right)$$

$$= -D_{\mathrm{KL}}(y\|\pi_{\theta_t}), \qquad (p = 2)$$

which implies,

$$\frac{d\{\log D_{\mathrm{KL}}(y\|\pi_{\theta_t})\}}{dt} = \frac{1}{D_{\mathrm{KL}}(y\|\pi_{\theta_t})} \cdot \frac{d\{D_{\mathrm{KL}}(y\|\pi_{\theta_t})\}}{dt} \leq -1.$$

Taking integral, we have,

$$\log D_{\mathrm{KL}}(y\|\pi_{\theta_t}) - \log D_{\mathrm{KL}}(y\|\pi_{\theta_1}) \leq -(t-1),$$

which is equivalent to

$$D_{\mathrm{KL}}(y\|\pi_{\theta_t}) \leq D_{\mathrm{KL}}(y\|\pi_{\theta_1}) \cdot e^{-(t-1)}.$$

**Second part.** For the gradient descent, according to Lemma 16, we have,

$$D_{\mathrm{KL}}(y\|\pi_{\theta_{t+1}}) - D_{\mathrm{KL}}(y\|\pi_{\theta_t}) - \left\langle \frac{d\{D_{\mathrm{KL}}(y\|\pi_{\theta_t})\}}{d\theta_t}, \theta_{t+1} - \theta_t \right\rangle \leq \frac{\beta}{2} \cdot \|\theta_{t+1} - \theta_t\|_2^2, \quad (105)$$

where

$$\begin{aligned}
\beta &= \frac{6}{\|\theta_{t,\xi}\|_p^2} + 2 \cdot \left( \max_i \frac{y(i)}{\theta_{t,\xi}(i)^2} \right) \\
&= \frac{6}{\|\theta_{t,\xi}\|_p^2} + \frac{2}{\theta_{t,\xi}(a_y)^2}, \qquad (y \text{ is one-hot}) \quad (106)
\end{aligned}$$

and

$$\theta_{t,\xi} := \theta_t + \xi \cdot (\theta_{t+1} - \theta_t) = \theta_t - \xi \cdot \eta_t \cdot \frac{d\{D_{\mathrm{KL}}(y\|\pi_{\theta_t})\}}{d\theta_t}. \quad (107)$$

The $\ell_p$ gradient norm is upper bounded as,

$$\begin{aligned}
\left\| \frac{d\{D_{\mathrm{KL}}(y\|\pi_{\theta_t})\}}{d\theta_t} \right\|_p &= \left[ \sum_a \left| \frac{\pi_{\theta_t}(a) - y(a)}{\theta_t(a)} \right|^p \right]^{\frac{1}{p}} \\
&= \left[ \sum_{a \neq a_y} \left| \frac{\pi_{\theta_t}(a)}{\theta_t(a)} \right|^p + \left| \frac{\pi_{\theta_t}(a_y) - 1}{\theta_t(a_y)} \right|^p \right]^{\frac{1}{p}} \qquad (y(a_y) = 1) \\
&\leq \left[ \sum_{a \neq a_y} \left| \frac{\pi_{\theta_t}(a)}{\theta_t(a)} \right|^p + \frac{1}{|\theta_t(a_y)|^p} \right]^{\frac{1}{p}} \qquad (\pi_{\theta_t}(a_y) \in (0,1]) \\
&= \left[ \frac{1}{\|\theta_t\|_p^p} \sum_{a \neq a_y} \pi_{\theta_t}(a)^{p-1} + \frac{1}{|\theta_t(a_y)|^p} \right]^{\frac{1}{p}} \\
&\leq \left[ \frac{1}{\|\theta_t\|_p^p} + \frac{1}{|\theta_t(a_y)|^p} \right]^{\frac{1}{p}} \qquad (p = 2) \\
&\leq \frac{1}{\|\theta_t\|_p} + \frac{1}{|\theta_t(a_y)|} \qquad \left( \sqrt{x+y} \leq \sqrt{x} + \sqrt{y} \right) \quad (108)
\end{aligned}$$

Next, we have,

$$\theta_{t+1}(a_y) = \theta_t(a_y) - \eta_t \cdot \frac{p}{\theta_t(a_y)} \cdot (\pi_{\theta_t}(a_y) - 1) \begin{cases} \geq \theta_t(a_y), & \text{if } \theta_t(a_y) > 0, \\ \leq \theta_t(a_y), & \text{if } \theta_t(a_y) < 0. \end{cases}$$

Therefore we have $|\theta_{t+1}(a_y)| \geq |\theta_t(a_y)|$. On the other hand, for all $a \neq a_y$, we have,

$$\theta_{t+1}(a) = \theta_t(a) - \eta_t \cdot \frac{p}{\theta_t(a)} \cdot \pi_{\theta_t}(a) \begin{cases} \leq \theta_t(a_y), & \text{if } \theta_t(a_y) > 0, \\ \geq \theta_t(a_y), & \text{if } \theta_t(a_y) < 0. \end{cases}$$

Therefore we have for all $a \neq a_y$, $|\theta_{t+1}(a)| \leq |\theta_t(a)|$. Denote $\frac{1}{c_1} = \frac{|\theta_1(a_y)|}{\|\theta_1\|_p}$. We have, for all $t \geq 1$,

$$\frac{|\theta_t(a_y)|}{\|\theta_t\|_p} = \frac{|\theta_t(a_y)|}{\left(\sum_a |\theta_t(a)|^p\right)^{1/p}} \geq \frac{|\theta_t(a_y)|}{\left(\sum_{a \neq a_y} |\theta_1(a)|^p + |\theta_t(a_y)|^p\right)^{1/p}}$$

$$\geq \frac{|\theta_1(a_y)|}{\left(\sum_{a \neq a_y} |\theta_1(a)|^p + |\theta_1(a_y)|^p\right)^{1/p}}$$

$$= \frac{|\theta_1(a_y)|}{\|\theta_1\|_p} = \frac{1}{c_1}. \tag{109}$$

Combining Eqs. (108) and (109), we have,

$$\left\| \frac{d\{D_{\mathrm{KL}}(y\|\pi_{\theta_t})\}}{d\theta_t} \right\|_p \leq \frac{1}{\|\theta_t\|_p} + \frac{1}{|\theta_t(a_y)|} \leq \frac{1}{\|\theta_t\|_p} \cdot (1 + c_1). \tag{110}$$

Then we have,

$$\|\theta_{t,\xi}\|_p = \left\| \theta_t - \xi \cdot \eta_t \cdot \frac{d\{D_{\mathrm{KL}}(y\|\pi_{\theta_t})\}}{d\theta_t} \right\|_p \qquad \text{(by Eq. (107))}$$

$$\geq \|\theta_t\|_p - \xi \cdot \eta_t \cdot \left\| \frac{d\{D_{\mathrm{KL}}(y\|\pi_{\theta_t})\}}{d\theta_t} \right\|_p \qquad \text{(by triangle inequality)}$$

$$\geq \|\theta_t\|_p - \xi \cdot \eta_t \cdot \frac{1}{\|\theta_t\|_p} \cdot (1 + c_1). \qquad \text{(by Eq. (110))}$$

$$= \|\theta_t\|_p \cdot \left[ 1 - \xi \cdot \frac{1 + c_1}{4 \cdot (3 + c_1^2)} \right] \qquad \left( \eta_t = \frac{\|\theta_t\|_p^2}{4 \cdot (3 + c_1^2)} \right)$$

$$\geq \|\theta_t\|_p \cdot \left[ 1 - \frac{1 + c_1}{4 \cdot (3 + c_1^2)} \right] \qquad (\xi \in [0, 1])$$

$$= \|\theta_t\|_p \cdot \left[ \left( 1 - \frac{1}{\sqrt{2}} \right) \cdot \left( 1 - \frac{\sqrt{2} + 1}{2\sqrt{2}} \cdot \frac{1 + c_1}{3 + c_1^2} \right) + \frac{1}{\sqrt{2}} \right]$$

$$\geq \frac{\|\theta_t\|_p}{\sqrt{2}} \qquad (1/c_1 \in (0, 1], \ c_1 \geq 1) \tag{111}$$

Similar to Eq. (109), we have,

$$\beta = \frac{6}{\|\theta_{t,\xi}\|_p^2} + \frac{2}{\theta_{t,\xi}(a_y)^2} \qquad \text{(by Eq. (106))}$$

$$\leq \frac{1}{\|\theta_{t,\xi}\|_p^2} \cdot \left( 6 + 2 \cdot c_1^2 \right). \tag{112}$$

Combining the results, we have,

$$D_{\mathrm{KL}}(y\|\pi_{\theta_{t+1}}) - D_{\mathrm{KL}}(y\|\pi_{\theta_t}) - \left\langle \frac{d\{D_{\mathrm{KL}}(y\|\pi_{\theta_t})\}}{d\theta_t}, \theta_{t+1} - \theta_t \right\rangle$$

$$\leq \frac{1}{2} \cdot \frac{1}{\|\theta_{t,\xi}\|_p^2} \cdot \left( 6 + 2 \cdot c_1^2 \right) \cdot \|\theta_{t+1} - \theta_t\|_2^2 \qquad \text{(by Eqs. (105) and (112))}$$

$$\leq \frac{2}{\|\theta_t\|_p^2} \cdot \left( 3 + c_1^2 \right) \cdot \|\theta_{t+1} - \theta_t\|_2^2, \qquad \text{(by Eq. (111))}$$

which implies (using the update $\theta_{t+1} = \theta_t - \eta_t \cdot \frac{d\{D_{\mathrm{KL}}(y\|\pi_{\theta_t})\}}{d\theta_t}$),

$$D_{\mathrm{KL}}(y\|\pi_{\theta_{t+1}}) - D_{\mathrm{KL}}(y\|\pi_{\theta_t}) \le -\eta_t \cdot \left\|\frac{d\{D_{\mathrm{KL}}(y\|\pi_{\theta_t})\}}{d\theta_t}\right\|_2^2 + \frac{2 \cdot (3 + c_1^2)}{\|\theta_t\|_p^2} \cdot \eta_t^2 \cdot \left\|\frac{d\{D_{\mathrm{KL}}(y\|\pi_{\theta_t})\}}{d\theta_t}\right\|_2^2$$

$$= -\frac{\|\theta_t\|_p^2}{8 \cdot (3 + c_1^2)} \cdot \left\|\frac{d\{D_{\mathrm{KL}}(y\|\pi_{\theta_t})\}}{d\theta_t}\right\|_2^2 \qquad \left(\eta_t = \frac{\|\theta_t\|_p^2}{4 \cdot (3 + c_1^2)}\right)$$

$$\le -\frac{\|\theta_t\|_p^2}{8 \cdot (3 + c_1^2)} \cdot \frac{p^2}{\|\theta_t\|_p^2} \cdot \min_a \pi_{\theta_t}(a)^{1-2/p} \cdot D_{\mathrm{KL}}(y\|\pi_{\theta_t}) \qquad \text{(by Lemma 17)}$$

$$= -\frac{1}{2 \cdot (3 + c_1^2)} \cdot D_{\mathrm{KL}}(y\|\pi_{\theta_t}), \qquad (p = 2)$$

which is equivalent to,

$$D_{\mathrm{KL}}(y\|\pi_{\theta_t}) \le \left[1 - \frac{1}{2 \cdot (3 + c_1^2)}\right] \cdot D_{\mathrm{KL}}(y\|\pi_{\theta_{t-1}})$$

$$\le D_{\mathrm{KL}}(y\|\pi_{\theta_{t-1}}) \cdot \exp\left\{-\frac{1}{2 \cdot (3 + c_1^2)}\right\}$$

$$\le D_{\mathrm{KL}}(y\|\pi_{\theta_1}) \cdot \exp\left\{-\frac{(t-1)}{2 \cdot (3 + c_1^2)}\right\},$$

where $\frac{1}{c_1^2} = \frac{|\theta_1(a_y)|^2}{\|\theta_1\|_2^2} = \pi_{\theta_1}(a_y) \in (0, 1]$. $\qquad\qquad\square$

## E  Experimental Details and Additional Experiments

For the one-state MDPs, for each value of $K \in \{10, 50, 100\}$, the policy is parameterized by $\theta \in \mathbb{R}^K$. For SPG, $\pi_\theta = \mathrm{softmax}(\theta)$, and for EPG $\pi_\theta = f_p(\theta)$. The total number of runs for each algorithm under each $K$ value is 20. In each run, we randomly generate the reward $r \in [0, 1]^K$, and then randomly initialize $\pi_{\theta_1}$ within the $(K-1)$-dimensional probability simplex. SPG and EPG start from the same initial policy $\pi_{\theta_1}$. The total number of iterations is $T = 5 \times 10^4$. Fig. 5 shows the results of SPG and EPG with $p = 2$. The learning rate of SPG is set to be $\eta = 0.4$ [14]. The learning rate of EPG is $\eta_t = 0.2 \cdot \|\theta_t\|_p^2$ (Theorem 2).

**Additional experiments**  Fig. 7(a)-(c) show the results of EPG for $p \in \{2, 3, 4, 5\}$ in one-state MDPs, where each curve is the averaged result of 20 runs.

For the Four-room environment, the policy is $\pi_\theta = \mathrm{softmax}(\theta)$ for SPG, and $\pi_\theta = f_p(\theta)$ for EPG, and $\theta$ is the output of one hidden layer neural network with ReLU activation function and 64 hidden nodes. Fig. 5 shows the results of SPG and EPG with $p = 2$. The optimal value function $V^*$ is approximately calculated using value iteration with threshold of two consecutive iterations $\|V_t - V_{t+1}\|_2^2 \le 1 \times 10^{-10}$. In each iteration, the true objective is used by calculating the stationary distribution $d^{\pi_{\theta_t}}$ and the state-action value $Q^{\pi_{\theta_t}}$. We use Adam optimizer [10] and the total number of iterations is 500. The total number of runs for each algorithm is 20. The $p$ value for EPG is searched within $\{1, 2, 3, 4, 5\}$. The learning rates for SPG and EPG are searched within $\{0.001, 0.005, 0.01, 0.05, 0.1, 0.5\}$ and $0.01$ is used for both SPG and EPG.

Figure 7: *Results of EPG with different $p$ values on one-state MDPs and Four-room.*

**Additional experiments** Fig. 7(d) shows the results of EPG with $p \in \{1, 2, 3, 4, 5\}$ on Four-room environment, where each curve is the averaged result of 20 runs.

For the MNIST dataset, the policy is $\pi_\theta = \text{softmax}(\theta)$ for SPG and SCE, and $\pi_\theta = f_p(\theta)$ for EPG and ECE, where $\theta$ is the output of one hidden layer neural network with ReLU activation function and $512$ hidden nodes. We use SGD with momentum $0.9$ and the total number of epochs is $100$. The total number of runs for each algorithm is $20$. Fig. 6(a) and (b) show the results of SPG and EPG with $p = 4$. The learning rates for SPG and EPG are searched within $\{0.001, 0.005, 0.01, 0.05, 0.1, 0.5\}$ and $0.05$ is used for both SPG and EPG. The batchsize is searched within $\{10, 20, 50, 100, 200, 500\}$, and $20$ is used for for SPG and $50$ is used for EPG. The $p$ value for EPG is searched within $\{1, 2, 3, 4, 5\}$. Fig. 6(c) and (d) show the results of SCE and ECE with $p = 2$. The learning rate and batchsized are searched within the same range as above, and we use the learning rate $0.01$, and the batchsize $20$ for both SCE and ECE.

**Additional experiments** Fig. 8 shows the results of EPG with $p \in \{2, 3, 4, 5\}$ on MNIST, where each curve is the averaged result of 10 runs. The best result in terms of the test error is with $p = 5$.

(a) Training objective.    (b) Training error.    (c) Test error.

Figure 8: *Results of EPG with different p values on MNIST.*

## E.1 Failure of SPG Heuristics

According to Theorem 1, the SGW results from the SPG progresses being upper bounded by the small probabilities of the optimal action. Therefore, one may wonder if some simple heuristics would fix the SGW problem. We consider the following two natural heuristics.

(a) SPG and EPG with $p = 2$.    (b) SPG-L: SPG with large $\eta_t = 0.4/\pi_{\theta_t}(a^*)$.    (c) SPG-N: SPG with $\ell_2$ normalized gradient.

Figure 9: *Sub-optimality results $\left(\pi^* - \pi_{\theta_t}\right)^\top r$ on a single-state MDP with $K = 50$.*

**SPG-L: Large learning rate** Since the progress in each iteration of SPG is upper bounded by the order of $\pi_{\theta_t}(a^*)$, we could possibly use a large learning rate such as $\eta_t = 0.4/\pi_{\theta_t}(a^*)$ in SPG to compensate the slow progress around SGWs.

We run the SPG-L update on a constructed bandit problem with $K = 50$. Unfortunately, as shown in Fig. 9(b), after about $1 \times 10^6$ iterations, SPG-L still gets stuck on a sub-optimal plateau.

The intuitive explanation for the failure of SPG-L is that learning rate should be small enough to guarantee monotonic improvement [1, 14] for smooth functions, which means SPG with large (unbounded) learning rate is not even guaranteed to be asymptotically convergent.

**SPG-N: Normalized policy gradient** Another heuristic is to update using normalized policy gradient. The intuition is that since the progress is upper bounded by the original SPG norm, which has small scale, we could hope the normalized gradient would provide nicer progress. In particular, we normalize the policy gradient by dividing its $\ell_2$ norm, and then do update with $\eta = 0.4$.

We run the SPG-N update on the same problem as shown in Fig. 9(c), and it also failed. We do not have rigorous explanations for the failure of SPG-N. A speculation is that after $\ell_2$ normalization, the

normalized policy gradient still has small scale, since it is not necessary that the policy gradient norm is on the same scale as $\pi_{\theta_t}$, which is the quantity that needs to be canceled.

**EPG**    As shown in Fig. 9(a), EPG with $p = 2$ does not suffer from the plateau and works well.

## E.2    Comparing SPG, EPG, and MD

As noted in Remark 3, EPG cannot be reduced to MD with any regularizer. Also as shown in Fig. 3(b), EPG and MD with KL regularization behave similarly in the 3-action case. We conduct experiments on bandit problems with $K = 50, 100, 500$ actions to compare EPG with MD. In each iteration, all the algorithms use the same stochastic gradient to do updates. Each curve is averaged over 20 runs.

Figure 10:  *Sub-optimality results* $(\pi^* - \pi_{\theta_t})^\top r$ *on single-state MDPs using stochastic gradients.*

As shown in Fig. 10, EPG and MD with KL regularization have comparable performances, significantly outperforming SPG. However, EPG in its nature is a policy gradient method, which has cheap update in each iteration, while MD needs to solve an optimization problem to do one update.