[Reviews · NeurIPS 2020]

Review 1

Summary and Contributions: This paper identifies a fundamental phenomenon which occurs in both policy gradients (RL) and cross entropy (SL) objectives when using softmax activations. In policy gradients, learning seems to plateu. The authors show this is due to a gravity well where the gradient tends toward suboptimal actions, i.e., pulls the solution toward suboptimal results. While it is known that policy gradient converges to an optimal solution, this causes a substantial slowdown in convergence, as shown by their theory. The authors explain this phenomenon through Łojasiewicz Coefficient and propose the Escort Transform as a way to mitigate this problem. They then use this approach in a supervised learning setting and show that it can also improve performance through a change in learning rates. Finally, the authors provide several experiments to illustrate their theoretical results.

Strengths: - Novel and significant advancement in understanding PG and SL convergence under softmax activations. - Strong and tight theoretical and convergence results - Non trivial use of the Łojasiewicz coefficient due to inability to apply the Escort Transform to mirror descent - Very well written

Weaknesses: The main weakness in this work is the experiments section: 1. While this is mostly a theoretical paper, it is of interest to a very broad community, including practicians in both RL and SL. While testing theoretical result empirically (large scale) may be out of their scope of many theoretical papers, I believe that in the case of this paper the experiments would be easy to establish -- all that is needed is to take practical PG and SL algorithms and change the softmax with the escort transform. I think that large scale, non tabular experiments could really push the impact of this paper to a whole new level. 2. It is unclear to me how practical the escort transform is due to it not being differentiable. Could the authors elaborate on this? This is also related to my first point w.r.t. the applicability of the approach.

Correctness: The theoretical claims seem correct.

Clarity: The paper is very well written. The sections are organized well and it flows clearly. Perhaps a minor comment would be to add a little more explanation in the captions of the figures themselves, as they are not entirely clear.

Relation to Prior Work: The authors discuss the relevant related work I am aware of.

Reproducibility: Yes

Additional Feedback: Minor comments: - Figure 1d: What is the y axis? - Could you explain the tradeoff in p using the escort transform? Other than the fact that p->infty behaves like the softmax, why would one want to simply use p=1? Typos: Line 66: the the 286: advantages -> disadvantages? After reading the authors' comments and other reviews I have decided to keep my current score.


Review 2

Summary and Contributions: The paper considers the commonly used softmax function and argues it has undesired properties such as sensitivity to initial conditions and damped convergence over time. I've read the rebuttal and my score persists. While the paper is good in my opinion, I still feel it's biggest issues are the focus on RL (where softmax is used in numerous problems) and the lack of bottom line (along the lines of 'use escort p=3 in the following cases...'). I guess that stronger results would have to be left for future work.

Strengths: Good title. The paper attacks a very common practice which I highly appreciate. The claims seem to be sound, supporting empirical evaluation is given. Were the paper perceived as convincing, that could have very high significance in many applications. The work is very relevant to the NeurIPS community.

Weaknesses: I'v had several issues with the paper: 1. While Softmax is used in numerous problems and cases, the paper seems to put a lot of emphasis on RL. As an RL person that is one reason this paper reached my review, however - the core ideas and technicalities beyond the paper has very little to do with RL specifically (even the examples are more about bandits than RL). I would much assume such a paper should reach more optimization related reviewers\venues. This also makes me somewhat suspicious regarding the novelty of the paper, though I myself have little to contradict it with. 2.The authors encourage the use of escort transform over softmax. However, it feels as though some beneficial properties of softmax are ignored or not considered making the case one sided: a. Softmax works on numbers in the entire real range without requiring absolute values, while escort transform ignored negative values. While this on itself doesn't too bad, I would suspect that large step-sizes with respect to the normalized weights are more prone to cause jitter around 0 (as both the plus and negative solutions are identical). This probably means that larger step-sizes can be used with softmax. b. Softmax has a natural way to include "temperature". 3. It's a common and logical practice to use all initial probabilities equal 1/K. Some reference to that would been in order. 4. The paper deals with finding alternatives to softmax, but is the proposed escort transform the best? could there be something better? How to choose p? I feel like a stronger case should have been made in these prospects.

Correctness: Yes.

Clarity: Yes, the paper is very clear.

Relation to Prior Work: I can't say otherwise, but feel some more discussion in needed.

Reproducibility: Yes

Additional Feedback:


Review 3

Summary and Contributions: ##Update## The rebuttal adequately addressed my main concerns and I am consequently increasing my score to a 7. In particular I was pleased that the authors investigated the issues with the learning rate, and I would be happy if they mention this potential limitation in their revisions, and include the experimental results showing that the naive adaptive learning rate proposals I made would not be effective. It was also pleasing that they will discuss and compare with Neural Replicator Dynamics, and the additional experiment with sampled actions also looks promising. The reason I didn't increase my score further was that the current set of experiments is still rather simple, and it is difficult for me to assess whether the new method is likely to be widely used. Though, I feel that the contribution may well turn out to be much more influential. EPG learning rate bound: the learning rate can be "arbitrarily high" if theta is initialized arbitrarily high. This may be nitpicking, but if the initialization is not done "adversarially" there would be no theoretical issue with softmax either (at least not in the current derivation). In any case, my main concern was just that the theory did not seem to say that the issue with softmax can not be solved by using an adaptive learning rate, so the discussion around it should also be clear about this to avoid misleading the readers. I also took another look at the proofs, and I couldn't find any mistakes, but I added a few suggestions to the detailed comments section below. ############################################# The paper discusses phenomena related to softmax policy gradients, which cause the magnitude of the gradients to become very small close to deterministic policies. This can cause the learning to get stuck on plateaus near policies which pick a suboptimal action close to deterministically. These issues were illustrate well in simple MDP domains in section 2. The same phenomenon was recently discussed in another work in multiagent learning [Hennes et al. "Neural replicator dynamics" AAMAS2020], but the authors appeared unaware of this work (AAMAS2020 was quite close to the NeurIPS submission deadline, but on the other hand the paper was on arxiv for around a year. I am not sure what the verdict should be here). However, there is room for more research in this area, and they provide a complementary view of the issue. The main contributions included a theoretical analysis of the softmax policy gradients based on the Lojasiewicz coefficient. In particular, they derived a lower bound on the minimum escape time from a "bad" initialization (Thm 1), which showed that the escape time for softmax policy gradients are inversely proportional to the probability of the optimal action (thus, if the probability of the optimal action is very low, the escape time could be very large). Previous theoretical analyses of policy gradients gave good O(1/t) convergence rates in the standard setting and O(e^(-t)) convergence rates when entropy regularization is applied, so the current analysis is complementary to such analyses, and instead shows that the constants for these rates matter, and in practice, the optimization can perform poorly. The authors propose an alternative to the softmax, which they call the escort transform, and it is given by pi(a) = |theta(a)|/sum_{b in A}(|theta(b)|). They show that using an adaptive learning rate schedule, this transform obtains better bounds on the escape time, and better constants on the convergence rate. They also derive similar bounds in an MDP setting, as opposed to the bandit setting of the initial bounds. They prove that the escort transform cannot be derived from the Mirror Descent/Follow the Regularized Leader formulation. Note that the Neural Replicator Dynamics is essentially a way to combine Mirror Descent into a policy parameterized with a neural network, so their approach here is different to the other work tackling the same issue. They also discuss another issue related to the softmax, which they call softmax damping, which causes the steps to become very small when nearing the optimal solution, thus slow down the convergence in cross-entropy classification optimization. They perform experiments on a four room MDP reinforcement learning domain, where they compute the exact Q-values for the policy gradient considering all actions, and also perform experiments on MNIST classification compared to softmax policy gradients with a reward 1 for correct classification, as well as cross-entropy loss classification and achieve improved performance in all cases.

Strengths: Softmax is widely used, so investigating when it gives poor performance can have a high impact. The analysis for the escape time was very insightful, and appeared like a novel way to theoretically characterize the practical performance of policy gradient algorithms. The experiments included both simple toy problems that help verify the insights from the theory (e.g. the bandit problems for the plateaus, and simple cross-entropy toy problem in Fig 4 to examine the damping phenomenon), as well as slightly more real problems on MDPs and MNIST classification.

Weaknesses: The issues with softmax policy gradients were explored in the Neural Replicator Dynamics paper, but the current paper does not cite or discuss this paper. (It may be controversial whether they need to discuss it or not, as mentioned in the summary, but I guess 1 year on arxiv and already published in a peer reviewed venue may be enough to require a comparison). There were no experiments on larger tasks, e.g. classification on larger datasets, or reinforcement learning tasks in actor-critic methods that include approximated Q-functions, etc. So, it is unclear whether the escort policy gradient will work in practice. Considering that the method appears simple to implement, it is surprising that more elaborate experiments were not performed. It would be better if the paper at least gives some indication as to whether the method will work on harder problems or not---whether it will work out of the box, or whether making it work will require more research. There were no experiments exploring how the newly proposed escort policy gradient works with sampled actions (the experiments assumed the Q-values for each action were available). The work builds quite a lot on another recent work: "On the Global Convergence Rates of Softmax Policy Gradient Methods", ICML2020, which is not a serious weakness, but reduces the novelty of the analysis to some extent. For example, the point that softmax can be stuck close to suboptimal corners was also made in this previous work, although the analysis was not as extensive. It was unclear to me why the escort transform was chosen. It appeared picked out of the air, and justified retrospectively based on the convergence analysis. A more intuitive explanation would make the method more trustworthy. For example, it would be interesting to know why the authors chose this transform. The Lojasiewicz coefficient was used to analyze the specific cases of the softmax policy gradient and the escort policy gradient, but as far as I could tell, the universality of the concept was not established or explained. If I come up with a new policy gradient method, and derive the Lojasiewicz coefficient, will this tell me anything about the convergence rate, or do I need to do a new analysis? I also had some technical concerns, although maybe it is due to my misunderstanding. I discuss it below. In the analysis for the policy gradient, the bound for the escape time is based on bounding the sizes of the gradient steps. This bound is based on setting a constraint that the learning rate is in (0, 1]. On the other hand, for the escort policy gradient, no such constraint on the learning rate schedule is set. Instead, the learning rate is adaptive, and scaled by p-norm(theta). So, the learning rate can become arbitrarily high in this learning rate schedule. Therefore the comparison does not seem entirely fair. What if you use a rate schedule and simply scale the policy gradient magnitude by 1/pi(a*) (The inverse of the Lojasiewicz coefficient)? Note that I already read Algorithm 2 and Lemma 9, which stated that normalizing the thetas with the p-norm after each step is equivalent to using that rate schedule, but I am not entirely convinced, because that makes it not a pure policy gradient algorithm, so I am wondering whether a better rate schedule for the softmax PG may also improve its performance (Also, please add a pointer to algorithm 2 and Lemma 9 somewhere into the main paper to explain why the rate includes the norm in Thm 2). The authors may argue that the escort policy gradients scales with the p-norm of theta, which does not require knowing the reward values; however, I think such argument would not be sound, because in all experiments that they consider, they work in a setting when all rewards are observed at each step, so there would be no problem for picking out the optimal action for scaling the policy gradient step (e.g. in classification one knows what the optimal label is; in the cross-entropy case, the minimum probability action was necessary, not the optimal, so knowing the rewards isn't even necessary; and in the MDP domains, they computed all Q-values exactly). If they had performed experiments with sampled actions, I would have less to argue about on this point, but currently there are no such experiments. To elaborate further on the learning rates, the main problem appears to be that the softmax gradient magnitude can vary greatly based on the policy probabilities. A natural fix would then be to normalize this gradient magnitude somehow, e.g. just normalize the gradient with the current magnitude, and perform fixed size steps in the direction of the gradient, or some other means of normalizing, which allows the magnitude to vary only based on the reward values. The limiting factor in learning is not the magnitude of the learning rate, but the magnitude of the steps taken in the parameter space, so if the gradient is very small, there is no issue with setting a large learning rate, but the current analysis of the limitations of softmax does not consider this option.

Correctness: The theory is performed under the assumption that standard gradient descent is used, but some of the experiments are done using Adam, or SGD with momentum, so the theory does not appear valid in those situations. E.g. In the four rooms MDP domain, they used Adam; in MNIST they used SGD with momentum instead. Probably it would be good to perform experiments both with vanilla SGD, as well as these other optimizers used in practice. But at least it should be mentioned that strictly speaking the theory does not appear valid for those experiments. In the appendix they wrote: "We tuned the hyper-parameters and use the learning rate 0.01 for SPG and EPG." I didn't understand whether this meant they tested many different learning rates for SPG and EPG, or whether they just picked 0.01 without tuning it (but tuned some other parameters). In any case, testing several learning rates would be necessary, and the optimal learning rates should be selected separately (or by picking the best learning rate for SPG, the baseline method they're comparing to). Also, the method of tuning was not explained.

Clarity: Generally, the clarity was very good. I make a few comments: For Figure 1, when reading the text, I was confused what the rewards were, and how the initialization was done. It may be good to mention that these are explained in the figure caption. L66 "the the" to "the" Eq 4 and 5 are correct, but please also add a derivation in the appendix. The description of the 4-room environment was ambiguous to me. In particular, what does sub-goal mean? Does the episode terminate if the agent goes to a sub-goal? After visiting a sub-goal once, is it possible to receive the reward again when visiting the same sub-goal again? If not, how is it ensured that the environment stays an MDP (is the state space modified with boolean variables for whether the sub-goal was visited)? How were the theta values initialized? They said that they initialized the policy probabilities for softmax and escort policies at the same values. But, there are multiple theta values that give the same policy probabilities for the escort policy, so the initialization is not fully described. Although, as far I've understood, the particular rate schedule and implementation they are using means that the scaling of the initialization does not matter. However, such things should be mentioned in the description. The significance of the Lojasiewicz coefficient could perhaps be explained better. In particular, what is the significance of the (1-xi) term in |f(x) - f(x*)|^{1-xi}? Why is this a useful concept to derive the bounds? L269 plateaux -> plateaus L277 plateaux -> plateaus What is SGW? I guess it means Softmax gravity well, but the acronym is not defined anywhere.

Relation to Prior Work: Mostly, I think this was done very well, except that they seemed to be unaware of the Neural Replicator Dynamics paper.

Reproducibility: Yes

Additional Feedback: ## Update ## Perhaps, the statement of Thm. 1 may be better stated that there exists a reward vector, r, for each initialization of pi, s.t. the poor performance happens. (In the proof you actually pick a specific r, rather than a specific pi.) I think this makes more sense compared to the current one which says that there exists an initialization of pi (actually, the proof does not currently consider arbitrary structures of, r, so it is not clear to me that the statement as it is right now is correct). Line 123: "This problem is unavoidable if using SPG to perform updates (Theorem 1)." : I think such statements should explain the assumptions (e.g. bounded learning rates, non-adaptive learning rates) For the proofs, it may help the reader to describe the high-level proof strategy before going into the details, for proofs that span more than a page. It may help to explain more the significance of the terms (for example the reward gap could be explained more, given that the derivation did not consider arbitrary r, but used a specific r, where all actions were equally worse than the optimal one). Line 401: t1 seems like a typo. In Eq. 31 in the appendix, where did the 1 dot 1 come from? Where did the r disappear? I guess you assumed r is in [0, 1]. (After looking through it again, I see that r in [0, 1]^K was defined earlier, but reminding the reader may help) Lemma 4: The way you have written it, mu has to be a stationary distribution (not initial state distribution) for the theorem to be correct. If you consider a distribution that changes in each time-step, the discounted state visitation distribution has to be used. For example see (https://arxiv.org/abs/1906.07073) for a recent explanation of this. From Eq 73, I see that you are aware of this, but probably, the Lemma was just not written accurately (d should probably be defined fully as the discounted state visitation distribution). Lemma 7, what is rho? It may be good to make Theorem statements self-contained or give pointers to definitions. ######################### My main concern is the issue with the learning rate schedule. If there is some easy way to clear that up, then please let me know. In particular, I think that unless I've misunderstood something, such limitations of the analysis should be explained, and explored in more detail either theoretically or experimentally. Even if the proofs would not work out with adaptive learning rates, at least they would say that softmax is no good if the learning rates are constant/bounded, which would also be an insightful contribution. Also, I am interested in a response to the issues brought up in the correctness section, as well as the description of the 4 rooms domain. If there is any other major issue, and you have space to respond then please do. I think there are many directions in which the paper could be improved, e.g. comparison with Neural Replicator Dynamics, more elaborate experiments on larger problems with more realistic settings, greater explanation of the Lojasiewicz coefficient, and how this relates to the escape time in general settings.


Review 4

Summary and Contributions: This paper provides a theoretical analysis of why softmax causes difficulties for gradient descent: (1) sensitivity to parameter initialization (2) slow convergence when used with cross-entropy loss. Both phenomena are well explained under the idea of Lojasiewicz coefficient. The authors show that suboptimal points are easy to attract the optimization trajectory and L. coefficient of CE may vanish through training. The authors propose an alternative transform, ‘escort mapping’, and verified effectiveness with several simple experiments.

Strengths: Theoretical findings are well presented with proper visualization and explanation (especially with “softmax gravity well”. The work is based on several related works [1][9] and successfully extend the discussion. The analysis of the softmax function can be directly applied to many deep learning areas and seems to provide a solid explanation of why current RL is too difficult to train and too sensitive to random seed.

Weaknesses: How is Escort transform implemented with neural networks? Is \theta the weight of the last linear (fully connected) layer for classification?

Correctness: Theoretical flow seems to be correct, but I am not sure about the details.

Clarity: The paper is clearly written, and experiments properly support the idea.

Relation to Prior Work: The connection to previous works is well established. The paper extends the previous works on the convergence of MDPs and convergence rate and suggests a unified view of L. coefficient.

Reproducibility: Yes

Additional Feedback: An explanation of the effect of “p” escort norm would enrich the results. For example, what is expected when we increase ‘p’ from 2 to 4? ---- I've read the rebuttal and other reviews. I increased my score because the novelty and the impact seem to be top 50%.

[Author Response · NeurIPS 2020]

We thank the reviewers for careful reading and valuable comments.

**Additional experiments:** We emphasize that our main contributions are theoretical
rather than empirical, and the contributions are novel and substantive. As the reviewers
acknowledged, there are practical implications and we are undertaking systematic
experiments as follow-up. For example, Figure 1 gives an additional experiments on an
OpenAI Gym Algorithmic task, showing notable improvements of escort over softmax
in a more complicated sampled-action setting. Here, REINFORCE PG is used and
policies are parameterized by recurrent neural networks with $256$ hidden LSTM units.

Figure 1: *Gym RepeatCopy-v0*

**R1:** (**i**) Note that the escort is differentiable for $p \geq 2$. (**ii**) The y-axis is $\log T$ such that
$\pi_{\theta_T}(a^*) \geq 0.99$. (**iii**) In Thms 2&3, the constant $K^{1/p}$ becomes worse when $p \to 1$.

**R2:** (**i**) To our knowledge, our results are novel in both RL and SL, although RL is our main motivation and focus. (**ii**)
For $p \geq 2$ we do not observe jitter with the escort, due to smoothness. (**iii**) It is not our intent to claim escort is uniformly
better than softmax (or the best possible), but by showing its provable benefits we reveal an under-explored opportunity
that can hopefully inspire future work. That said, we will add a discussion to potential benefits of softmax in the final
version. (**iv**) Uniform initialization to $1/K$ is common both in the RL literature and beyond. We appreciate it if the
reviewer has a suggestion for reference. While in Fig. 6(a)-(b), SPG plateaus even if the initialization is nearly uniform.

**R3:** *NRD paper:* Thanks for pointing it out, which we will cite and discuss. That paper makes a similar observation
that PG has an action-dependent scaling factor dependent on the current policy, which can slow down update dynamics.
They focus on the interesting but different multi-agent setting, and no convergence rate analysis for PG is provided.
Furthermore, (**i**) the theoretical overlap between that paper and ours is minimal; (**ii**) MD is discussed in Remark 2, and
Fig. 3(b,c) show MD is similar to $p = 2$ in that case; (**iii**) the NRD paper did not contemplate the escort transform, which
requires non-trivial technical novelty to analyze. We will give a detailed comparison and discussion in the final version.

*ICML2020 paper of Mei et al.:* That paper only makes an observation, without a deeper look at the causes, analysis or
solutions to fix it. These are the main contributions of the our submission.

*Experiments:* We tried EPG in sampled-action versions of the experiments, with or without Adam, and it outperformed
SPG. These results and also details for hyper-parameter search (mainly for learning rate and batch-size) will be added.

*Why the escort transform:* This transform is a natural choice due to its simplicity, and has a history in the physics
literature [2]. Empirical evidence in SL shows that escort with $p = 2$ performs better than softmax in MNIST and
CIFAR-10 [5]. We will add more intuition and motivation to Sec. 3 in the final version.

*Łojasiewicz coefficient:* Given current analysis, we believe that for any function that satisfies properties like Lemmas 2–3,
a conclusion like Thm. 2 follows by a similar derivation, so a simple answer to the question is yes. We can discuss a
more general characterization, but this is of independent interest and is outside the scope of of the paper.

*Learning rate intuitions:* We appreciate the attempts to understand the effect of learning rate on convergence of SPG
and EPG and we went through a similar process. We will be happy to discuss these in the paper. Unfortunately, none of
the alternatives suggested appear to be viable in their current forms:

*Learning rate analysis for SPG:* (**i**) A large/unbounded $\eta$ is not guaranteed to produce monotonic improvement, which
is a basic convergence requirement; e.g., [1, Thm. 5.1] requires $\eta < 1$. (**ii**) Thm. 1 applies to a provable update, since
SPG with $\eta = 0.4$ has $O(1/t)$ rate [11, Thm. 2]. (**iii**) For MDPs, the coefficient is $\min_{s \in \mathcal{S}} \pi_{\theta_t}(a^*(s)|s)$ (Line 498 in the
appendix, also [11, Thm. 4, Eq. (317)]), which cannot be calculated even from the true $Q^\pi$-values since $a^*$ is determined
by $Q^*$ not $Q^\pi$. (**iv**) For the alternative learning rate approaches suggested ($1/\pi_{\theta_t}(a^*)$ and $\ell_2$-norm normalized SPG),
we conducted experiments similar to Fig. 5(a). Both suggestions fail for $K = 50$ or $100$ (plateaus after $10^5$ iterations).

*Learning rate analysis for EPG:* It is easy to establish that $\|\theta_t\|_p$ is finitely bounded from above and below. First,
$\|\theta_t\|_p \geq |\theta_t(a^*)|$, and Eq. (5) implies $|\theta_t(a^*)| \geq |\theta_{t-1}(a^*)| \geq |\theta_1(a^*)|$. Second, $\|\theta_t\|_p^p$ keeps decreasing after $\pi_{\theta_t}^\top r > c$,
where $c < r(a^*)$ depends on reward and initialization. Therefore the EPG learning rate cannot be "arbitrarily high". The
4-room and MNIST experiments work reasonably well using constant learning rates like $0.01$.

*Clarity & correctness:* (**i**) We will fix the typos and clarify the descriptions. (**ii**) At a (sub-)goal state, the agent can step
away then step back to receive rewards. "Sub-goal" just means goals with lower rewards. (**iii**) For escort initialization,
we use $\theta_1(a) = \pi_{\theta_1}(a)^{1/p}$ for all $a$. (**iv**) $\xi$ is defined in [11, Def. 1], which impacts the rates [11, Lemma 8, 16].

**R4:** (**i**) In 4-room and MNIST, we use Eq. (3), where $\theta$ is the output of the last hidden layer. (**ii**) Escort becomes closer
to softmax when $p$ increases (Remark 1). In Fig. 3(b), $p = 2$ is far from the sub-optimal corner than $p = 4$, and in
Fig. 3(c), $p = 4$ has a short "plateau" due to getting close to the sub-optimal corner. We will discuss further in the final
version.

[Meta-Review · NeurIPS 2020]

This paper is proposing alternative to common practices in machine learning: Softmax Policy Gradient for RL and softmax parameterization in classification when minimizing cross-entropy loss. The limitation of softmax in these two cases are well explained, and the paper will be interesting for a wide range of the NeurIPS community.